# TimeMixer++: A General Time Series Pattern Machine for Universal Predictive Analysis

**Shiyu Wang**[*♠], **Jiawei Li**[*1,2], **Xiaoming Shi, Zhou Ye, Baichuan Mo**[3]**, Wenze Lin**[4]**,**
**Shengtong Ju**[4]**, Zhixuan Chu**[†4,5,6]**, Ming Jin**[†1]

[1]Griffith University [2]The Hong Kong University of Science and Technology (Guangzhou)
[3]Massachusetts Institute of Technology [4]Zhejiang University
[5]The State Key Laboratory of Blockchain and Data Security
[6]Hangzhou High-Tech Zone (Binjiang) Institute of Blockchain and Data Security
`kwuking@gmail.com, jarvis-li@outlook.com, sxm728@hotmail.com`
`yezhou199032@gmail.com, baichuan@mit.edu, zhixuanchu@zju.edu.cn`
`{linwenze75, shengtongju, mingjinedu}@gmail.com`

## Abstract

Time series analysis plays a critical role in numerous applications, supporting tasks such as forecasting, classification, anomaly detection, and imputation. In this work, we present the *time series pattern machine* (TSPM), a model designed to excel in a broad range of time series tasks through powerful representation and pattern extraction capabilities. Traditional time series models often struggle to capture universal patterns, limiting their effectiveness across diverse tasks. To address this, we define multiple scales in the time domain and various resolutions in the frequency domain, employing various mixing strategies to extract intricate, task-adaptive time series patterns. Specifically, we introduce TimeMixer++, a general-purpose TSPM that processes multi-scale time series using (1) *multi-resolution time imaging* (MRTI), (2) *time image decomposition* (TID), (3) *multi-scale mixing* (MCM), and (4) *multi-resolution mixing* (MRM) to extract comprehensive temporal patterns. MRTI transforms multi-scale time series into multi-resolution time images, capturing patterns across both temporal and frequency domains. TID leverages dual-axis attention to extract seasonal and trend patterns, while MCM hierarchically aggregates these patterns across scales. MRM adaptively integrates all representations across resolutions. TimeMixer++ achieves state-of-the-art performance across 8 time series analytical tasks, consistently surpassing both general-purpose and task-specific models. Our work marks a promising step toward the next generation of TSPMs, paving the way for further advancements in time series analysis.

## 1 Introduction

Time series analysis is crucial for identifying and predicting temporal patterns across various domains, including weather forecasting (Bi et al., 2023), medical symptom classification (Kiyasseh et al., 2021), anomaly detection in spacecraft monitoring (Xu et al., 2022), and imputing missing data in wearable sensors (Wu et al., 2020). These diverse applications highlight the versatility and importance of time series analysis in addressing real-world challenges. A key advancement in this field is the development of *time series pattern machines* (TSPMs), which aim to create a unified model architecture capable of handling a broad range of time series tasks across domains (Zhou et al., 2023; Wu et al., 2023).

At the core of TSPMs is their ability to recognize and generalize time series patterns inherent in time series data, enabling the model to uncover meaningful temporal structures and adapt to varying time series task scenarios. A line of research (Lai et al., 2018b; Zhao et al., 2017) has utilized recurrent neural networks (RNNs) to capture sequential patterns. However, these methods often struggle to capture long-term dependencies due to limitations like Markovian assumptions and inefficiencies. Temporal convolutional networks (TCNs) (Wu et al., 2023; Wang et al., 2023a; Liu et al., 2022a; Wang et al., 2024a) efficiently capture local patterns but face challenges with long-range dependencies (e.g.,

---

* Equal contribution ♠ Project lead † Corresponding author

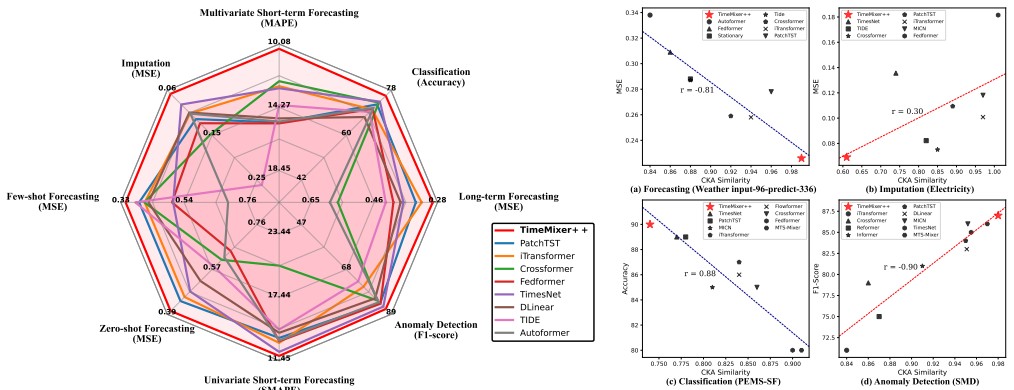

Figure 1: Benchmarking model performance across eight tasks **(left)** and representation analysis in four tasks **(right)**. For each model on the right, the centered kernel alignment (CKA) similarity (Kornblith et al., 2019) is computed between the representations from the first and last layers.

seasonality and trends) because of their fixed receptive fields. While some approaches reshape time series into 2D tensors based on frequency domain information (Wu et al., 2023) or downsample the time domain (Liu et al., 2022a), they fall short in comprehensively capturing long-range patterns. In contrast, transformer-based architectures (Nie et al., 2023; Liu et al., 2024; Zhou et al., 2022b; Wang et al., 2022; Shi et al., 2024) leverage token-wise self-attention to model long-range dependencies by allowing each token to attend to all others, overcoming the limitations of fixed receptive fields. However, unlike language tasks where tokens usually belong to distinct contexts, time series data often involve overlapping contexts at a single time point, such as daily, weekly, and seasonal patterns occurring simultaneously. This overlap makes it difficult to represent time series patterns effectively as tokens, posing challenges for transformer-based models in fully capturing the relevant temporal structures.

The recognition of the above challenges naturally raises a pivotal question:

> *What capabilities must a model possess, and what challenges must it overcome, to function as a TSPM?*

Before addressing the design of TSPMs, we first reconsider how time series are generated from continuous real-world processes sampled at various scales. For example, daily data capture hourly fluctuations, while yearly data reflect long-term trends and seasonal cycles. This multi-scale, multi-periodicity nature presents a significant challenge for model design, as each scale emphasizes different temporal dynamics that must be effectively captured. Figure 1 illustrates this challenge in constructing a general TSPM. Specifically, lower CKA similarity (Kornblith et al., 2019) indicates more diverse representations across layers, which is advantageous for tasks like imputation and anomaly detection that require capturing irregular patterns and handling missing data. In these cases, diverse representations across layers help manage variations across scales and periodicities. Conversely, forecasting and classification tasks benefit from higher CKA similarity, where consistent representations across layers better capture stable trends and periodic patterns. This contrast emphasizes the challenge of designing *a universal model flexible enough to adapt to multi-scale and multi-periodicity patterns across various analytical tasks, which may favor either diverse or consistent representations.*

To address the aforementioned question and challenges, we propose TIMEMIXER++, a general-purpose TSPM designed to capture general, task-adaptive time series patterns by tackling the complexities of multi-scale and multi-periodicity dynamics. The key idea is to simultaneously capture intricate time series patterns across multiple scales in the time domain and various resolutions in the frequency domain. Specifically, TIMEMIXER++ processes multi-scale time series using (1) *multi-resolution time imaging* (MRTI), (2) *time image decomposition* (TID), (3) *multi-scale mixing* (MCM), and (4) *multi-resolution mixing* (MRM) to uncover comprehensive patterns. MRTI transforms multi-scale time series into multi-resolution time images, enabling pattern extraction across both temporal and frequency domains. TID applies dual-axis attention to disentangle seasonal and trend patterns in the latent space, while MCM hierarchically aggregates these patterns across

different scales. Finally, MRM adaptively integrates all representations across resolutions. As shown in Figure 1, TIMEMIXER++ achieves state-of-the-art performance across 8 analytical tasks, outperforming both general-purpose and task-specific models. Its adaptability is reflected in its varying CKA similarity scores across different tasks, indicating its ability to capture diverse task-specific patterns more effectively than other models. Our contributions are summarized as follows:

1. We introduce TIMEMIXER++, a general-purpose time series analysis model that processes multi-scale, multi-periodicity data by transforming time series into multi-resolution time images, enabling efficient pattern extraction across both temporal and frequency domains.

2. To capture intricate patterns, we disentangle seasonality and trend from time images using time image decomposition, followed by adaptive aggregation through multi-scale mixing and multi-resolution mixing, enabling patterns integration across scales and periodicities.

3. TIMEMIXER++ sets a new state-of-the-art across 8 time series analytical tasks in different benchmarks, consistently outperforming both general-purpose and task-specific models. This marks a significant step forward in the development of next-generation TSPMs.

## 2 RELATED WORK

**Time Series Analysis.** A pivotal aspect of time series analysis is the ability to extract diverse patterns from various time series while building powerful representations. This challenge has been explored across various model architectures. Traditional models like ARIMA (Anderson & Kendall, 1976) and STL (Cleveland et al., 1990) are effective for periodic and trend patterns but struggle with non-linear dynamics. Deep learning models, such as those by (Lai et al., 2018b) and (Zhao et al., 2017), capture sequential dependencies but face limitations with long-term dependencies. TCNs (Franceschi et al., 2019) improve local pattern extraction but are limited in capturing long-range dependencies. TimesNet (Wu et al., 2023) enhances long-range pattern extraction by treating time series as 2D signals, while MLP-based methods (Zeng et al., 2023; Ekambaram et al., 2023; Liu et al., 2023; Wang et al., 2023c) offer simplicity and effectiveness. Transformer-based models like PatchTST (Nie et al., 2023) and iTransformer (Liu et al., 2024) leverage self-attention to model long-range dependencies, demonstrating good forecasting performance. Given the strengths and limitations discussed above, there is a growing need for a TSPM capable of effectively extracting diverse patterns, adapting to various time series analytical tasks, and possessing strong generalization capabilities. As illustrated in Figure 1, TIMEMIXER++ meets this requirement by constructing robust representational capabilities, thereby demonstrating its potential for universal time series analysis.

**Hierarchical Time Series Modeling.** Numerous methodologies have been advanced utilizing specialized deep learning architectures for time series analysis, with an emphasis on the decomposition and integration of temporal patterns. For example, several studies (Wu et al., 2021; Wang et al., 2024b; Zhou et al., 2022b; Luo et al., 2023; Wang et al., 2023b) utilize moving averages to discern seasonal and trend components, which are subsequently modeled using attention mechanisms (Wu et al., 2021; Zhou et al., 2022b; Shi et al., 2025), convolutional networks (Wang et al., 2023a), or hierarchical MLP layers (Wang et al., 2024b; Wang, 2024). These components are individually processed prior to aggregation to yield the final output. Nonetheless, such approaches frequently depend on predefined and rigid operations for the disentanglement of seasonality and trends, thereby constraining their adaptability to complex and dynamic patterns. In contrast, as depicted in Figure 2, we propose a more flexible methodology that disentangles seasonality and trend directly within the latent space via dual-axis attention, thereby enhancing adaptability to a diverse range of time series patterns and task scenarios. Furthermore, by adopting a multi-scale, multi-resolution analytical framework (Mozer, 1991; Harti, 1993), we facilitate hierarchical interaction and integration across different scales and resolutions, substantially enhancing the effectiveness of time series modeling.

## 3 TIMEMIXER++

Building on the multi-scale and multi-periodic characteristics of time series, we introduce TIMEMIXER++, a general-purpose time series pattern machine that processes multi-scale time series using an encoder-only architecture, as shown in Figure 2. The architecture generally comprises three components: (1) *input projection*, (2) a stack of *Mixerblocks*, and (3) *output projection*.

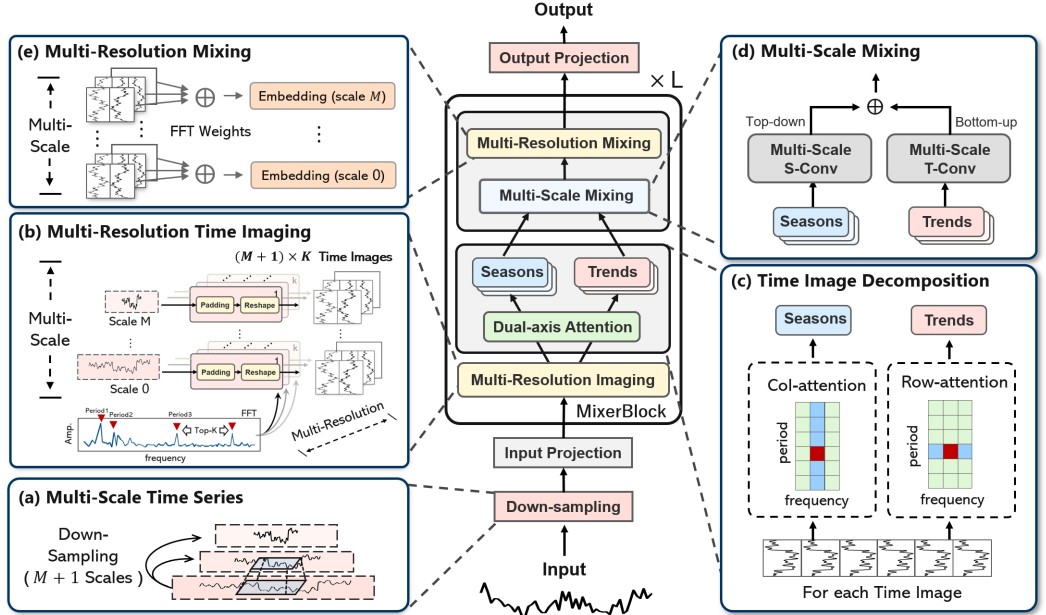

Figure 2: The framework of TIMEMIXER++. The multi-scale time series is first embedded through an input projection layer, followed by $L$ stacked *MixerBlocks*. Each block converts the multi-scale input into multi-resolution time images, disentangles seasonality and trend via dual-axis attention, and mixes these patterns using multi-scale and multi-resolution mixing.

**Multi-scale Time Series.** We approach time series analysis using a multi-scale framework. Given an input time series $\mathbf{x}_0 \in \mathbb{R}^{T \times C}$, where $T$ represents the sequence length and $C$ the number of variables, we generate a multi-scale representation through downsampling. Specifically, the input time series $\mathbf{x}_0$ is progressively downsampled across $M$ scales using convolution operations with a stride of $2^1$, producing the multi-scale set $\mathcal{X}_{init} = \{\mathbf{x}_0, \cdots, \mathbf{x}_M\}$, where $\mathbf{x}_m \in \mathbb{R}^{\lfloor \frac{T}{2^m} \rfloor \times C}$. The downsampling process follows the recursive relationship:

$$\mathbf{x}_m = \text{Conv}(\mathbf{x}_{m-1}, \text{stride} = 2), \quad m \in \{1, \cdots, M\}. \tag{1}$$

### 3.1 STRUCTURE OVERVIEW

**Input Projection.** Previous studies (2023; 2023) employ a channel-independence strategy to avoid projecting multiple variables into indistinguishable channels (Liu et al., 2024). In contrast, we adopt channel mixing to capture cross-variable interactions, which are crucial for revealing comprehensive patterns in time series data. The input projection has two components: channel mixing and embedding. We first apply self-attention to the variate dimensions at the coarsest scale $\mathbf{x}_M \in \mathbb{R}^{\lfloor \frac{T}{2^M} \rfloor \times C}$, as it retains the most global context, facilitating the more effective integration of information across variables. This is formulated as follows:

$$\mathbf{x}_M = \text{Channel-Attn}(\mathbf{Q}_M, \mathbf{K}_M, \mathbf{V}_M), \tag{2}$$

where $\text{Channel-Attn}$ denotes the variate-wise self-attention for channel mixing. The queries, keys, and values $\mathbf{Q}_M, \mathbf{K}_M, \mathbf{V}_M \in \mathbb{R}^{C \times \lfloor \frac{T}{2^M} \rfloor}$ are derived from linear projections of $\mathbf{x}_M$. Then, we embed all multi-scale time series into a deep pattern set $\mathcal{X}^0$ using an embedding layer, which can be expressed as $\mathcal{X}^0 = \{\mathbf{x}_0^0, \cdots, \mathbf{x}_M^0\} = \text{Embed}(\mathcal{X}_{init})$, where $\mathbf{x}_m^0 \in \mathbb{R}^{\lfloor \frac{T}{2^m} \rfloor \times d_{\text{model}}}$ and $d_{\text{model}}$ represents the dimensionality of the deep patterns.

**MixerBlocks.** Next, we apply a stack of $L$ *Mixerblocks* with the goal to capture intricate patterns across scales in the time domain and resolutions in the frequency domain. Within the *MixerBlocks*, we convert multi-scale time series into multi-resolution time images, disentangle seasonal and

---

[1]Setting stride = 2 maximizes the number of scales from recursive downsampling.

trend patterns through time image decomposition, and aggregate these patterns across different scales and resolutions. The forward propagation is defined as $\mathcal{X}^{l+1} = \text{MixerBlock}(\mathcal{X}^l)$, where $\mathcal{X}^l = \{\mathbf{x}_0^l, \cdots, \mathbf{x}_M^l\}$ and $\mathbf{x}_m^l \in \mathbb{R}^{\lfloor \frac{T}{2^m} \rfloor \times d_{\text{model}}}$. We will elaborate on this block in the next section.

**Output Projection.** After $L \times$ *MixerBlocks*, we obtain the multi-scale representation set $\mathcal{X}^L$. Since different scales capture distinct temporal patterns and tasks vary in demands, as discussed in Section 1, we propose using multiple prediction heads, each specialized for a specific scale, and ensembling their outputs. This design is task-adaptive, allowing each head to focus on relevant features at its scale, while the ensemble aggregates complementary information to enhance prediction robustness.

$$output = \text{Ensemble}(\{\text{Head}_m(\mathbf{x}_m^L)\}_{m=0}^M), \tag{3}$$

where $\text{Ensemble}(\cdot)$ denotes the ensemble method (e.g., averaging or weighted sum), and $\text{Head}_m(\cdot)$ is the prediction head for the $m$-th scale, typically a linear layer.

## 3.2 MIXERBLOCK

We organize a stack of *MixerBlocks* in a residual way. For the $(l+1)$-th block, the input is the multi-scale representation set $\mathcal{X}^l$, and the forward propagation can be formalized as:

$$\mathcal{X}^{l+1} = \text{LayerNorm}\left(\mathcal{X}^l + \text{MixerBlock}\left(\mathcal{X}^l\right)\right), \tag{4}$$

where $\text{LayerNorm}$ normalizes patterns across scales and can stabilize the training. Time series exhibits complex multi-scale and multi-periodic dynamics. Multi-resolution analysis (Harti, 1993) models time series as a composite of various periodic components in the frequency domain. We introduce multi-resolution time images, converting 1D multi-scale time series into 2D images based on frequency analysis while preserving the original data. This captures intricate patterns across time and frequency domains, enabling efficient use of convolution methods for extracting temporal patterns and enhancing versatility across tasks. Specifically, we processes multi-scale time series using (1) *multi-resolution time imaging* (MRTI), (2) *time image decomposition* (TID), (3) *multi-scale mixing* (MCM), and (4) *multi-resolution mixing* (MRM) to uncover comprehensive time series patterns.

**Multi-Resolution Time Imaging.** At the start of each *MixerBlock*, we convert the input $\mathcal{X}^l$ into $(M+1) \times K$ multi-resolution time images via frequency analysis (Wu et al., 2023). To capture representative periodic patterns, we first identify periods from the coarsest scale $\mathbf{x}_M^l$, which enables global interaction. Specifically, we apply the fast fourier transform (FFT) on $\mathbf{x}_M^l$ and select the top-$K$ frequencies with the highest amplitudes:

$$\mathbf{A}, \{f_1, \cdots, f_K\}, \{p_1, \cdots, p_K\} = \text{FFT}(\mathbf{x}_M^l), \tag{5}$$

where $\mathbf{A} = \{A_{f_1}, \cdots, A_{f_K}\}$ represents the unnormalized amplitudes, $\{f_1, \cdots, f_K\}$ are the top-$K$ frequencies, and $p_k = \left\lceil \frac{T}{f_k} \right\rceil, k \in \{1, \ldots, K\}$ denotes the corresponding period lengths. Each time series representation $\mathbf{x}_m^l$ is then reshaped along the temporal dimension as follows:

$$\begin{aligned} \text{MRTI}(\mathcal{X}^l) = \{\mathscr{Z}_m^l\}_{m=0}^M &= \left\{ \mathbf{z}_m^{(l,k)} \mid m = 0, \ldots, M; \; k = 1, \ldots, K \right\} \\ &= \left\{ \underset{1D \rightarrow 2D}{\text{Reshape}_{m,k}}(\text{Padding}_{m,k}(\mathbf{x}_m^l)) \mid m = 0, \ldots, M; \; k = 1, \ldots, K \right\}, \end{aligned} \tag{6}$$

where $\text{Padding}_{m,k}(\cdot)$ zero-pads the time series to a length of $p_k \cdot \lceil \frac{\lfloor \frac{T}{2^m} \rfloor}{p_k} \rceil$, and $\underset{1D \rightarrow 2D}{\text{Reshape}_{m,k}}(\cdot)$ converts it into a $p_k \times \lceil \frac{\lfloor \frac{T}{2^m} \rfloor}{p_k} \rceil$ image, denoted as $\mathbf{z}_m^{(l,k)}$. Here, $p_k$ represents the number of rows (period length), and the number of columns, denoted by $f_{\text{m,k}} = \lceil \frac{\lfloor \frac{T}{2^m} \rfloor}{p_k} \rceil$, represent the corresponding frequency for scale $m$.

**Time Image Decomposition.** Time series patterns are inherently nested, with overlapping scales and periods. For example, weekly sales data reflects both daily shopping habits and broader seasonal trends. Conventional methods (Wu et al., 2021; Wang et al., 2024b) use moving averages across the entire series, often blurring distinct patterns. To address this, we utilize multi-resolution time

images, where each image $\mathbf{z}_m^{(l,k)} \in \mathbb{R}^{p_k \times f_{\mathrm{m,k}} \times d_{\mathrm{model}}}$ encodes a specific scale and period, enabling finer disentanglement of seasonality and trend. By applying 2D convolution to these images, we capture long-range patterns and enhance temporal dependency extraction. Columns in each image correspond to time series segments within a period, while rows represent consistent time points across periods, facilitating dual-axis attention: column-axis attention ($\mathrm{Attention}_{\mathrm{col}}$) captures seasonality within periods, and row-axis attention ($\mathrm{Attention}_{\mathrm{row}}$) extracts trend across periods. Each axis-specific attention focuses on one axis, preserving efficiency by transposing the non-target axis to the batch dimension. For column-axis attention, queries, keys, and values $\mathbf{Q}_{\mathrm{col}}, \mathbf{K}_{\mathrm{col}}, \mathbf{V}_{\mathrm{col}} \in \mathbb{R}^{f_{\mathrm{m,k}} \times d_{\mathrm{model}}}$ are computed via 2D convolution, which are shared across all images, and similarly for row-axis attention $\mathbf{Q}_{\mathrm{row}}, \mathbf{K}_{\mathrm{row}}, \mathbf{V}_{\mathrm{row}}$. The seasonal and trend components are then computed as:

$$\mathbf{s}_m^{(l,k)} = \mathrm{Attention}_{\mathrm{col}}(\mathbf{Q}_{\mathrm{col}}, \mathbf{K}_{\mathrm{col}}, \mathbf{V}_{\mathrm{col}}), \quad \mathbf{t}_m^{(l,k)} = \mathrm{Attention}_{\mathrm{row}}(\mathbf{Q}_{\mathrm{row}}, \mathbf{K}_{\mathrm{row}}, \mathbf{V}_{\mathrm{row}}), \quad (7)$$

where $\mathbf{s}_m^{(l,k)}, \mathbf{t}_m^{(l,k)} \in \mathbb{R}^{p_k \times f_{\mathrm{m,k}} \times d_{\mathrm{model}}}$ represent the seasonal and trend images, respectively. Here, the transposed axis is restored to recover the original image size after the attention.

**Multi-scale Mixing.** For each period $p_k$, we obtain $M + 1$ seasonal time images and $M + 1$ trend time images, denoted as $\{\mathbf{s}_m^{(l,k)}\}_{m=0}^M$ and $\{\mathbf{t}_m^{(l,k)}\}_{m=0}^M$, respectively. The 2D structure allows us to model both seasonal and trend patterns using 2D convolutional layers, which are more efficient and effective at capturing long-term dependencies than traditional linear layers (Wang et al., 2024b). For multi-scale seasonal time images, longer patterns can be interpreted as compositions of shorter ones (e.g., a yearly rainfall pattern formed by monthly changes). Therefore, we mix the seasonal patterns from fine-scale to coarse-scale. To facilitate this bottom-up information flow, we apply the 2D convolutional layers at the $m$-th scale in a residual manner, formalized as:

$$\text{for } m: 1 \to M \text{ do:} \quad \mathbf{s}_m^{(l,k)} = \mathbf{s}_m^{(l,k)} + \text{2D-Conv}(\mathbf{s}_{m-1}^{(l,k)}), \quad (8)$$

where 2D-Conv is composed of two 2D convolutional layers with a temporal stride of 2. Unlike seasonal patterns, for multi-scale trend time images, coarser scales naturally highlight the overall trend. Therefore, we adopt a top-down mixing strategy and apply the 2D transposed convolutional layer at the $m$-th scale in a residual manner, formalized as:

$$\text{for } m: M - 1 \to 0 \text{ do:} \quad \mathbf{t}_m^{(l,k)} = \mathbf{t}_m^{(l,k)} + \text{2D-TransConv}(\mathbf{t}_{m+1}^{(l,k)}), \quad (9)$$

where 2D-TransConv is composed of two 2D transposed convolutional layers with a temporal stride of 2. After mixing, the seasonal and trend patterns are aggregated via summation and reshaped back to a 1D structure, as follows:

$$\mathbf{z}_m^{(l,k)} = \underset{2D \to 1D}{\mathrm{Reshape}_{m,k}}(\mathbf{s}_m^{(l,k)} + \mathbf{t}_m^{(l,k)}), \quad m \in \{0, \cdots, M\}, \quad (10)$$

where $\underset{2D \to 1D}{\mathrm{Reshape}_{m,k}}(\cdot)$ convert a $p_k \times f_{m,k}$ image into a time series of length $p_k \cdot f_{m,k}$.

**Multi-resolution Mixing.** Finally, at each scale, we mix the $K$ periods adaptively. The amplitudes $\mathbf{A}$ capture the importance of each period, and we aggregate the patterns $\{\mathbf{z}_m^{(l,k)}\}_{k=1}^K$ as follows:

$$\{\hat{\mathbf{A}}_{f_k}\}_{k=1}^K = \mathrm{Softmax}(\{\mathbf{A}_{f_k}\}_{k=1}^K), \quad \mathbf{x}_m^l = \sum_{k=1}^K \hat{\mathbf{A}}_{f_k} \circ \mathbf{z}_m^{(l,k)}, \quad m \in \{0, \cdots, M\}, \quad (11)$$

where $\mathrm{Softmax}$ normalizes the weights, and $\circ$ denotes element-wise multiplication.

## 4 EXPERIMENTS

To verify the effectiveness of the proposed TIMEMIXER++ as a general time series pattern machine, we perform extensive experiments across 8 well-established analytical tasks, including (1) long-term forecasting, (2) univariate and (3) multivariate short-term forecasting, (4) imputation, (5) classification, (6) anomaly detection, as well as (7) few-shot and (8) zero-shot forecasting. Overall, as summarized in Figure 1, **TIMEMIXER++ consistently surpasses contemporary state-of-the-art models in a range of critical time series analysis tasks, which is demonstrated by its superior performance across 30 well-known benchmarks and against 27 advanced baselines**. The detailed experimental configurations and implementations are in Appendix A.

## 4.1 MAIN RESULTS

### 4.1.1 LONG-TERM FORECASTING

**Setups.** Long-term forecasting is pivotal for strategic planning in areas such as weather prediction, traffic management, and energy utilization. To comprehensively assess our model's effectiveness over extended periods, we perform experiments on 8 widely-used real-world datasets, including the four subsets of the ETT datasets (ETTh1, ETTh2, ETTm1, ETTm2), as well as Weather, Solar-Energy, Electricity, and Traffic, consistent with prior benchmarks set by Zhou et al. (2021b); Liu et al. (2022a).

Table 1: Long-term forecasting results. We average the results across 4 prediction lengths: $\{96, 192, 336, 720\}$. The best performance is highlighted in **red**, and the second-best is underlined. Full results can be found in Appendix H.

| Models | TimeMixer++ (Ours) | | TimeMixer (2024b) | | iTransformer (2024) | | PatchTST (2023) | | Crossformer (2023) | | TiDE (2023a) | | TimesNet (2023) | | DLinear (2023) | | SCINet (2022a) | | FEDformer (2022b) | | Stationary (2022c) | | Autoformer (2021) | |
|---|---|---|---|---|---|---|---|---|---|---|---|---|---|---|---|---|---|---|---|---|---|---|---|---|
| Metric | MSE | MAE | MSE | MAE | MSE | MAE | MSE | MAE | MSE | MAE | MSE | MAE | MSE | MAE | MSE | MAE | MSE | MAE | MSE | MAE | MSE | MAE | MSE | MAE |
| Electricity | **0.165** | **0.253** | 0.182 | 0.272 | 0.178 | 0.270 | 0.205 | 0.290 | 0.244 | 0.334 | 0.251 | 0.344 | 0.192 | 0.295 | 0.212 | 0.300 | 0.268 | 0.365 | 0.214 | 0.327 | 0.193 | 0.296 | 0.227 | 0.338 |
| ETT (Avg) | **0.349** | 0.399 | 0.367 | 0.388 | 0.383 | **0.377** | 0.381 | 0.397 | 0.685 | 0.578 | 0.482 | 0.470 | 0.391 | 0.404 | 0.442 | 0.444 | 0.689 | 0.597 | 0.408 | 0.428 | 0.471 | 0.464 | 0.465 | 0.459 |
| Exchange | 0.357 | 0.391 | 0.391 | 0.453 | 0.378 | **0.360** | 0.403 | 0.404 | 0.940 | 0.707 | 0.370 | 0.413 | 0.416 | 0.443 | 0.354 | 0.414 | 0.750 | 0.626 | 0.519 | 0.429 | 0.461 | 0.454 | 0.613 | 0.539 |
| Traffic | **0.416** | **0.264** | 0.484 | 0.297 | 0.428 | 0.282 | 0.481 | 0.304 | 0.550 | 0.304 | 0.760 | 0.473 | 0.620 | 0.336 | 0.625 | 0.383 | 0.804 | 0.509 | 0.610 | 0.376 | 0.624 | 0.340 | 0.628 | 0.379 |
| Weather | **0.226** | **0.262** | 0.240 | 0.271 | 0.258 | 0.278 | 0.259 | 0.281 | 0.259 | 0.315 | 0.271 | 0.320 | 0.259 | 0.287 | 0.265 | 0.317 | 0.292 | 0.363 | 0.309 | 0.360 | 0.288 | 0.314 | 0.338 | 0.382 |
| Solar-Energy | **0.203** | **0.238** | 0.216 | 0.280 | 0.233 | 0.262 | 0.270 | 0.307 | 0.641 | 0.639 | 0.347 | 0.417 | 0.301 | 0.319 | 0.330 | 0.401 | 0.282 | 0.375 | 0.291 | 0.381 | 0.261 | 0.381 | 0.885 | 0.711 |

**Results.** Table 1 shows TIMEMIXER++ outperforms other models in long-term forecasting across various datasets. On Electricity, it surpasses iTransformer by **7.3%** in MSE and **6.3%** in MAE. For ETT (Avg), TIMEMIXER++ achieves **4.9%** lower MSE than TimeMixer. On the challenging Solar-Energy dataset (Table 8), it exceeds the second-best model by **6.0%** in MSE and **9.2%** in MAE, demonstrating its robustness in handling complex high-dimensional time series.

### 4.1.2 UNIVARIATE SHORT-TERM FORECASTING

**Setups.** Univariate short-term forecasting is crucial for demand planning and marketing. We evaluate our model using the M4 Competition dataset Makridakis et al. (2018), comprising $100,000$ marketing time series with six frequencies from hourly to yearly, enabling comprehensive assessment across varied temporal resolutions.

Table 2: Univariate short-term forecasting results, averaged across all M4 subsets. Full results are available in Appendix H

| Models | TimeMixer++ (Ours) | TimeMixer (2024b) | iTransformer (2024) | TiDE (2023a) | TimesNet (2023) | N-HiTS (2023) | N-BEATS (2019) | PatchTST (2023) | MICN (2023a) | FiLM (2022a) | LightTS (2022a) | DLinear (2023) | FED. (2022b) | Stationary (2022c) | Auto. (2021) |
|---|---|---|---|---|---|---|---|---|---|---|---|---|---|---|---|
| SMAPE | **11.448** | 11.723 | 12.684 | 13.950 | 11.829 | 11.927 | 11.851 | 13.152 | 19.638 | 14.863 | 13.525 | 13.639 | 12.840 | 12.780 | 12.909 |
| MASE | **1.487** | 1.559 | 1.764 | 1.940 | 1.585 | 1.613 | 1.559 | 1.945 | 5.947 | 2.207 | 2.111 | 2.095 | 1.701 | 1.756 | 1.771 |
| OWA | **0.821** | 0.840 | 0.929 | 1.020 | 0.851 | 0.861 | 0.855 | 0.998 | 2.279 | 1.125 | 1.051 | 1.051 | 0.918 | 0.930 | 0.939 |

**Results.** Table 2 demonstrates that TimeMixer++ significantly outperforms state-of-the-art models across all metrics. Compared to iTransformer, it reduces SMAPE by **9.7%** and MASE by **15.7%**, with even larger improvements over TiDE, achieving up to a **23.3%** reduction in MASE. Additionally, TimeMixer++ records the lowest OWA, outperforming TimesNet by **3.5%** and iTransformer by **11.6%**.

### 4.1.3 MULTIVARIATE SHORT-TERM FORECASTING

**Setups.** We further evaluate the short-term forecasting performance in multivariate settings on the PeMS benchmark (Chen et al., 2001), which includes four publicly available high-dimensional traffic network datasets: PEMS03, PEMS04, PEMS07, and PEMS08. These datasets feature a large number of variables, ranging from 170 to 883, offering a challenging testbed for assessing the scalability and effectiveness of our model in predicting complex time series patterns across multiple variables.
**Results.** The results in Table 3 highlight the superior performance of TIMEMIXER++ across all key metrics in multivariate short-term forecasting. TIMEMIXER++ achieves a **19.9%** reduction in MAE

Table 3: Results of multivariate short-term forecasting, averaged across all PEMS datasets. Full results can be found in Table 18 of Appendix H.

| Models | TimeMixer++ (Ours) | TimeMixer (2024b) | iTransformer (2024) | TiDE (2023a) | SCINet (2022a) | Crossformer (2023) | PatchTST (2023) | TimesNet (2023) | MICN (2023a) | DLinear (2023) | FEDformer (2022b) | Stationary (2022c) | Autoformer (2021) |
|---|---|---|---|---|---|---|---|---|---|---|---|---|---|
| MAE | **15.91** | 17.41 | 19.87 | 21.86 | 19.12 | 19.03 | 23.01 | 20.54 | 19.34 | 23.31 | 23.50 | 21.32 | 22.62 |
| MAPE | **10.08** | 10.59 | 12.55 | 13.80 | 12.24 | 12.22 | 14.95 | 12.69 | 12.38 | 14.68 | 15.01 | 14.09 | 14.89 |
| RMSE | **27.06** | 28.01 | 31.29 | 34.42 | 30.12 | 30.17 | 36.05 | 33.25 | 30.40 | 37.32 | 36.78 | 36.20 | 34.49 |

and a **19.6%** reduction in MAPE compared to iTransformer, and an **8.6%** and **4.8%** reduction in MAE and MAPE, respectively, compared to TimeMixer. Notably, PatchTST, a strong baseline, is outperformed by TIMEMIXER++ with a **30.8%** improvement in MAE, **32.5%** in MAPE, and **24.9%** in RMSE, highlighting the effectiveness of TIMEMIXER++ in handling high-dimensional datasets.

### 4.1.4 IMPUTATION

**Setups.** Accurate imputation of missing values is crucial in time series analysis, affecting predictive models in real-world applications. To evaluate our model's imputation capabilities, we use datasets from electricity and weather domains, selecting ETT (Zhou et al. (2021b)), Electricity (UCI), and Weather (Wetterstation) as benchmarks.

Table 4: Results of imputation task across six datasets. To evaluate our model performance, we randomly mask $\{12.5\%, 25\%, 37.5\%, 50\%\}$ of the time points in time series of length 1024. The final results are averaged across these 4 different masking ratios.

| Models | TimeMixer++ (Ours) | | TimeMixer (2024b) | | iTransformer (2024) | | PatchTST (2023) | | Crossformer (2023) | | FEDformer (2022b) | | TIDE (2023a) | | DLinear (2023) | | TimesNet (2023) | | MICN (2023a) | | Autoformer (2021) | |
|---|---|---|---|---|---|---|---|---|---|---|---|---|---|---|---|---|---|---|---|---|---|
| Metric | MSE | MAE | MSE | MAE | MSE | MAE | MSE | MAE | MSE | MAE | MSE | MAE | MSE | MAE | MSE | MAE | MSE | MAE | MSE | MAE | MSE | MAE |
| ETT(Avg) | **0.055** | **0.154** | 0.097 | 0.220 | 0.096 | 0.205 | 0.120 | 0.225 | 0.150 | 0.258 | 0.124 | 0.230 | 0.314 | 0.366 | 0.115 | 0.229 | 0.079 | 0.182 | 0.119 | 0.234 | 0.104 | 0.215 |
| ECL | **0.109** | **0.197** | 0.142 | 0.261 | 0.140 | 0.223 | 0.129 | 0.198 | 0.125 | 0.204 | 0.181 | 0.314 | 0.182 | 0.202 | 0.080 | 0.200 | 0.135 | 0.255 | 0.138 | 0.246 | 0.141 | 0.234 |
| Weather | **0.049** | **0.078** | 0.091 | 0.114 | 0.095 | 0.102 | 0.082 | 0.149 | 0.150 | 0.111 | 0.064 | 0.139 | 0.063 | 0.131 | 0.071 | 0.107 | 0.061 | 0.098 | 0.075 | 0.126 | 0.066 | 0.107 |

**Results.** Table 4 presents TIMEMIXER++'s performance in imputing missing values across six datasets. It consistently outperforms all baselines, achieving the lowest MSE and MAE in the majority of cases. Compared to the second-best model, TimesNet, TIMEMIXER++ reduces MSE by an average of **25.7%** and MAE by **17.4%**. Notably, TIMEMIXER++ excels even when handling imputation tasks with input lengths of up to 1024, demonstrating its robust capability as a TSPM.

### 4.1.5 FEW-SHOT FORECASTING

**Setups.** Transformer-based models excel in various forecasting scenarios, especially with limited data. To evaluate their transferability and pattern recognition, we test across 6 diverse datasets, training each model on only 10% of available timesteps. This approach assesses adaptability to sparse data and the ability to discern general patterns, which is crucial for real-world predictive analysis where data is often limited.

Table 5: Few-shot learning on 10% training data. All results are averaged from 4 prediction lengths: $\{96, 192, 336, 720\}$.

| Models | TimeMixer++ (Ours) | | TimeMixer (2024b) | | iTransformer (2024) | | TiDE (2023a) | | Crossformer (2023) | | DLinear (2023) | | PatchTST (2023) | | TimesNet (2023) | | FEDformer (2022b) | | Autoformer (2021) | | Stationary (2022c) | | ETSformer (2022) | | LightTS (2022b) | | Informer (2021b) | | Reformer (2020) | |
|---|---|---|---|---|---|---|---|---|---|---|---|---|---|---|---|---|---|---|---|---|---|---|---|---|---|---|---|---|---|---|
| Metric | MSE | MAE | MSE | MAE | MSE | MAE | MSE | MAE | MSE | MAE | MSE | MAE | MSE | MAE | MSE | MAE | MSE | MAE | MSE | MAE | MSE | MAE | MSE | MAE | MSE | MAE | MSE | MAE | MSE | MAE |
| ETT(Avg) | **0.396** | **0.421** | 0.453 | 0.445 | 0.458 | 0.497 | 0.432 | 0.444 | 0.470 | 0.471 | 0.506 | 0.484 | 0.461 | 0.446 | 0.586 | 0.496 | 0.573 | 0.532 | 0.834 | 0.663 | 0.627 | 0.510 | 0.875 | 0.687 | 1.497 | 0.875 | 2.408 | 1.146 | 2.535 | 1.191 |
| Weather | **0.241** | **0.271** | 0.242 | 0.281 | 0.291 | 0.331 | 0.249 | 0.291 | 0.267 | 0.306 | 0.241 | 0.283 | 0.242 | 0.279 | 0.279 | 0.301 | 0.284 | 0.324 | 0.300 | 0.342 | 0.318 | 0.323 | 0.318 | 0.360 | 0.289 | 0.322 | 0.597 | 0.495 | 0.546 | 0.469 |
| ECL | **0.168** | **0.271** | 0.187 | 0.277 | 0.241 | 0.337 | 0.196 | 0.289 | 0.214 | 0.308 | 0.180 | 0.280 | 0.180 | 0.273 | 0.323 | 0.392 | 0.346 | 0.427 | 0.431 | 0.478 | 0.444 | 0.480 | 0.660 | 0.617 | 0.441 | 0.489 | 1.195 | 0.891 | 0.965 | 0.768 |

In few-shot learning, TIMEMIXER++ achieves superior performance across all datasets, reducing MSE by **13.2%** compared to PatchTST. DLinear performs well on some datasets but degrades in zero-shot experiments, suggesting overfitting. TIMEMIXER++ outperforms TimeMixer with **9.4%** lower MSE and **4.6%** lower MAE, attributed to its attention mechanisms enhancing general time series pattern recognition.

#### 4.1.6 ZERO-SHOT FORECASTING

**Setups.** We explore zero-shot learning to evaluate models' ability to generalize across different contexts. As shown in Table 6, models trained on dataset $D_a$ are evaluated on unseen dataset $D_b$ without further training. This direct transfer ($D_a \rightarrow D_b$) tests models' adaptability and predictive robustness across disparate datasets.

Table 6: Zero-shot learning results. The results are averaged from 4 different prediction lengths: $\{96, 192, 336, 720\}$.

| Methods | TimeMixer++ (Ours) | | TimeMixer (2024b) | | LLMTime (2023) | | DLinear (2023) | | PatchTST (2023) | | TimesNet (2023) | | iTransformer (2024) | | Crossformer (2023) | | Fedformer (2022b) | | Autoformer (2021) | | TiDE (2023a) | |
|---|---|---|---|---|---|---|---|---|---|---|---|---|---|---|---|---|---|---|---|---|---|---|
| Metric | MSE | MAE | MSE | MAE | MSE | MAE | MSE | MAE | MSE | MAE | MSE | MAE | MSE | MAE | MSE | MAE | MSE | MAE | MSE | MAE | MSE | MAE |
| $ETTh1 \rightarrow ETTh2$ | **0.367** | **0.391** | 0.427 | 0.424 | 0.992 | 0.708 | 0.493 | 0.488 | 0.380 | 0.405 | 0.421 | 0.431 | 0.481 | 0.474 | 0.555 | 0.574 | 0.712 | 0.693 | 0.634 | 0.651 | 0.593 | 0.582 |
| $ETTh1 \rightarrow ETTm2$ | **0.301** | **0.357** | 0.361 | 0.397 | 1.867 | 0.869 | 0.415 | 0.452 | 0.314 | 0.360 | 0.327 | 0.361 | 0.311 | 0.361 | 0.613 | 0.629 | 0.681 | 0.588 | 0.647 | 0.609 | 0.563 | 0.547 |
| $ETTh2 \rightarrow ETTh1$ | **0.511** | **0.498** | 0.679 | 0.577 | 1.961 | 0.981 | 0.703 | 0.574 | 0.565 | 0.513 | 0.865 | 0.621 | 0.552 | 0.511 | 0.587 | 0.518 | 0.612 | 0.624 | 0.599 | 0.571 | 0.588 | 0.556 |
| $ETTm1 \rightarrow ETTh2$ | **0.417** | **0.422** | 0.452 | 0.441 | 0.992 | 0.708 | 0.464 | 0.475 | 0.439 | 0.438 | 0.457 | 0.454 | 0.434 | 0.438 | 0.624 | 0.541 | 0.533 | 0.594 | 0.579 | 0.568 | 0.543 | 0.535 |
| $ETTm1 \rightarrow ETTm2$ | **0.291** | **0.331** | 0.329 | 0.357 | 1.867 | 0.869 | 0.335 | 0.389 | 0.296 | 0.334 | 0.322 | 0.354 | 0.324 | **0.331** | 0.595 | 0.572 | 0.612 | 0.611 | 0.603 | 0.592 | 0.534 | 0.527 |
| $ETTm2 \rightarrow ETTm1$ | **0.427** | **0.448** | 0.554 | 0.478 | 1.933 | 0.984 | 0.649 | 0.537 | 0.568 | 0.492 | 0.769 | 0.567 | 0.559 | 0.491 | 0.611 | 0.593 | 0.577 | 0.601 | 0.594 | 0.597 | 0.585 | 0.571 |

**Results.** As demonstrated in Table 6, TIMEMIXER++ consistently outperforms other models in our zero-shot learning evaluation across all datasets. Notably, TIMEMIXER++ achieves a significant reduction in MSE by **13.1%** and in MAE by **5.9%** compared to iTransformer. Moreover, it exhibits a reduction in MSE of **9.6%** and in MAE of **3.8%** when compared with PatchTST. These improvements demonstrate the superior generalization capability and robustness of TIMEMIXER++ in handling unseen data patterns, highlighting its potential for real-world applications where adaptability to new scenarios is crucial.

#### 4.1.7 CLASSIFICATION AND ANOMALY DETECTION

**Setups.** Classification and anomaly detection test models' ability to capture coarse and fine-grained patterns in time series. We use 10 multivariate datasets from UEA Time Series Classification Archive (2018) for classification. For anomaly detection, we evaluate on SMD (2019), SWaT (2016), PSM (2021), MSL and SMAP (2018).

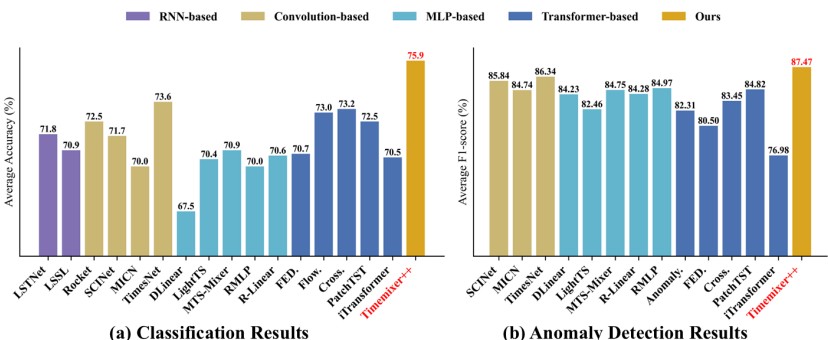

Figure 3: Results of classification and anomaly detection. The results are averaged from several datasets. Higher accuracy and F1 score indicate better performance. *. indicates the Transformer-based models. See Table 19 and 20 in the Appendix H for full results.

**Results.** Results for both tasks are shown in Figure 4.1.7. For classification, TIMEMIXER++ achieves **75.9%** accuracy, surpassing TimesNet by **2.3%** and outperforming other models. Forecasting models like iTransformer and PatchTST perform poorly, highlighting TIMEMIXER++'s versatility. In anomaly detection, TIMEMIXER++ achieves an F1-score of **87.47%**, exceeding TimesNet by **2.59%**, SCINet by **3.09%**, and Anomaly Transformer by **6.62%**. These results emphasize TIMEMIXER++'s strong pattern learning capability, attributed to its multi-scale and multi-resolution architecture.

## 4.2 MODEL ANALYSIS

**Ablation Study.** To verify the effectiveness of each component of TIMEMIXER++, we conducted an ablation study by removing individual components (w/o). The results are in Table 7. TIMEMIXER++ with all components—*channel mixing*, *time image decompose*, *multi-scale mixing*, and *multi-resolution mixing*—achieves the best performance. On datasets like ECL, Traffic, and Solar, channel-mixing improves performance by **5.36%**. Time image decomposition yields an **8.81%** improvement, especially on seasonal datasets like ECL and Traffic. Multi-scale mixing provides a **6.25%** improvement, particularly for less predictable datasets like ETT. Multi-resolution mixing adds a **5.10%** improvement, highlighting the importance of a multi-resolution hybrid ensemble. We provide more ablation studies in Appednix C.

Table 7: MSE for long-term forecasting across 8 benchmarks, evaluated with different model components. We provide more analysis on other tasks in Tables 11 and 12.

|  | ETTh1 | ETTh2 | ETTm1 | ETTm2 | ECL | Traffic | Weather | Solar | Average | Promotion |
|---|---|---|---|---|---|---|---|---|---|---|
| TIMEMIXER++ | **0.419** | **0.339** | **0.369** | **0.269** | **0.165** | **0.416** | **0.226** | **0.203** | **0.300** | - |
| w/o channel mixing | 0.424 | 0.346 | 0.374 | 0.271 | 0.197 | 0.442 | 0.233 | 0.245 | 0.317 | 5.36% |
| w/o time image decomposition | 0.441 | 0.358 | 0.409 | 0.291 | 0.198 | 0.445 | 0.251 | 0.241 | 0.329 | 8.81% |
| w/o multi-scale mixing | 0.447 | 0.361 | 0.391 | 0.284 | 0.172 | 0.427 | 0.239 | 0.234 | 0.320 | 6.25% |
| w/o multi-resolution mixing | 0.431 | 0.350 | 0.374 | 0.280 | 0.181 | 0.432 | 0.241 | 0.233 | 0.316 | 5.10% |

**Representation Analysis.** Our analysis, depicted in Figure 4, present the original, seasonality, and trend images across two scales and three resolutions (periods: $12, 8, 6$; frequencies: $16, 24, 32$). TIMEMIXER++ demonstrates efficacy in the separation of distinct seasonality and trends, precisely capturing multi-periodicities and time-varying trends. Notably, the periodic characteristics vary across different scales and resolutions. This hierarchical structure permits the simultaneous capture of these features, underscoring the robust representational capabilities of TIMEMIXER++ as a pattern machine.

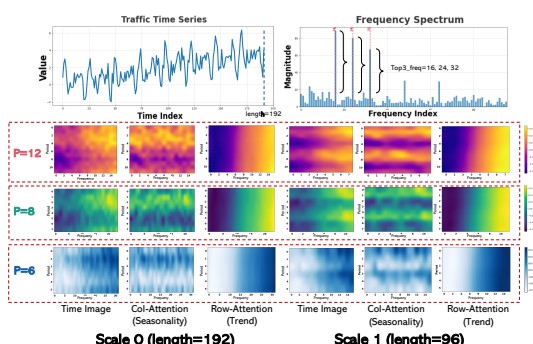

Figure 4: Visualization of representation on Time Image under Traffic dataset. More showcases in Figure 10, 11, 13.

Furthermore, as on the right side of Figure 1, from the perspective of representation learning, TIMEMIXER++ shows superior performance in prediction and anomaly detection with higher CKA similarity 2019, compared to imputation and classification tasks. Lower CKA similarity indicates more distinctive layer-wise representations, suggesting a hierarchical structure. Figure 1 demonstrates that TIMEMIXER++ captures distinct low-level representations for forecasting and anomaly detection, and hierarchical ones for imputation and classification. This highlights TIMEMIXER++'s potential as a general time series pattern machine, capable of identifying diverse patterns across tasks and domains, essential for universal predictive analysis in time series. See Appendix D for more details.

## 5 CONCLUSION

In this paper, we present TIMEMIXER++, a novel framework designed as a universal time series pattern machine for predictive analysis. By leveraging multi-resolution imaging, we construct time images at various resolutions, enabling enhanced representation of temporal dynamics. The use of dual-axis attention allows for effective decomposition of these time images, disentangling seasonal and trend components within deep representations. With multi-scale and multi-resolution mixing techniques, TIMEMIXER++ seamlessly fuses and extracts information across different hierarchical levels, demonstrating strong representational capabilities. Through extensive experiments and comprehensive evaluations, TIMEMIXER++ consistently outperforms existing general-purpose and task-specific time series models , establishing itself as a state-of-the-art solution with significant potential for broad applications in time series analysis. Limitations and directions for future research are discussed in Appendix K.

ACKNOWLEDGEMENT

M. Jin was supported in part by the NVIDIA Academic Grant Program and CSIRO – National Science Foundation (US) AI Research Collaboration Program.

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

## A  IMPLEMENTATION DETAILS

**Datasets Details.**    We evaluate the performance of different models for long-term forecasting on 8 well-established datasets, including Weather, Traffic, Electricity, Exchange, Solar-Energy, and ETT datasets (ETTh1, ETTh2, ETTm1, ETTm2). Furthermore, we adopt PeMS and M4 datasets for short-term forecasting. We detail the descriptions of the dataset in Table 8.To comprehensively evaluate the model's performance in time series analysis tasks, we further introduced datasets for classification and anomaly detection. The classification task is designed to test the model's ability to capture coarse-grained patterns in time series data, while anomaly detection focuses on the recognition of fine-grained patterns. Specifically, we used 10 multivariate datasets from the UEA Time Series Classification Archive (2018) for the evaluation of classification tasks. For anomaly detection, we selected datasets such as SMD (2019), SWaT (2016), PSM (2021), MSL, and SMAP (2018). We detail the descriptions of the datasets for classification and anomaly detection in Table 9 and Table 10.

Table 8: Dataset detailed descriptions. The dataset size is organized in (Train, Validation, Test).

| Tasks | Dataset | Dim | Series Length | Dataset Size | Frequency | Forecastability* | Information |
|---|---|---|---|---|---|---|---|
| | ETTm1 | 7 | {96, 192, 336, 720} | (34465, 11521, 11521) | 15min | 0.46 | Temperature |
| | ETTm2 | 7 | {96, 192, 336, 720} | (34465, 11521, 11521) | 15min | 0.55 | Temperature |
| | ETTh1 | 7 | {96, 192, 336, 720} | (8545, 2881, 2881) | 15 min | 0.38 | Temperature |
| | ETTh2 | 7 | {96, 192, 336, 720} | (8545, 2881, 2881) | 15 min | 0.45 | Temperature |
| Long-term | Electricity | 321 | {96, 192, 336, 720} | (18317, 2633, 5261) | Hourly | 0.77 | Electricity |
| Forecasting | Traffic | 862 | {96, 192, 336, 720} | (12185, 1757, 3509) | Hourly | 0.68 | Transportation |
| | Exchange | 8 | {96, 192, 336, 720} | (5120, 665, 1422) | Daily | 0.41 | Weather |
| | Weather | 21 | {96, 192, 336, 720} | (36792, 5271, 10540) | 10 min | 0.75 | Weather |
| | Solar-Energy | 137 | {96, 192, 336, 720} | (36601, 5161, 10417) | 10min | 0.33 | Electricity |
| | PEMS03 | 358 | 12 | (15617,5135,5135) | 5min | 0.65 | Transportation |
| | PEMS04 | 307 | 12 | (10172,3375,3375) | 5min | 0.45 | Transportation |
| | PEMS07 | 883 | 12 | (16911,5622,5622) | 5min | 0.58 | Transportation |
| Short-term | PEMS08 | 170 | 12 | (10690,3548,265) | 5min | 0.52 | Transportation |
| Forecasting | M4-Yearly | 1 | 6 | (23000, 0, 23000) | Yearly | 0.43 | Demographic |
| | M4-Quarterly | 1 | 8 | (24000, 0, 24000) | Quarterly | 0.47 | Finance |
| | M4-Monthly | 1 | 18 | (48000, 0, 48000) | Monthly | 0.44 | Industry |
| | M4-Weakly | 1 | 13 | (359, 0, 359) | Weakly | 0.43 | Macro |
| | M4-Daily | 1 | 14 | (4227, 0, 4227) | Daily | 0.44 | Micro |
| | M4-Hourly | 1 | 48 | (414, 0, 414) | Hourly | 0.46 | Other |

* The forecastability is calculated by one minus the entropy of Fourier decomposition of time series (Goerg, 2013). A larger value indicates better predictability.

Table 9: Datasets and mapping details of UEA dataset (Bagnall et al., 2018).

| Dataset | Sample Numbers(train set,test set) | Variable Number | Series Length |
|---|---|---|---|
| EthanolConcentration | (261, 263) | 3 | 1751 |
| FaceDetection | (5890, 3524) | 144 | 62 |
| Handwriting | (150, 850) | 3 | 152 |
| Heartbeat | (204, 205) | 61 | 405 |
| JapaneseVowels | (270, 370) | 12 | 29 |
| PEMSSF | (267, 173) | 963 | 144 |
| SelfRegulationSCP1 | (268, 293) | 6 | 896 |
| SelfRegulationSCP2 | (200, 180) | 7 | 1152 |
| SpokenArabicDigits | (6599, 2199) | 13 | 93 |
| UWaveGestureLibrary | (120, 320) | 3 | 315 |

Table 10: Datasets and mapping details of anomaly detection dataset.

| Dataset | Dataset sizes(train set,val set, test set) | Variable Number | Sliding Window Length |
|---------|--------------------------------------------|-----------------|-----------------------|
| SMD | (566724, 141681, 708420) | 38 | 100 |
| MSL | (44653, 11664, 73729) | 55 | 100 |
| SMAP | (108146, 27037, 427617) | 25 | 100 |
| SWaT | (396000, 99000, 449919) | 51 | 100 |
| PSM | (105984, 26497, 87841) | 25 | 100 |

**Baseline Details.** To assess the effectiveness of our method across various tasks, we select 27 advanced baseline models spanning a wide range of architectures. Specifically, we utilize CNN-based models: MICN (2023a), SCINet (2022a), and TimesNet (2023); MLP-based models: TimeMixer (2024b), LightTS (2022a), and DLinear (2023); RMLP&RLinear (2023a) and Transformer-based models: iTransformer (2024), PatchTST (2023), Crossformer (2023), FED-former (2022b), Stationary (2022c), Autoformer (2021), and Informer (2021b). These models have demonstrated superior capabilities in temporal modeling and provide a robust framework for comparative analysis. For specific tasks, TiDE (2023b), FiLM (2022a), N-HiTS (2023), and N-BEATS (2019) address long- or short-term forecasting; Anomaly Transformer (2022) and MTS-Mixers (2023b) target anomaly detection; while Rocket (2023a), LSTNet (2018c), LSSL (2022a), and Flowformer (2022) are utilized for classification. Few/zero-shot forecasting tasks employ ETS-former (2022), Reformer (2020), and LLMTime (2023).

**Metric Details.** Regarding metrics, we utilize the mean square error (MSE) and mean absolute error (MAE) for long-term forecasting. In the case of short-term forecasting, we follow the metrics of SCINet (Liu et al., 2022a) on the PeMS datasets, including mean absolute error (MAE), mean absolute percentage error (MAPE), root mean squared error (RMSE). As for the M4 datasets, we follow the methodology of N-BEATS (Oreshkin et al., 2019) and implement the symmetric mean absolute percentage error (SMAPE), mean absolute scaled error (MASE), and overall weighted average (OWA) as metrics. It is worth noting that OWA is a specific metric utilized in the M4 competition. The calculations of these metrics are:

$$\text{RMSE} = \Big(\sum_{i=1}^{F}(\mathbf{X}_i - \widehat{\mathbf{X}}_i)^2\Big)^{\frac{1}{2}}, \qquad\qquad \text{MAE} = \sum_{i=1}^{F}|\mathbf{X}_i - \widehat{\mathbf{X}}_i|,$$

$$\text{SMAPE} = \frac{200}{F}\sum_{i=1}^{F}\frac{|\mathbf{X}_i - \widehat{\mathbf{X}}_i|}{|\mathbf{X}_i| + |\widehat{\mathbf{X}}_i|}, \qquad \text{MAPE} = \frac{100}{F}\sum_{i=1}^{F}\frac{|\mathbf{X}_i - \widehat{\mathbf{X}}_i|}{|\mathbf{X}_i|},$$

$$\text{MASE} = \frac{1}{F}\sum_{i=1}^{F}\frac{|\mathbf{X}_i - \widehat{\mathbf{X}}_i|}{\frac{1}{F-s}\sum_{j=s+1}^{F}|\mathbf{X}_j - \mathbf{X}_{j-s}|}, \quad \text{OWA} = \frac{1}{2}\left[\frac{\text{SMAPE}}{\text{SMAPE}_{\text{Naïve2}}} + \frac{\text{MASE}}{\text{MASE}_{\text{Naïve2}}}\right],$$

where $s$ is the periodicity of the data. $\mathbf{X}, \widehat{\mathbf{X}} \in \mathbb{R}^{F \times C}$ are the ground truth and prediction results of the future with $F$ time pints and $C$ dimensions. $\mathbf{X}_i$ means the $i$-th future time point.

**Experiment Details.** All experiments were run three times, implemented in Pytorch (Paszke et al., 2019), and conducted on multi NVIDIA A100 80GB GPUs. We set the initial learning rate as a range from $10^{-3}$ to $10^{-1}$ and used the ADAM optimizer (Kingma & Ba, 2015) with L2 loss for model optimization. And the batch size was set to be 512. We set the number of resolutions $K$ to range from 1 to 5. Moreover, we set the number of *MixerBlocks* $L$ to range from 1 to 3. We choose the number of scales $M$ according to the time series length to balance performance and efficiency. To handle longer series in long-term forecasting, we usually set $M$ to 3. As for short-term forecasting with limited series length, we usually set $M$ to 1. We pretrained the model with learning rate decay after linear warm-up. For baselines under the same experimental settings as our main study, we directly report the results from TimesNet (Wu et al., 2023). In scenarios where experimental settings differed or tasks were not previously implemented, we reproduced the baseline results referring to the benchmark framework from the Time-Series Library [2]. The source code and pretrained model will be provided in GitHub (https://github.com/kwuking/TimeMixer).

---

[2] https://github.com/thuml/Time-Series-Library

## B   DETAILS OF MODEL DESIGN

In this section, we present a comprehensive exposition of our model design, encompassing five key components: *channel mixing and embedding* (input projection), *multi-resolution time imaging*, *time image decomposition*, *multi-scale mixing*, and *multi-resolution mixing*. To enhance comprehension, we provide visual illustrations that afford an intuitive understanding of our structural design.

**Channel Mixing and Embedding.**   We employ a channel mixing approach to effectively capture inter-variable dependencies crucial for uncovering rich temporal patterns. Our method first applies variate-wise self-attention, as formulated in Equation 2, at the coarsest temporal scale $\mathbf{x}_M \in \mathbb{R}^{\lfloor \frac{T}{2^M} \rfloor \times C}$, ensuring the preservation of global context. This mechanism fuses information across variables, enabling the extraction of inter-variable patterns. Subsequently, the multivariate time series is projected into an embedding space via the function $\mathrm{Embed}(\cdot) : \mathbb{R}^{\lfloor \frac{T}{2^M} \rfloor \times C} \rightarrow \mathbb{R}^{\lfloor \frac{T}{2^M} \rfloor \times d_{\mathrm{model}}}$, capturing temporal structure at different scales and facilitating comprehensive pattern learning across the input time series.

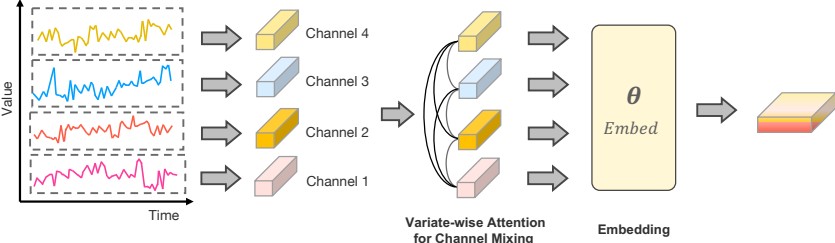

Figure 5: Illustration of the channel mixing approach and embedding function in the input projection process. This process highlights how variate-wise self-attention captures inter-variable dependencies at the coarsest scale, followed by the projection into an embedding space.

**Multi-resolution Time Imaging.**   Starting from the coarsest scale $\mathbf{x}_M^l$, the fast fourier transform (FFT) is applied to extract the top-$K$ frequencies, corresponding to dominant periods in the series. These top-$K$ periods, which capture global patterns, are applied across all scales. At each scale $m$, the input time series $\mathbf{x}_m^l$ is reshaped into $K$ time images by padding the series according to the identified periods and reshaping it (Equation. 6). The size of each image, denoted as $\mathbf{z}_m^{(l,k)}$, is $p_k \times f_{m,k}$. As shown in Figure 6, this process produces multi-resolution time images that capture both temporal and frequency domain patterns, enabling the extraction of comprehensive periodic structures.

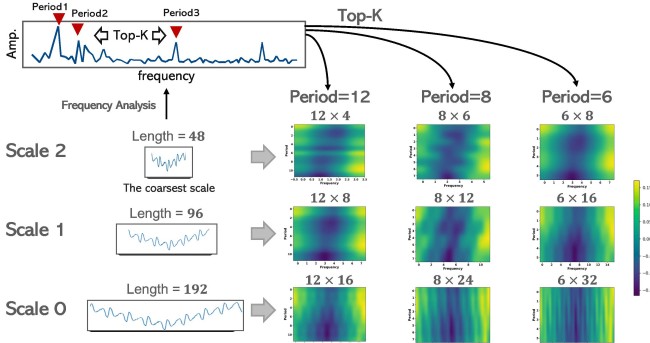

Figure 6: Multi-resolution Time Imaging. We illustrate the generation of multi-resolution images using the top-3 frequencies and three scales.

**Time Image Decomposition.**   As depicted in Figure 7. each time image $\mathbf{z}_m^{(l,k)} \in \mathbb{R}^{p_k \times f_{\mathrm{m,k}} \times d_{\mathrm{model}}}$ corresponds to a specific scale and period. The columns represent time segments within each period, while the rows track consistent points across periods. This structure allows us to apply column-axis

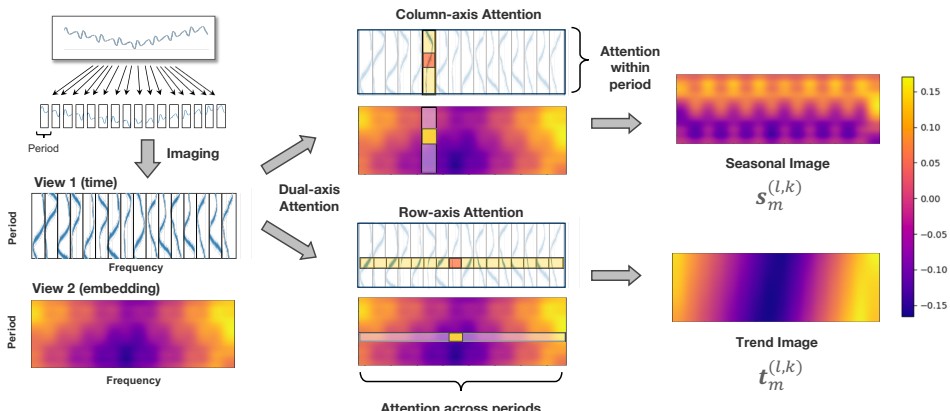

Figure 7: Time Image Decomposition. We demonstrate how the identified periods are used to convert the time series into the time image, and how dual-axis attention is applied to extract both seasonal and trend patterns from this image.

attention, capturing seasonality within a period, and row-axis attention, capturing trend across periods. Column-axis attention processes temporal dependencies within periods using 2D convolution to compute the queries, keys, and values ($\mathbf{Q}_{\text{col}}, \mathbf{K}_{\text{col}}, \mathbf{V}_{\text{col}} \in \mathbb{R}^{f_{\text{m,k}} \times d_{\text{model}}}$), with the row axis transposed into the batch dimension. Similarly, row-axis attention employs 2D convolution to compute queries, keys, and values ($\mathbf{Q}_{\text{row}}, \mathbf{K}_{\text{row}}, \mathbf{V}_{\text{row}} \in \mathbb{R}^{p_k \times d_{\text{model}}}$), where the column axis is transposed into the batch dimension. By leveraging this dual-axis attention, we disentangle seasonality and trends for each image. The seasonal image $\mathbf{s}_m^{(l,k)}$ and the trend image $\mathbf{t}_m^{(l,k)}$ effectively preserve key patterns, facilitating the extraction of long-range dependencies and enabling clearer temporal analysis.

**Multi-scale Mixing.** The Figure 8 illustrates the process of multi-scale mixing as formalized in Equations 8 and 9. In this approach, we hierarchically mix the multi-scale seasonal and trend patterns. For seasonal patterns, the mixing begins at the finest scale $\mathbf{s}_0^{(l,k)}$ and proceeds in a bottom-up fashion through successive scales. Conversely, for trend patterns, the mixing starts from the coarsest scale $\mathbf{t}_M^{(l,k)}$ and flows top-down to finer scales. This hierarchical flow enables effective integration of both long-term and short-term patterns, allowing finer patterns to be aggregated into seasonal representations, while coarser trend information is propagated downward to refine trend representations at finer scales. The effectiveness of this multi-scale mixing is further demonstrated by the representation analysis provided in Appendix D.

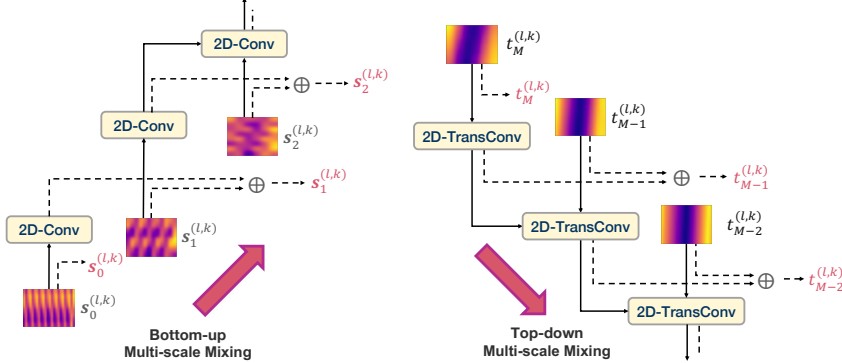

Figure 8: Multi-Scale Mixing. We illustrate the hierarchical mixing of multi-scale seasonal and trend images. Each scale's output (red symbol) integrates all preceding information. 2D convolutions are used in the bottom-up path, while transposed convolutions are applied in the top-down path to accommodate changes in size.

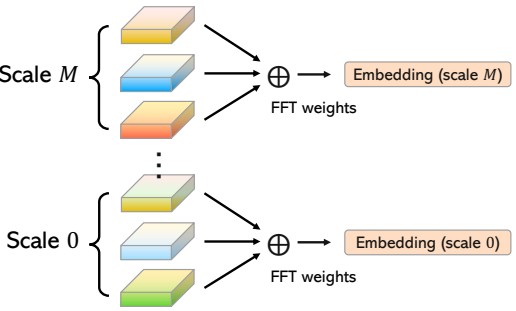

Figure 9: Multi-resolution Mixing. At each scale, $K$ period-based representations are fused after season-trend mixing to produce $M$ scale-specific embeddings, which can be further ensembled for the final output.

**Multi-resolution Mixing.**    As shown in Figure 9, at each scale $m$, the $K$ period-based representations, denoted as $\mathbf{z}_m^{(l,k)}$, are first weighted by their corresponding FFT amplitudes $\mathbf{A}_{f_k}$, which capture the importance of each period, and then summed to produce the final representation for that scale. This process, repeated across all scales, yields a comprehensive embedding for each scale, capturing multi-resolution information.

## C    ADDITIONAL ABLATION STUDIES

To verify the effectiveness of each component of TIMEMIXER++, we conducted a detailed ablation study by performing experiments that remove individual components (w/o) across various tasks, including univariate short-term forecasting, multivariate short-term forecasting, and anomaly detection. Based on the Table 11 12 provided, the conclusions are as follows: TIMEMIXER++ outperforms other configurations in short-term forecasting with the lowest average SMAPE and MAPE scores, indicating the importance of each component. As for PEMS datasets with large variable dimensions, the most significant improvement is observed with channel mixing, showing a 14.95% improvement. In anomaly detection, TIMEMIXER++ achieves the highest average F1 score, with time image decomposition contributing the most to performance, showing a 9.8% improvement. Other components like channel mixing, multi-scale mixing, and multi-resolution mixing also enhance performance. Overall, each component plays a crucial role in the effectiveness of TIMEMIXER++ for all tasks.

Table 11: SMAPE&MAPE for short-term forecasting across 5 benchmarks, evaluated with different model components.

|  | M4 | PEMS03 | PEMS04 | PEMS07 | PEMS08 | Average | Promotion |
|---|---|---|---|---|---|---|---|
| TIMEMIXER++ | **11.45** | **13.43** | **11.34** | **7.32** | **8.21** | **10.35** | - |
| w/o channel mixing | 11.44 | 15.57 | 13.31 | 9.74 | 10.78 | 12.17 | 14.95% |
| w/o time image decompostion | 12.37 | 15.59 | 12.97 | 9.65 | 9.97 | 12.11 | 14.51% |
| w/o multi-scale mixing | 11.98 | 14.97 | 13.02 | 9.17 | 9.69 | 11.79 | 12.06% |
| w/o multi-resolution mixing | 11.87 | 15.02 | 13.14 | 8.72 | 9.53 | 11.68 | 11.23% |

Table 12: F1 score for anomaly detection across 5 benchmarks, evaluated with different model components.

|  | SMD | MSL | SMAP | SWAT | PSM | Average | Promotion |
|---|---|---|---|---|---|---|---|
| TIMEMIXER++ | **86.50** | **85.82** | **73.10** | **94.64** | **97.60** | **87.47** | - |
| w/o channel mixing | 84.51 | 74.03 | 70.91 | 90.41 | 96.17 | 83.21 | 4.94% |
| w/o time image decompostion | 81.21 | 72.43 | 66.02 | 82.41 | 92.53 | 78.92 | 9.84% |
| w/o multi-scale mixing | 82.37 | 75.12 | 92.79 | 86.48 | 94.53 | 86.26 | 1.46% |
| w/o multi-resolution mixing | 83.37 | 79.24 | 77.49 | 88.46 | 96.02 | 86.26 | 2.99% |

## D    ADDITIONAL REPRESENTATION ANALYSIS

To evaluate the representational capabilities of TIMEMIXER++, we selected three datasets: Traffic, Electricity, and ETTm1, each exhibiting distinct periodic and trend characteristics. We conducted comprehensive analyses across various scales and resolutions. Initially, 1D convolution was employed to downsample the original time series, yielding different scales. Subsequently, frequency spectrum analysis was utilized to identify the primary frequency components within the time series, selecting the top three components by magnitude as the primary resolutions. This process transformed the original time series into a multi-scale time image. Time image decomposition was then applied to disentangle seasonality and trends from the original time image, resulting in distinct seasonality and trend images. Hierarchical mixing was performed to facilitate interactions across different scales. The visualization of the resulting representations is depicted in Figures 10, 11, and 13.

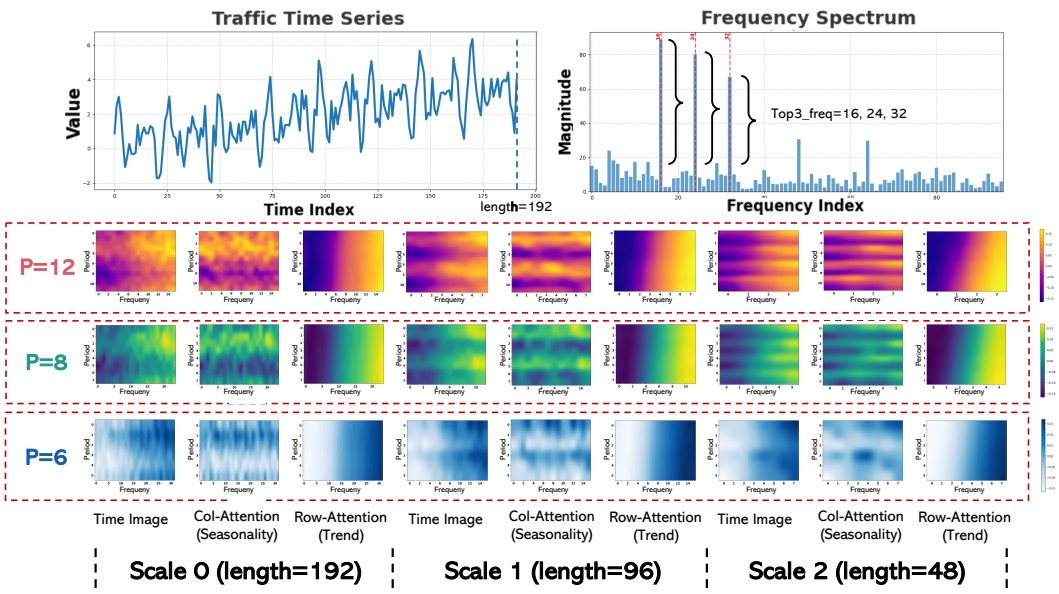

Figure 10: Visualization of representation on Time Image under Traffic dataset. We provide different scales and different resolutions in Time Image, seasonality, and trend.

As shown in the Figure 10, we visualized the representation of the time series in the Traffic dataset across three different scales (length: $192, 96, 48$) and three resolutions (periods: $12, 8, 6$; frequencies: $16, 24, 32$). From the visualization, we can observe that we successfully disentangled the seasonality and trend within the traffic time series. The seasonal patterns vary across different scales and resolutions: at a fine scale (scale: $0$), higher frequency seasonality (period:$32$; freq: $6$) is more prominent, while at a coarse scale (scale: $2$), lower frequency seasonality (period:$16$; freq: $12$) is more evident. Additionally, we successfully isolated the upward trend across all scales and resolutions. This demonstrates the strong representational capability of TIMEMIXER++, highlighting its potential as a general time series pattern machine.

We can observe the visualization of the representation of the ETTm1 dataset, which exhibits multi-period characteristics and a downward trend, in Figure 11. We obtained findings consistent with previous observations: higher frequency components are more prominent at a fine scale, while lower frequency components are more easily observed at a coarse scale. This observation aligns perfectly with theoretical intuition (Harti, 1993), as the fine scale retains more detailed information, such as higher frequency seasonality, from a microscopic perspective, whereas the coarse scale provides a macroscopic view where more low-frequency global information is more evident. Moreover, it is important to note that the powerful representational capability of TIMEMIXER++ allows it to accurately capture the downward trend. These conclusions further underscore the necessity of our multi-scale and multi-resolution design.

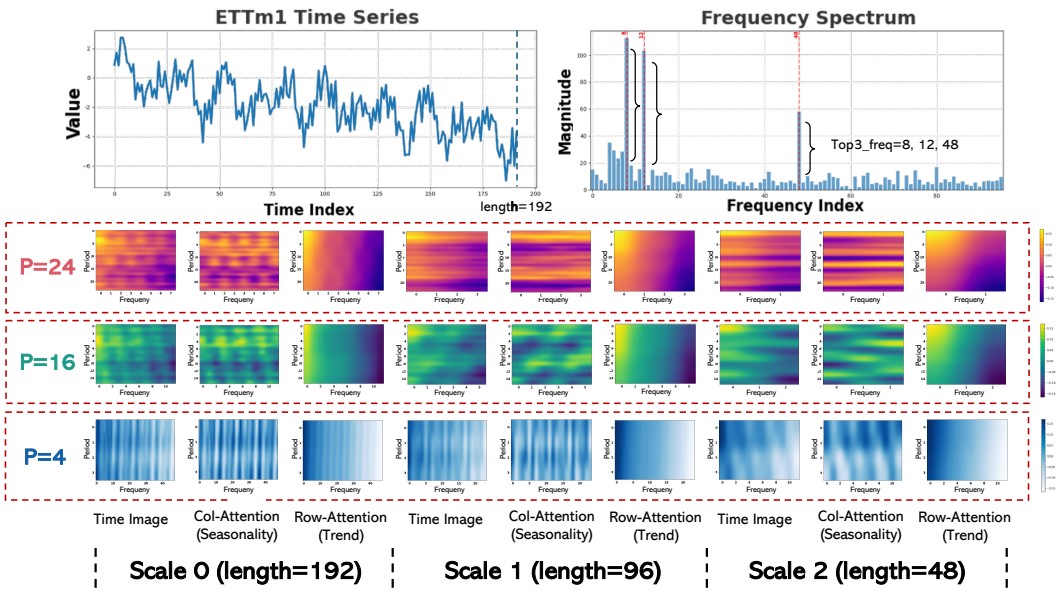

Figure 11: Visualization of representation on Time Image under ETTm1 dataset. We provide different scales and different resolutions in Time Image, seasonality, and trend.

In Figure 13, we present the visualization of the representation of the Electricity dataset. A prominent feature is that the trend is time-varying, initially declining and then rising over time. TIMEMIXER++ successfully captures this time-varying characteristic in its representation. From the figure, we can observe the gradient features in the trend image, which undoubtedly demonstrate the powerful representational capability of the TIMEMIXER++. Especially when combined with our multi-scale and multi-resolution design, the representation exhibits hierarchical performance across different scales and resolutions. This greatly enhances the richness of the representation, making TIMEMIXER++ adept at handling universal predictive analysis.

Through the preceding analyses, it is evident that the paradigm of multi-scale time series modeling (Mozer, 1991) and multi-resolution analysis (Harti, 1993) endows TIMEMIXER++ with robust representational capabilities. This enables proficient handling of high-frequency, low-frequency, seasonal, and trend components within time series data. Such capabilities are pivotal to achieving state-of-the-art performance across diverse and comprehensive time series analysis tasks, underscoring its essential role as a time series pattern machine.

As shown in Figure 12, TIMEMIXER++ consistently excels across all four tasks, achieving state-of-the-art performance in each scenario. The model effectively adapts its representation transformations to meet the demands of different tasks. Whether preserving representations or enabling significant

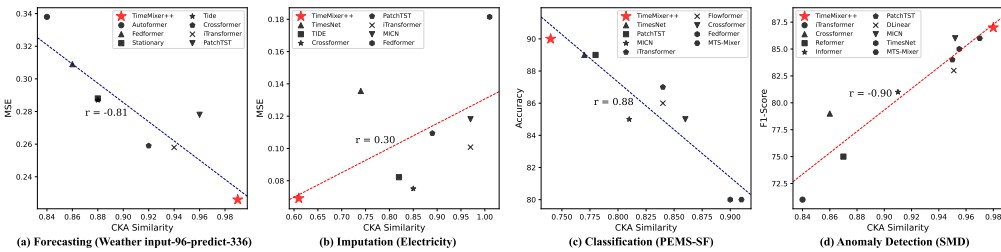

Figure 12: Representation analysis in four tasks. For each model, the centered kernel alignment (CKA) similarity is computed between representations from the first and the last layers.

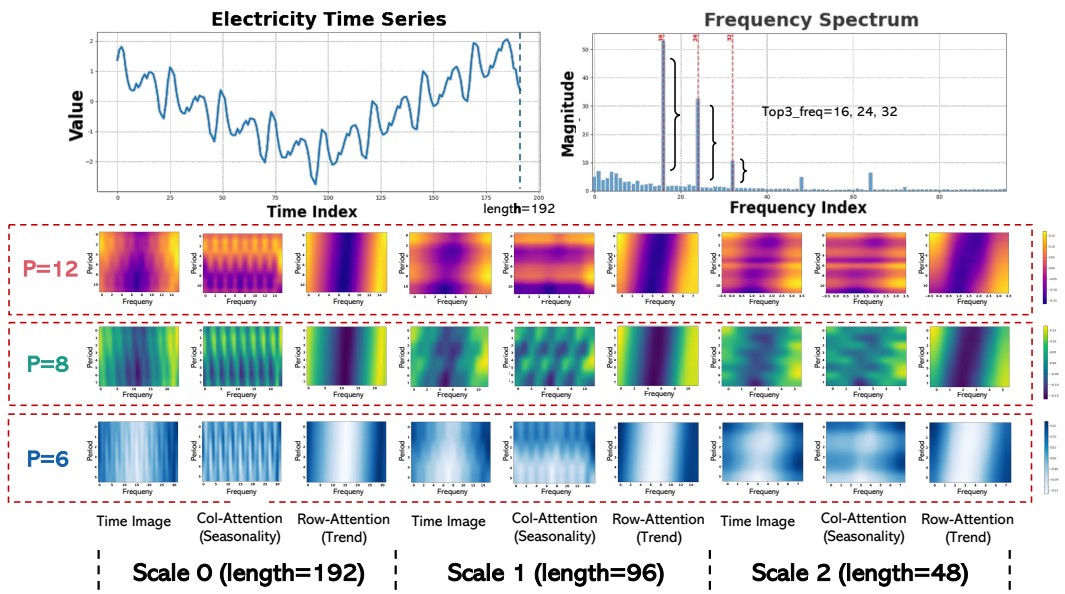

Figure 13: Visualization of representation on Time Image under Electricity dataset. We provide different scales and different resolutions in Time Image, seasonality, and trend.

changes across layers, TimeMixer++ demonstrates its versatility and robustness in handling a wide range of time-series analysis challenges.

## E EFFICIENCY ANALYSIS

We comprehensively compare the forecasting and imputation in performance, training speed, and memory footprint of the following models: TIMEMIXER++, iTransformer(Liu et al., 2024), PatchTST(Nie et al., 2023), TimeMixer(Wang et al., 2024b), TIDE(Das et al., 2023b), Fedformer(Zhou et al., 2022b), TimesNet(Wu et al., 2023), MICN(Wang et al., 2023a), and SCINet(Liu et al., 2022a).

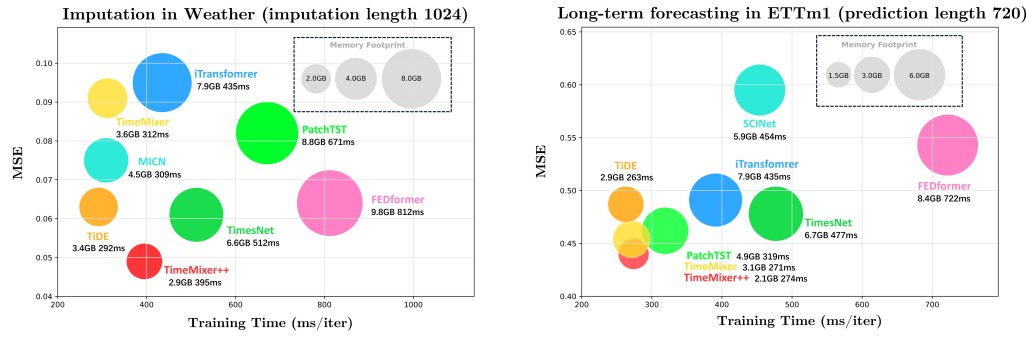

Figure 14: Model efficiency comparison under imputation and long-term forecasting.

As shown in Figure 14, TIMEMIXER++ achieves a comparable balance among memory footprint, training time, and performance in both the Weather imputation and long-term forecasting tasks on ETTm1, delivering the MSE scores.

## F  HYPERPARAMTER SENSITIVITY

We conduct a hyperparameter sensitivity analysis focusing on the four important hyperparameters within TIMEMIXER++: namely, the number of scales $M$, the number of layers $L$, the time series input length $T$, and the selection of the top $K$ periods with the highest amplitudes in the spectrogram. The related findings are presented in Figure 15. Based on our analysis, we have made the following observations: (1) As the number of scales increases, the MSE generally decreases. Increasing the number of scales benefits model performance across all prediction lengths, with noticeable improvements observed between 3 and 4 scales. Considering overall performance and efficiency, the

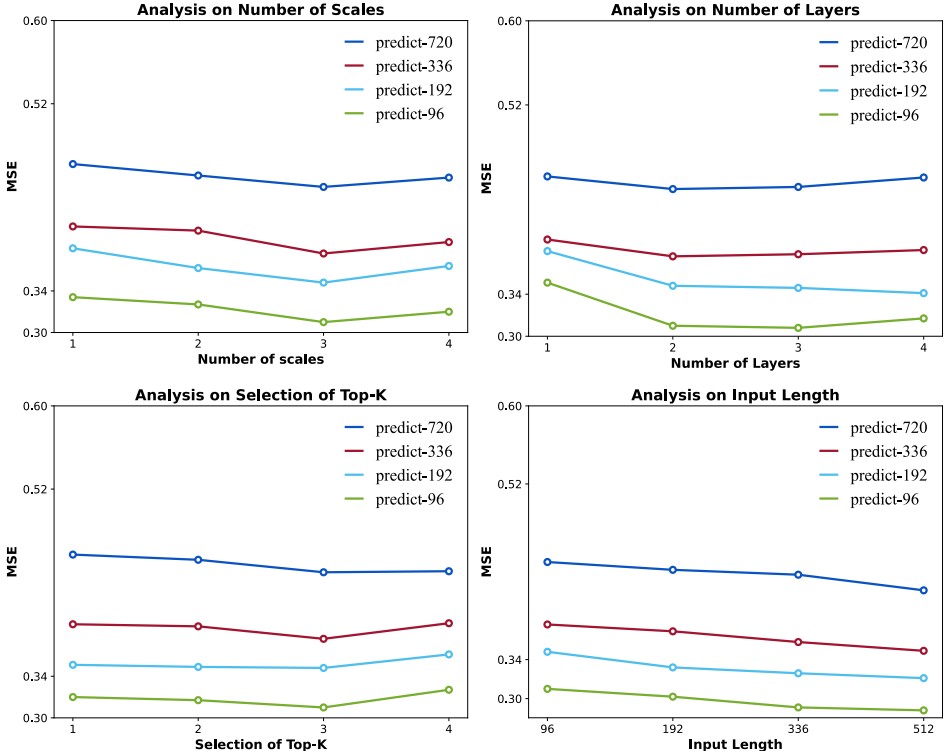

Figure 15: Analysis of hyperparameter sensitivity on ETTm1 dataset.

marginal benefits of increasing $M$ significantly diminish, so we can set $M$ from 1 to 3. (2) Adding more layers typically reduces MSE, particularly between 1 and 2 layers where the change is most pronounced. For shorter prediction lengths (e.g., predict-96), increasing the number of layers results in more significant performance gains. (3) Increasing the selection of Top-K generally leads to a reduction in MSE. For longer prediction lengths, the choice of Top-K has a more substantial impact on the results. (4) As the input length increases, the MSE gradually decreases. Longer input lengths help improve prediction accuracy across all prediction lengths, which indicates that using longer inputs may achieve better predictive performance for the TIMEMIXER++.

## G  ERROR BARS

In this paper, we repeat all the experiments three times. Here we report the standard deviation of our model and the second best model, as well as the statistical significance test in Table 13, 14, 15.

Table 13: Standard deviation and statistical tests for our TIMEMIXER++ and second-best method (iTransformer) on ETT, Weather, Solar-Energy, Electricity and Traffic datasets.

| Model | TimeMixer++ | | iTransformer (2024) | | Confidence |
|---|---|---|---|---|---|
| Dataset | MSE | MAE | MSE | MAE | Level |
| Weather | $0.226 \pm 0.008$ | $0.262 \pm 0.007$ | $0.258 \pm 0.009$ | $0.278 \pm 0.006$ | 99% |
| Solar-Energy | $0.203 \pm 0.027$ | $0.238 \pm 0.026$ | $0.233 \pm 0.009$ | $0.262 \pm 0.011$ | 95% |
| Electricity | $0.165 \pm 0.017$ | $0.253 \pm 0.019$ | $0.178 \pm 0.002$ | $0.270 \pm 0.017$ | 99% |
| Traffic | $0.416 \pm 0.027$ | $0.264 \pm 0.030$ | $0.428 \pm 0.008$ | $0.282 \pm 0.027$ | 95% |
| ETTh1 | $0.419 \pm 0.023$ | $0.432 \pm 0.021$ | $0.454 \pm 0.004$ | $0.447 \pm 0.007$ | 99% |
| ETTh2 | $0.339 \pm 0.020$ | $0.380 \pm 0.019$ | $0.383 \pm 0.004$ | $0.407 \pm 0.007$ | 95% |
| ETTm1 | $0.369 \pm 0.019$ | $0.378 \pm 0.026$ | $0.407 \pm 0.004$ | $0.410 \pm 0.009$ | 99% |
| ETTm2 | $0.269 \pm 0.021$ | $0.320 \pm 0.019$ | $0.288 \pm 0.010$ | $0.332 \pm 0.003$ | 95% |

Table 14: Standard deviation and statistical tests for our TimeMixer++ method and second-best method (TimeMixer) on PEMS dataset.

| Model | TimeMixer++ | | | TimeMixer (2024b) | | | Confidence |
|---|---|---|---|---|---|---|---|
| Dataset | MAE | MAPE | RMSE | MAE | MAPE | RMSE | Level |
| PEMS03 | $13.99 \pm 0.271$ | $13.43 \pm 0.292$ | $24.03 \pm 0.269$ | $14.63 \pm 0.471$ | $14.54 \pm 0.502$ | $23.28 \pm 0.468$ | 99% |
| PEMS04 | $17.46 \pm 0.951$ | $11.34 \pm 0.970$ | $28.83 \pm 0.916$ | $19.21 \pm 0.511$ | $12.53 \pm 0.523$ | $30.92 \pm 0.519$ | 95% |
| PEMS07 | $18.38 \pm 0.991$ | $7.32 \pm 0.977$ | $31.75 \pm 0.890$ | $20.57 \pm 0.372$ | $8.62 \pm 0.399$ | $33.59 \pm 0.375$ | 95% |
| PEMS08 | $13.81 \pm 0.827$ | $8.21 \pm 0.836$ | $23.62 \pm 0.877$ | $15.22 \pm 0.311$ | $9.67 \pm 0.332$ | $24.26 \pm 0.317$ | 99% |

Table 15: Standard deviation and statistical tests for our TimeMixer++ method and second-best method (TimesMixer) on M4 dataset.

| Model | TimeMixer++ | | | TimesMixer (2024b) | | | Confidence |
|---|---|---|---|---|---|---|---|
| Dataset | SMAPE | MASE | OWA | SMAPE | MASE | OWA | Level |
| Yearly | $13.179 \pm 0.021$ | $2.934 \pm 0.012$ | $0.769 \pm 0.001$ | $13.206 \pm 0.121$ | $2.916 \pm 0.022$ | $0.776 \pm 0.002$ | 95% |
| Quarterly | $9.755 \pm 0.001$ | $1.159 \pm 0.005$ | $0.865 \pm 0.009$ | $9.996 \pm 0.101$ | $1.166 \pm 0.015$ | $0.825 \pm 0.008$ | 95% |
| Monthly | $12.432 \pm 0.015$ | $0.904 \pm 0.012$ | $0.841 \pm 0.001$ | $12.605 \pm 0.115$ | $0.919 \pm 0.011$ | $0.869 \pm 0.003$ | 95% |
| Others | $4.698 \pm 0.114$ | $2.931 \pm 0.027$ | $1.01 \pm 0.011$ | $4.564 \pm 0.114$ | $3.115 \pm 0.027$ | $0.982 \pm 0.011$ | 99% |
| Averaged | $11.448 \pm 0.007$ | $1.487 \pm 0.010$ | $0.821 \pm 0.002$ | $11.723 \pm 0.011$ | $1.559 \pm 0.022$ | $0.840 \pm 0.001$ | 99% |

# H  FULL RESULTS

Due to the space limitation of the main text, we place the full results of all experiments in the following: long-term forecasting in Table 16, univariate short-term forecasting in Table 17, multivariate short-term forecasting in Table 18, classification in Table 19 and anomaly detection in Table 20.

# I  SHOWCASES

To assess the performance of various models, we perform a qualitative comparison by visualizing the final dimension of the forecasting results derived from the test set of each dataset (Figures 16, 17, 18, 19, 20, 22, 23, 24, 25, 21). Among the various models, TIMEMIXER++ exhibits superior performance.

# J  BROADER IMPACT

**Real-world applications**  TIMEMIXER++ has achieved state-of-the-art (SOTA) performance as a general time series pattern machine in various time series analysis tasks, including forecasting,

Table 16: Full results for the long-term forecasting task. We compare extensive competitive models under different prediction lengths. *Avg* is averaged from all four prediction lengths, that $\{96, 192, 336, 720\}$.

| Models | | TimeMixer++ (Ours) | | TimeMixer (2024b) | | iTransformer 2024 | | PatchTST 2023 | | Crossformer 2023 | | TiDE 2023a | | TimesNet 2023 | | DLinear 2023 | | SCINet 2022a | | FEDformer 2022b | | Stationary 2022c | | Autoformer 2021 | |
|---|---|---|---|---|---|---|---|---|---|---|---|---|---|---|---|---|---|---|---|---|---|---|---|---|---|
| Metric | | MSE | MAE | MSE | MAE | MSE | MAE | MSE | MAE | MSE | MAE | MSE | MAE | MSE | MAE | MSE | MAE | MSE | MAE | MSE | MAE | MSE | MAE | MSE | MAE |
| Weather | 96 | 0.155 | 0.205 | 0.163 | 0.209 | 0.174 | 0.214 | 0.186 | 0.227 | 0.195 | 0.271 | 0.202 | 0.261 | 0.172 | 0.220 | 0.195 | 0.252 | 0.221 | 0.306 | 0.217 | 0.296 | 0.173 | 0.223 | 0.266 | 0.336 |
| | 192 | 0.201 | 0.245 | 0.208 | 0.250 | 0.221 | 0.254 | 0.234 | 0.265 | 0.209 | 0.277 | 0.242 | 0.298 | 0.219 | 0.261 | 0.237 | 0.295 | 0.261 | 0.340 | 0.276 | 0.336 | 0.245 | 0.285 | 0.307 | 0.367 |
| | 336 | 0.237 | 0.265 | 0.251 | 0.287 | 0.278 | 0.296 | 0.284 | 0.301 | 0.273 | 0.332 | 0.287 | 0.335 | 0.280 | 0.306 | 0.282 | 0.331 | 0.309 | 0.378 | 0.339 | 0.380 | 0.321 | 0.338 | 0.359 | 0.395 |
| | 720 | 0.312 | 0.334 | 0.339 | 0.341 | 0.358 | 0.347 | 0.356 | 0.349 | 0.379 | 0.401 | 0.351 | 0.386 | 0.365 | 0.359 | 0.345 | 0.382 | 0.377 | 0.427 | 0.403 | 0.428 | 0.414 | 0.410 | 0.419 | 0.428 |
| | Avg | 0.226 | 0.262 | 0.240 | 0.271 | 0.258 | 0.278 | 0.265 | 0.285 | 0.264 | 0.320 | 0.271 | 0.320 | 0.259 | 0.287 | 0.265 | 0.315 | 0.292 | 0.363 | 0.309 | 0.360 | 0.288 | 0.314 | 0.338 | 0.382 |
| Solar-Energy | 96 | 0.171 | 0.231 | 0.189 | 0.259 | 0.203 | 0.237 | 0.265 | 0.323 | 0.232 | 0.302 | 0.312 | 0.399 | 0.373 | 0.358 | 0.290 | 0.378 | 0.237 | 0.344 | 0.286 | 0.341 | 0.321 | 0.380 | 0.456 | 0.446 |
| | 192 | 0.218 | 0.263 | 0.222 | 0.283 | 0.233 | 0.261 | 0.288 | 0.332 | 0.371 | 0.410 | 0.339 | 0.416 | 0.397 | 0.376 | 0.320 | 0.398 | 0.280 | 0.380 | 0.291 | 0.337 | 0.346 | 0.369 | 0.588 | 0.561 |
| | 336 | 0.212 | 0.269 | 0.231 | 0.292 | 0.248 | 0.273 | 0.301 | 0.339 | 0.495 | 0.515 | 0.368 | 0.430 | 0.420 | 0.380 | 0.353 | 0.415 | 0.304 | 0.389 | 0.354 | 0.416 | 0.357 | 0.387 | 0.595 | 0.588 |
| | 720 | 0.212 | 0.270 | 0.223 | 0.285 | 0.249 | 0.275 | 0.295 | 0.336 | 0.526 | 0.542 | 0.370 | 0.425 | 0.420 | 0.381 | 0.357 | 0.413 | 0.308 | 0.388 | 0.380 | 0.437 | 0.375 | 0.424 | 0.733 | 0.633 |
| | Avg | 0.203 | 0.238 | 0.216 | 0.280 | 0.233 | 0.262 | 0.287 | 0.333 | 0.406 | 0.442 | 0.347 | 0.417 | 0.403 | 0.374 | 0.330 | 0.401 | 0.282 | 0.375 | 0.328 | 0.383 | 0.350 | 0.390 | 0.586 | 0.557 |
| Electricity | 96 | 0.135 | 0.222 | 0.153 | 0.247 | 0.148 | 0.240 | 0.190 | 0.296 | 0.219 | 0.314 | 0.237 | 0.329 | 0.168 | 0.272 | 0.210 | 0.302 | 0.247 | 0.345 | 0.193 | 0.308 | 0.169 | 0.273 | 0.201 | 0.317 |
| | 192 | 0.147 | 0.235 | 0.166 | 0.256 | 0.162 | 0.253 | 0.199 | 0.304 | 0.231 | 0.322 | 0.236 | 0.330 | 0.184 | 0.322 | 0.210 | 0.305 | 0.257 | 0.355 | 0.201 | 0.315 | 0.182 | 0.286 | 0.222 | 0.334 |
| | 336 | 0.164 | 0.245 | 0.185 | 0.277 | 0.178 | 0.269 | 0.217 | 0.319 | 0.246 | 0.337 | 0.249 | 0.344 | 0.198 | 0.300 | 0.223 | 0.319 | 0.269 | 0.369 | 0.214 | 0.329 | 0.200 | 0.304 | 0.231 | 0.443 |
| | 720 | 0.212 | 0.310 | 0.225 | 0.310 | 0.225 | 0.317 | 0.258 | 0.352 | 0.280 | 0.363 | 0.284 | 0.373 | 0.220 | 0.320 | 0.258 | 0.350 | 0.299 | 0.390 | 0.246 | 0.355 | 0.222 | 0.321 | 0.254 | 0.361 |
| | Avg | 0.165 | 0.253 | 0.182 | 0.272 | 0.178 | 0.270 | 0.216 | 0.318 | 0.244 | 0.334 | 0.251 | 0.344 | 0.192 | 0.304 | 0.225 | 0.319 | 0.268 | 0.365 | 0.214 | 0.327 | 0.193 | 0.296 | 0.227 | 0.338 |
| Traffic | 96 | 0.392 | 0.253 | 0.462 | 0.285 | 0.395 | 0.268 | 0.526 | 0.347 | 0.644 | 0.429 | 0.805 | 0.493 | 0.593 | 0.321 | 0.650 | 0.396 | 0.788 | 0.499 | 0.587 | 0.366 | 0.612 | 0.338 | 0.613 | 0.388 |
| | 192 | 0.402 | 0.258 | 0.473 | 0.296 | 0.417 | 0.276 | 0.522 | 0.332 | 0.665 | 0.431 | 0.756 | 0.474 | 0.617 | 0.336 | 0.598 | 0.370 | 0.789 | 0.505 | 0.604 | 0.373 | 0.613 | 0.340 | 0.616 | 0.382 |
| | 336 | 0.428 | 0.263 | 0.498 | 0.296 | 0.433 | 0.283 | 0.517 | 0.334 | 0.674 | 0.420 | 0.762 | 0.477 | 0.629 | 0.336 | 0.605 | 0.373 | 0.797 | 0.508 | 0.621 | 0.383 | 0.618 | 0.328 | 0.622 | 0.337 |
| | 720 | 0.441 | 0.282 | 0.506 | 0.313 | 0.467 | 0.302 | 0.552 | 0.352 | 0.683 | 0.424 | 0.719 | 0.449 | 0.640 | 0.350 | 0.645 | 0.394 | 0.841 | 0.523 | 0.626 | 0.382 | 0.653 | 0.355 | 0.660 | 0.408 |
| | Avg | 0.416 | 0.264 | 0.484 | 0.297 | 0.428 | 0.282 | 0.529 | 0.341 | 0.667 | 0.426 | 0.760 | 0.473 | 0.620 | 0.336 | 0.625 | 0.383 | 0.804 | 0.509 | 0.610 | 0.376 | 0.624 | 0.340 | 0.628 | 0.379 |
| Exchange | 96 | 0.085 | 0.214 | 0.090 | 0.235 | 0.086 | 0.206 | 0.088 | 0.205 | 0.256 | 0.367 | 0.094 | 0.218 | 0.107 | 0.234 | 0.088 | 0.218 | 0.267 | 0.396 | 0.148 | 0.278 | 0.111 | 0.237 | 0.197 | 0.323 |
| | 192 | 0.175 | 0.313 | 0.187 | 0.343 | 0.177 | 0.299 | 0.176 | 0.299 | 0.470 | 0.509 | 0.184 | 0.307 | 0.226 | 0.344 | 0.176 | 0.315 | 0.351 | 0.459 | 0.271 | 0.315 | 0.219 | 0.335 | 0.300 | 0.369 |
| | 336 | 0.316 | 0.420 | 0.353 | 0.473 | 0.331 | 0.417 | 0.301 | 0.397 | 1.268 | 0.883 | 0.349 | 0.431 | 0.367 | 0.448 | 0.313 | 0.427 | 1.324 | 0.853 | 0.460 | 0.427 | 0.421 | 0.476 | 0.509 | 0.524 |
| | 720 | 0.851 | 0.689 | 0.934 | 0.761 | 0.847 | 0.691 | 0.901 | 0.714 | 1.767 | 1.068 | 0.852 | 0.698 | 0.964 | 0.746 | 0.839 | 0.695 | 1.058 | 0.797 | 1.195 | 0.695 | 1.092 | 0.769 | 1.447 | 0.941 |
| | Avg | 0.357 | 0.391 | 0.391 | 0.453 | 0.360 | 0.403 | 0.367 | 0.404 | 0.940 | 0.707 | 0.370 | 0.413 | 0.416 | 0.443 | 0.354 | 0.414 | 0.750 | 0.626 | 0.519 | 0.429 | 0.461 | 0.454 | 0.613 | 0.539 |
| ETTh1 | 96 | 0.361 | 0.403 | 0.375 | 0.400 | 0.386 | 0.405 | 0.460 | 0.447 | 0.423 | 0.448 | 0.479 | 0.464 | 0.384 | 0.402 | 0.397 | 0.412 | 0.654 | 0.599 | 0.395 | 0.424 | 0.513 | 0.491 | 0.449 | 0.459 |
| | 192 | 0.416 | 0.441 | 0.429 | 0.421 | 0.441 | 0.512 | 0.477 | 0.429 | 0.471 | 0.474 | 0.525 | 0.492 | 0.436 | 0.429 | 0.446 | 0.441 | 0.719 | 0.631 | 0.469 | 0.470 | 0.534 | 0.504 | 0.500 | 0.482 |
| | 336 | 0.430 | 0.434 | 0.484 | 0.458 | 0.487 | 0.458 | 0.546 | 0.496 | 0.570 | 0.546 | 0.565 | 0.515 | 0.491 | 0.469 | 0.489 | 0.467 | 0.778 | 0.659 | 0.530 | 0.499 | 0.588 | 0.535 | 0.521 | 0.496 |
| | 720 | 0.467 | 0.451 | 0.498 | 0.482 | 0.503 | 0.491 | 0.544 | 0.517 | 0.653 | 0.621 | 0.594 | 0.558 | 0.521 | 0.500 | 0.513 | 0.510 | 0.836 | 0.699 | 0.598 | 0.544 | 0.643 | 0.616 | 0.514 | 0.512 |
| | Avg | 0.419 | 0.432 | 0.447 | 0.440 | 0.454 | 0.447 | 0.516 | 0.484 | 0.529 | 0.522 | 0.541 | 0.507 | 0.458 | 0.450 | 0.461 | 0.457 | 0.747 | 0.647 | 0.498 | 0.484 | 0.570 | 0.537 | 0.496 | 0.487 |
| ETTh2 | 96 | 0.276 | 0.328 | 0.289 | 0.341 | 0.297 | 0.349 | 0.308 | 0.355 | 0.745 | 0.584 | 0.400 | 0.440 | 0.340 | 0.374 | 0.340 | 0.394 | 0.707 | 0.621 | 0.358 | 0.397 | 0.476 | 0.458 | 0.346 | 0.388 |
| | 192 | 0.342 | 0.379 | 0.372 | 0.392 | 0.380 | 0.400 | 0.393 | 0.405 | 0.877 | 0.656 | 0.528 | 0.509 | 0.402 | 0.414 | 0.482 | 0.479 | 0.860 | 0.689 | 0.429 | 0.439 | 0.512 | 0.493 | 0.456 | 0.452 |
| | 336 | 0.346 | 0.398 | 0.386 | 0.414 | 0.428 | 0.432 | 0.427 | 0.436 | 1.043 | 0.731 | 0.643 | 0.571 | 0.452 | 0.452 | 0.591 | 0.541 | 1.000 | 0.744 | 0.496 | 0.487 | 0.552 | 0.551 | 0.482 | 0.486 |
| | 720 | 0.392 | 0.415 | 0.412 | 0.434 | 0.427 | 0.445 | 0.436 | 0.450 | 1.104 | 0.763 | 0.874 | 0.679 | 0.462 | 0.468 | 0.839 | 0.661 | 1.249 | 0.838 | 0.463 | 0.474 | 0.562 | 0.560 | 0.515 | 0.511 |
| | Avg | 0.339 | 0.380 | 0.364 | 0.395 | 0.383 | 0.407 | 0.391 | 0.411 | 0.942 | 0.684 | 0.611 | 0.550 | 0.414 | 0.427 | 0.563 | 0.519 | 0.954 | 0.723 | 0.437 | 0.449 | 0.526 | 0.516 | 0.450 | 0.459 |
| ETTm1 | 96 | 0.310 | 0.334 | 0.320 | 0.357 | 0.334 | 0.368 | 0.352 | 0.374 | 0.404 | 0.426 | 0.364 | 0.387 | 0.338 | 0.375 | 0.346 | 0.374 | 0.418 | 0.438 | 0.379 | 0.419 | 0.386 | 0.398 | 0.505 | 0.475 |
| | 192 | 0.348 | 0.362 | 0.361 | 0.381 | 0.390 | 0.393 | 0.374 | 0.387 | 0.450 | 0.451 | 0.398 | 0.404 | 0.374 | 0.387 | 0.382 | 0.391 | 0.439 | 0.450 | 0.426 | 0.441 | 0.459 | 0.444 | 0.553 | 0.496 |
| | 336 | 0.376 | 0.391 | 0.390 | 0.404 | 0.426 | 0.420 | 0.421 | 0.414 | 0.532 | 0.515 | 0.428 | 0.425 | 0.410 | 0.411 | 0.415 | 0.415 | 0.490 | 0.485 | 0.445 | 0.459 | 0.495 | 0.464 | 0.621 | 0.537 |
| | 720 | 0.440 | 0.423 | 0.454 | 0.441 | 0.491 | 0.459 | 0.462 | 0.449 | 0.666 | 0.589 | 0.487 | 0.461 | 0.478 | 0.450 | 0.473 | 0.451 | 0.595 | 0.550 | 0.543 | 0.490 | 0.585 | 0.516 | 0.671 | 0.561 |
| | Avg | 0.369 | 0.378 | 0.381 | 0.395 | 0.407 | 0.410 | 0.406 | 0.407 | 0.513 | 0.495 | 0.419 | 0.419 | 0.400 | 0.406 | 0.404 | 0.408 | 0.485 | 0.481 | 0.448 | 0.452 | 0.481 | 0.456 | 0.588 | 0.517 |
| ETTm2 | 96 | 0.170 | 0.245 | 0.175 | 0.258 | 0.180 | 0.264 | 0.183 | 0.270 | 0.287 | 0.366 | 0.207 | 0.305 | 0.187 | 0.267 | 0.193 | 0.293 | 0.286 | 0.377 | 0.203 | 0.287 | 0.192 | 0.274 | 0.255 | 0.339 |
| | 192 | 0.229 | 0.291 | 0.237 | 0.299 | 0.250 | 0.309 | 0.255 | 0.314 | 0.414 | 0.492 | 0.290 | 0.364 | 0.249 | 0.309 | 0.284 | 0.361 | 0.399 | 0.445 | 0.269 | 0.328 | 0.280 | 0.339 | 0.281 | 0.340 |
| | 336 | 0.303 | 0.343 | 0.298 | 0.340 | 0.311 | 0.348 | 0.309 | 0.347 | 0.597 | 0.542 | 0.377 | 0.422 | 0.321 | 0.351 | 0.382 | 0.429 | 0.637 | 0.591 | 0.325 | 0.366 | 0.334 | 0.361 | 0.339 | 0.372 |
| | 720 | 0.373 | 0.399 | 0.391 | 0.396 | 0.412 | 0.407 | 0.412 | 0.404 | 1.730 | 1.042 | 0.558 | 0.524 | 0.408 | 0.403 | 0.558 | 0.525 | 0.960 | 0.735 | 0.421 | 0.415 | 0.417 | 0.413 | 0.433 | 0.432 |
| | Avg | 0.269 | 0.320 | 0.275 | 0.323 | 0.288 | 0.332 | 0.290 | 0.334 | 0.757 | 0.610 | 0.358 | 0.404 | 0.291 | 0.333 | 0.354 | 0.402 | 0.954 | 0.723 | 0.305 | 0.349 | 0.306 | 0.347 | 0.327 | 0.371 |

Table 17: Short-term forecasting results in the M4 dataset with a single variate. All prediction lengths are in $[6, 48]$. A lower SMAPE, MASE or OWA indicates a better prediction. ∗. in the Transformers indicates the name of ∗former. *Stationary* means the Non-stationary Transformer.

| Models | | TimeMixer++ (Ours) | TimeMixer (2024b) | iTransformer (2024) | TiDE (2023a) | TimesNet (2023) | N-HiTS (2023) | N-BEATS* (2019) | PatchTST (2023) | MICN (2023a) | FiLM (2022a) | LightTS (2022a) | DLinear (2023) | FED. (2022b) | Stationary (2022c) | Auto. (2021) |
|---|---|---|---|---|---|---|---|---|---|---|---|---|---|---|---|---|
| Yearly | SMAPE | **13.179** | 13.206 | 13.923 | 15.320 | 13.387 | 13.418 | 13.436 | 16.463 | 25.022 | 17.431 | 14.247 | 16.965 | 13.728 | 13.717 | 13.974 |
| | MASE | 2.934 | **2.916** | 3.214 | 3.540 | 2.996 | 3.045 | 3.043 | 3.967 | 7.162 | 4.043 | 3.109 | 4.283 | 3.048 | 3.078 | 3.134 |
| | OWA | **0.769** | 0.776 | 0.830 | 0.910 | 0.786 | 0.793 | 0.794 | 1.003 | 1.667 | 1.042 | 0.827 | 1.058 | 0.803 | 0.807 | 0.822 |
| Quarterly | SMAPE | **9.755** | 9.996 | 10.757 | 11.830 | 10.100 | 10.202 | 10.124 | 10.644 | 15.214 | 12.925 | 11.364 | 12.145 | 10.792 | 10.958 | 11.338 |
| | MASE | **1.159** | 1.166 | 1.283 | 1.410 | 1.182 | 1.194 | 1.169 | 1.278 | 1.963 | 1.664 | 1.328 | 1.520 | 1.283 | 1.325 | 1.365 |
| | OWA | 0.865 | **0.825** | 0.956 | 1.050 | 0.890 | 0.899 | 0.886 | 0.949 | 1.407 | 1.193 | 1.000 | 1.106 | 0.958 | 0.981 | 1.012 |
| Monthly | SMAPE | **12.432** | 12.605 | 13.796 | 15.180 | 12.670 | 12.791 | 12.677 | 13.399 | 16.943 | 15.407 | 14.014 | 13.514 | 14.260 | 13.917 | 13.958 |
| | MASE | **0.904** | 0.919 | 1.083 | 1.190 | 0.933 | 0.969 | 0.937 | 1.031 | 1.442 | 1.298 | 1.053 | 1.037 | 1.102 | 1.097 | 1.103 |
| | OWA | **0.841** | 0.869 | 0.987 | 1.090 | 0.878 | 0.880 | 0.880 | 0.949 | 1.265 | 1.144 | 0.981 | 1.037 | 1.012 | 0.998 | 1.002 |
| Others | SMAPE | 4.698 | **4.564** | 5.569 | 6.120 | 4.891 | 5.061 | 4.925 | 6.558 | 41.985 | 7.134 | 15.880 | 6.709 | 4.954 | 6.302 | 5.485 |
| | MASE | **2.931** | 3.115 | 3.940 | 4.330 | 3.302 | 3.216 | 3.391 | 4.511 | 62.734 | 5.09 | 11.434 | 4.953 | 3.264 | 4.064 | 3.865 |
| | OWA | 1.01 | **0.982** | 1.207 | 1.330 | 1.035 | 1.040 | 1.053 | 1.401 | 14.313 | 1.553 | 3.474 | 1.487 | 1.036 | 1.304 | 1.187 |
| Weighted Average | SMAPE | **11.448** | 11.723 | 12.684 | 13.950 | 11.829 | 11.927 | 11.851 | 13.152 | 19.638 | 14.863 | 13.525 | 13.639 | 12.840 | 12.780 | 12.909 |
| | MASE | **1.487** | 1.559 | 1.764 | 1.940 | 1.585 | 1.613 | 1.559 | 1.945 | 5.947 | 2.207 | 2.111 | 2.095 | 1.701 | 1.756 | 1.771 |
| | OWA | **0.821** | 0.840 | 0.929 | 1.020 | 0.851 | 0.861 | 0.855 | 0.998 | 2.279 | 1.125 | 1.051 | 1.051 | 0.918 | 0.930 | 0.939 |

∗ The original paper of N-BEATS (2019) adopts a special ensemble method to promote the performance. For fair comparisons, we remove the ensemble and only compare the pure forecasting models.

Table 18: Short-term forecasting results in the PEMS datasets with multiple variates. All input lengths are 96 and prediction lengths are 12. A lower MAE, MAPE or RMSE indicates a better prediction.

| Models | | TimeMixer++ (Ours) | TimeMixer (2024b) | iTransformer (2024) | TiDE (2023a) | SCINet (2022a) | Crossformer (2023) | PatchTST (2023) | TimesNet (2023) | MICN (2023a) | DLinear (2023) | FEDformer (2022b) | Stationary (2022c) | Autoformer (2021) |
|---|---|---|---|---|---|---|---|---|---|---|---|---|---|---|
| PEMS03 | MAE | **13.99** | 14.63 | 16.72 | 18.39 | 15.97 | 15.64 | 18.95 | 16.41 | 15.71 | 19.70 | 19.00 | 17.64 | 18.08 |
| | MAPE | 13.43 | **11.54** | 15.81 | 17.39 | 15.89 | 15.74 | 17.29 | 15.17 | 15.67 | 18.35 | 18.57 | 17.56 | 18.75 |
| | RMSE | 24.03 | **23.28** | 27.81 | 30.59 | 25.20 | 25.56 | 30.15 | 26.72 | 24.55 | 32.35 | 30.05 | 28.37 | 27.82 |
| PEMS04 | MAE | **17.46** | 19.21 | 21.81 | 23.99 | 20.35 | 20.38 | 24.62 | 21.63 | 24.62 | 26.51 | 25.00 | 22.34 | 25.00 |
| | MAPE | **11.34** | 12.53 | 13.42 | 14.76 | 12.84 | 12.84 | 16.65 | 13.15 | 13.53 | 16.12 | 16.76 | 14.85 | 16.70 |
| | RMSE | **28.83** | 30.92 | 33.91 | 37.30 | 32.31 | 32.41 | 40.46 | 34.90 | 34.39 | 39.51 | 41.81 | 35.47 | 38.02 |
| PEMS07 | MAE | **18.38** | 20.57 | 23.01 | 25.31 | 22.79 | 22.54 | 27.87 | 25.12 | 22.28 | 28.65 | 27.92 | 26.02 | 26.92 |
| | MAPE | **7.32** | 8.62 | 10.02 | 11.02 | 9.41 | 9.38 | 11.26 | 10.60 | 9.57 | 12.15 | 12.29 | 11.75 | 11.83 |
| | RMSE | **31.75** | 33.59 | 35.56 | 39.12 | 35.61 | 35.49 | 42.56 | 40.71 | 35.40 | 45.02 | 42.29 | 42.34 | 40.60 |
| PEMS08 | MAE | **13.81** | 15.22 | 17.94 | 19.73 | 17.38 | 17.56 | 20.35 | 19.01 | 17.76 | 20.26 | 20.56 | 19.29 | 20.47 |
| | MAPE | **8.21** | 9.67 | 10.93 | 12.02 | 10.80 | 10.92 | 13.15 | 11.83 | 10.76 | 12.09 | 12.41 | 12.21 | 12.27 |
| | RMSE | **23.62** | 24.26 | 27.88 | 30.67 | 27.34 | 27.21 | 31.04 | 30.65 | 27.26 | 32.38 | 32.97 | 38.62 | 31.52 |

Table 19: Full results for the anomaly detection task. The P, R and F1 represent the precision, recall and F1-score (%) respectively. F1-score is the harmonic mean of precision and recall. A higher value of P, R and F1 indicates a better performance.

| Datasets | | SMD | | | MSL | | | SMAP | | | SWaT | | | PSM | | | Avg F1 |
|---|---|---|---|---|---|---|---|---|---|---|---|---|---|---|---|---|---|
| Metrics | | P | R | F1 | P | R | F1 | P | R | F1 | P | R | F1 | P | R | F1 | (%) |
| LSTM | (1997) | 78.52 | 65.47 | 71.41 | 78.04 | 86.22 | 81.93 | 91.06 | 57.49 | 70.48 | 78.06 | 91.72 | 84.34 | 69.24 | 99.53 | 81.67 | 77.97 |
| Transformer | (2017) | 83.58 | 76.13 | 79.56 | 71.57 | 87.37 | 78.68 | 89.37 | 57.12 | 69.70 | 68.84 | 96.53 | 80.37 | 62.75 | 96.56 | 76.07 | 76.88 |
| LogTrans | (2019) | 83.46 | 70.13 | 76.21 | 73.05 | 87.37 | 79.57 | 89.15 | 57.59 | 69.97 | 68.67 | 97.32 | 80.52 | 63.06 | 98.00 | 76.74 | 76.60 |
| TCN | (2019) | 84.06 | 79.07 | 81.49 | 75.11 | 82.44 | 78.60 | 86.90 | 59.23 | 70.45 | 76.59 | 95.71 | 85.09 | 54.59 | 99.77 | 70.57 | 77.24 |
| Reformer | (2020) | 82.58 | 69.24 | 75.32 | 85.51 | 83.31 | 84.40 | 90.91 | 57.44 | 70.40 | 72.50 | 96.53 | 82.80 | 59.93 | 95.38 | 73.61 | 77.31 |
| Informer | (2021a) | 86.60 | 77.23 | 81.65 | 81.77 | 86.48 | 84.06 | 90.11 | 57.13 | 69.92 | 70.29 | 96.75 | 81.43 | 64.27 | 96.33 | 77.10 | 78.83 |
| Anomaly* | (2022) | 88.91 | 82.23 | 85.49 | 79.61 | 87.37 | 83.31 | 91.85 | 58.11 | 71.18 | 72.51 | 97.32 | 83.10 | 68.35 | 94.72 | 79.40 | 80.50 |
| Pyraformer | (2022b) | 85.61 | 80.61 | 83.04 | 83.81 | 85.93 | 84.86 | 92.54 | 57.71 | 71.09 | 87.92 | 96.00 | 91.78 | 71.67 | 96.02 | 82.08 | 82.57 |
| Autoformer | (2021) | 88.06 | 82.35 | 85.11 | 77.27 | 80.92 | 79.05 | 90.40 | 58.62 | 71.12 | 89.85 | 95.81 | 92.74 | 99.08 | 88.15 | 93.29 | 84.26 |
| LSSL | (2022b) | 78.51 | 65.32 | 71.31 | 77.55 | 88.18 | 82.53 | 89.43 | 53.43 | 66.90 | 79.05 | 93.72 | 85.76 | 66.02 | 92.93 | 77.20 | 76.74 |
| Stationary | (2022c) | 88.33 | 81.21 | 84.62 | 68.55 | 89.14 | 77.50 | 89.37 | 59.02 | 71.09 | 68.03 | 96.75 | 79.88 | 97.82 | 96.76 | 97.29 | 82.08 |
| DLinear | (2023) | 83.62 | 71.52 | 77.10 | 84.34 | 85.42 | 84.88 | 92.32 | 55.41 | 69.26 | 80.91 | 95.30 | 87.52 | 98.28 | 89.26 | 93.55 | 82.46 |
| ETSformer | (2022) | 87.44 | 79.23 | 83.13 | 85.13 | 84.93 | 85.03 | 92.25 | 55.75 | 69.50 | 90.02 | 80.36 | 84.91 | 99.31 | 85.28 | 91.76 | 82.87 |
| LightTS | (2022a) | 87.10 | 78.42 | 82.53 | 82.40 | 75.78 | 78.95 | 92.58 | 55.27 | 69.21 | 91.98 | 94.72 | 93.33 | 98.37 | 95.97 | 97.15 | 84.23 |
| FEDformer | (2022b) | 87.95 | 82.39 | 85.08 | 77.14 | 80.07 | 78.57 | 90.47 | 58.10 | 70.76 | 90.17 | 96.42 | 93.19 | 97.31 | 97.16 | 97.23 | 84.97 |
| TimesNet | (2023) | 88.66 | 83.14 | 85.81 | 83.92 | 86.42 | 85.15 | 92.52 | 58.29 | 71.52 | 86.76 | 97.32 | 91.74 | 98.19 | 96.76 | 97.47 | 86.34 |
| TiDE | (2023a) | 76.00 | 63.00 | 68.91 | 84.00 | 60.00 | 70.18 | 88.00 | 50.00 | 64.00 | 98.00 | 63.00 | 76.73 | 93.00 | 92.00 | 92.50 | 74.46 |
| iTransformer | (2024) | 78.45 | 65.10 | 71.15 | 86.15 | 62.65 | 72.54 | 90.67 | 52.96 | 66.87 | 99.96 | 65.55 | 79.18 | 95.65 | 94.69 | 95.17 | 76.98 |
| TimesMixer++ | (Ours) | 88.59 | 84.50 | **86.50** | 89.73 | 82.23 | **85.82** | 93.47 | 60.02 | **73.10** | 92.96 | 94.33 | **94.64** | 98.33 | 96.90 | **97.60** | **87.47** |

∗ The original paper of Anomaly Transformer (Xu et al., 2022) adopts the temporal association and reconstruction error as a joint anomaly criterion. For fair comparisons, we only use reconstruction error here.

Table 20: Full results for the classification task. ∗. in the Transformers indicates the name of ∗former. We report the classification accuracy (%) as the result. The standard deviation is within 0.1%.

| Datasets / Models | Classical methods | | | RNN | | | TCN | Transformers | | | | | | | | | | MLP | | | CNN | TimesMixer++ |
|---|---|---|---|---|---|---|---|---|---|---|---|---|---|---|---|---|---|---|---|---|---|---|
| | DTW (1994) | XGBoost (2016) | Rocket (2020) | LSTM (1997) | LSTNet (2018a) | LSSL (2022b) | TCN (2019) | Trans. (2017) | Re. (2020) | In. (2021a) | Pyra. (2022b) | Auto. (2021) | Station. (2022c) | FED. (2022b) | ETS. (2022) | Flow. (2022) | iTrans. (2024) | DLinear (2023) | LightTS. (2022a) | TiDE (2023) | TimesNet (2023a) | (Ours) |
| EthanolConcentration | 32.3 | 43.7 | 45.2 | 32.3 | 39.9 | 31.1 | 28.9 | 32.7 | 31.9 | 31.6 | 30.8 | 31.6 | 32.7 | 28.1 | 31.2 | 33.8 | 28.1 | 32.6 | 29.7 | 27.1 | 35.7 | 39.9 |
| FaceDetection | 52.9 | 63.3 | 64.7 | 57.7 | 65.7 | 66.7 | 52.8 | 67.3 | 68.6 | 67.0 | 65.7 | 68.4 | 68.0 | 66.0 | 66.3 | 67.6 | 66.3 | 68.0 | 67.5 | 65.3 | 68.6 | 71.8 |
| Handwriting | 28.6 | 15.8 | 58.8 | 15.2 | 25.8 | 24.6 | 53.3 | 32.0 | 27.4 | 32.8 | 29.4 | 36.7 | 31.6 | 28.0 | 32.5 | 33.8 | 24.2 | 27.0 | 26.1 | 23.2 | 32.1 | 26.5 |
| Heartbeat | 71.7 | 73.2 | 75.6 | 72.2 | 77.1 | 72.7 | 75.6 | 76.1 | 77.1 | 80.5 | 75.6 | 74.6 | 73.7 | 73.7 | 71.2 | 77.6 | 75.6 | 75.1 | 75.1 | 74.6 | 78.0 | 79.1 |
| JapaneseVowels | 94.9 | 86.5 | 96.2 | 79.7 | 98.1 | 98.4 | 98.9 | 98.7 | 97.8 | 98.9 | 98.4 | 96.2 | 99.2 | 98.4 | 95.9 | 98.9 | 96.6 | 96.2 | 96.2 | 95.6 | 98.4 | 97.9 |
| PEMS-SF | 71.1 | 98.3 | 75.1 | 39.9 | 86.7 | 86.1 | 68.8 | 82.1 | 82.7 | 81.5 | 83.2 | 82.7 | 87.3 | 80.9 | 86.0 | 83.8 | 87.9 | 75.1 | 88.4 | 86.9 | 89.6 | 91.0 |
| SelfRegulationSCP1 | 77.7 | 84.6 | 90.8 | 68.9 | 84.0 | 90.8 | 84.6 | 92.2 | 90.4 | 90.1 | 88.1 | 84.0 | 89.4 | 88.7 | 89.6 | 92.5 | 90.2 | 87.3 | 89.8 | 89.2 | 91.8 | 93.1 |
| SelfRegulationSCP2 | 53.9 | 48.9 | 53.3 | 46.6 | 52.8 | 52.2 | 55.6 | 53.9 | 56.7 | 53.3 | 53.3 | 50.6 | 57.2 | 54.4 | 55.0 | 56.1 | 54.4 | 50.5 | 51.1 | 53.4 | 57.2 | 65.6 |
| SpokenArabicDigits | 96.3 | 69.6 | 71.2 | 31.9 | 100.0 | 100.0 | 95.6 | 98.4 | 97.0 | 100.0 | 99.6 | 100.0 | 100.0 | 100.0 | 100.0 | 98.8 | 96.0 | 81.4 | 100.0 | 95.0 | 99.0 | 99.8 |
| UWaveGestureLibrary | 90.3 | 75.9 | 94.4 | 41.2 | 87.8 | 85.9 | 88.4 | 85.6 | 85.6 | 85.6 | 83.4 | 85.9 | 87.5 | 85.3 | 85.0 | 86.6 | 85.9 | 82.1 | 80.3 | 84.9 | 85.3 | 88.2 |
| Average Accuracy | 67.0 | 66.0 | 72.5 | 48.6 | 71.8 | 70.9 | 70.3 | 71.9 | 71.5 | 72.1 | 70.8 | 71.1 | 72.7 | 70.7 | 71.0 | 73.0 | 70.5 | 67.5 | 70.4 | 69.5 | _73.6_ | **75.3** |

classification, anomaly detection, as well as few-shot and zero-shot tasks, demonstrating remarkable capabilities. This gives it broad prospects in various real-world applications, such as energy and power forecasting with significant seasonal fluctuations, complex and variable weather forecasting, rapidly changing financial market predictions, and demand forecasting in supply chains, all of which it is highly applicable to. It can also excel in various anomaly detection scenarios commonly found in the industry. By leveraging the capabilities of TIMEMIXER++, we can effectively promote the development of various real-world applications related to time series analysis tasks.

**Academic research**  As the pioneering study on a general time series pattern machine, we posit that TIMEMIXER++ holds significant potential to advance research in the domain of time series analysis. Our innovative approach involves converting time series data into time images and implementing hierarchical mixing across different scales and resolutions, which can provide substantial inspiration for future research endeavors in this field. Furthermore, it is noteworthy that our method of employing axial attention within the depth space to extract seasonality and trends from time images surpasses traditional shallow decomposition techniques, such as moving averages and FFT-based decomposition. This represents the first effective methodology for decomposing time series within the deep embedding, promising to catalyze further scholarly investigation.

**Model Robustness**  The robustness of TIMEMIXER++ is evidenced by its performance across a diverse range of time series analysis tasks. In our extensive evaluation, TIMEMIXER++ was tested on 8 different types of tasks and over 30 well-known benchmarks, competing against 27 advanced baseline models. The results highlight its ability to consistently deliver high performance, demonstrating resilience to the variability and complexity inherent in time series data. This robustness is indicative of TIMEMIXER++'s capability to maintain accuracy and reliability across various scenarios, making it a versatile tool in the field of time series analysis. Its robust nature ensures that it can effectively handle noise and fluctuations within data, providing stable and dependable outcomes even in challenging conditions.

# K  LIMITATIONS AND FUTURE WORK

TIMEMIXER++ consistently delivers state-of-the-art performance across a wide range of tasks, including long-term and short-term forecasting, classification, anomaly detection, as well as few-shot and zero-shot learning. This underscores its exceptional representational capacity and robust generalization as a time series pattern machine. However, it is important to acknowledge the recent shift in focus toward large time series language and foundation models, which emphasize continuous scaling of data and parameters, becoming a dominant paradigm in the field. In contrast, due to limitations in the quality and scale of available time series data, the parameter sizes of current advanced deep time series models remain relatively modest. Addressing the challenge of applying scaling laws to time series pattern machines is therefore essential. In this study, we introduced an effective backbone model as a first step toward building a universal time series pattern machine. Future research will focus on constructing large-scale time series datasets to further explore scaling laws for TIMEMIXER++, an exciting and promising direction for continued investigation.

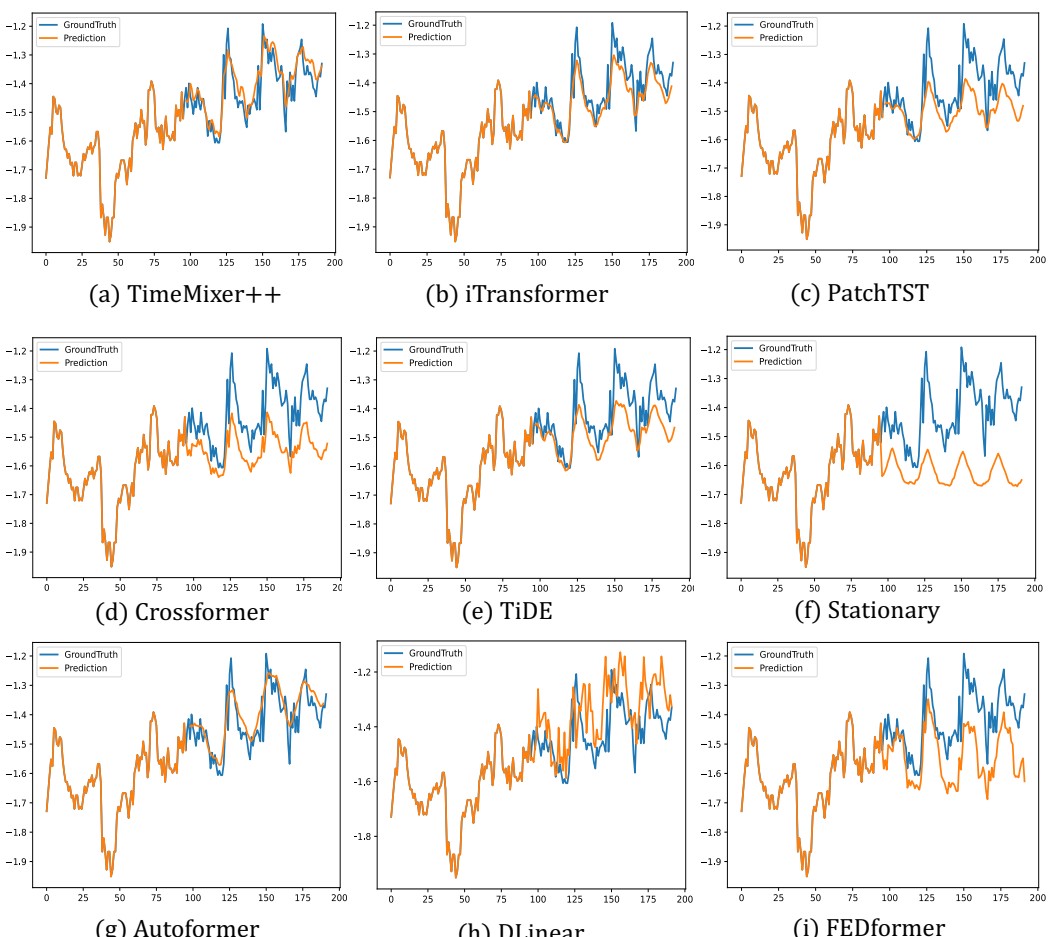

Figure 16: Prediction cases from ETTh1 by different models under the input-96-predict-96 settings. Blue lines are the ground truths and orange lines are the model predictions.

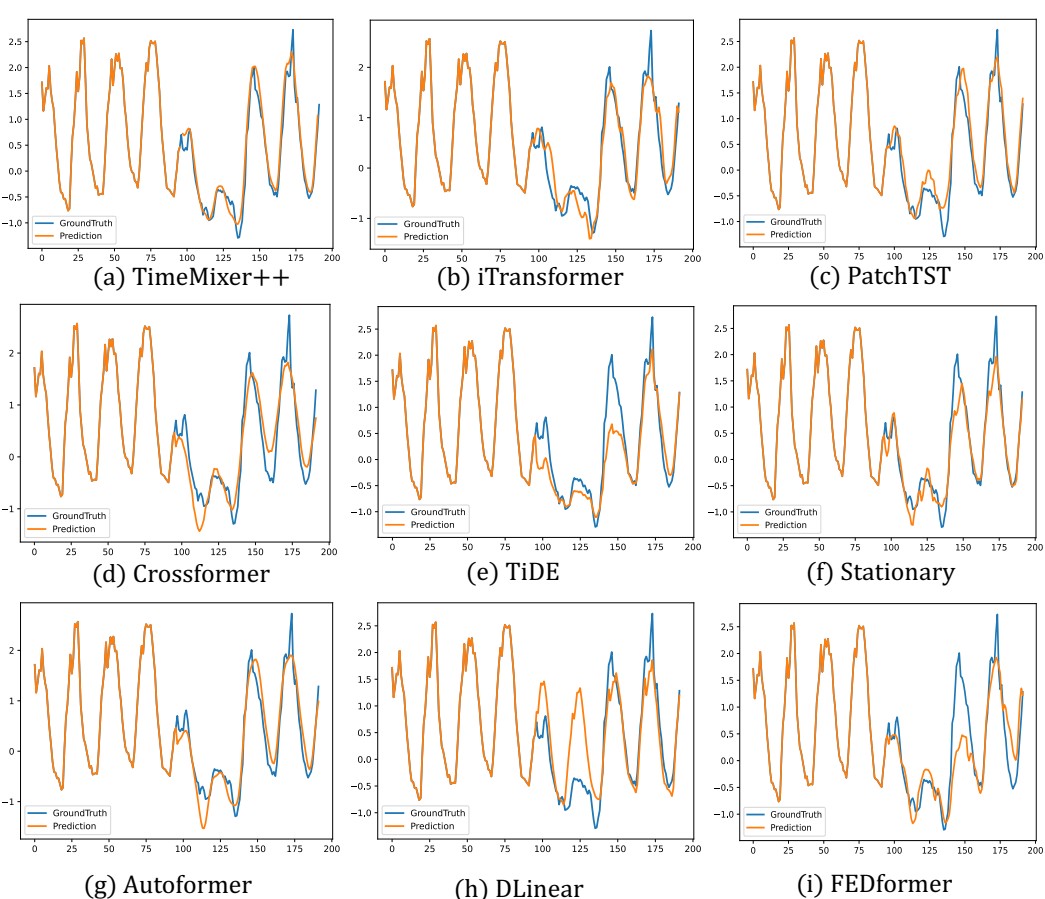

Figure 17: Prediction cases from Electricity by different models under input-96-predict-96 settings.

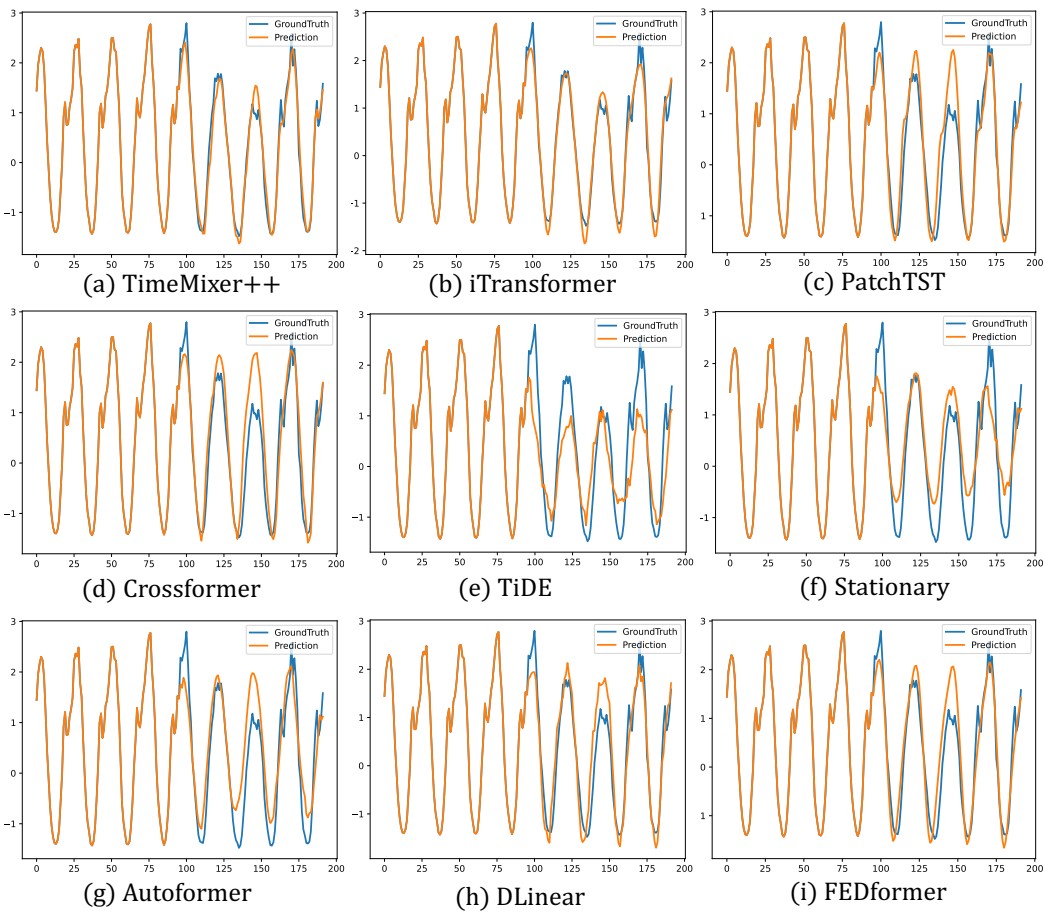

Figure 18: Prediction cases from Traffic by different models under the input-96-predict-96 settings.

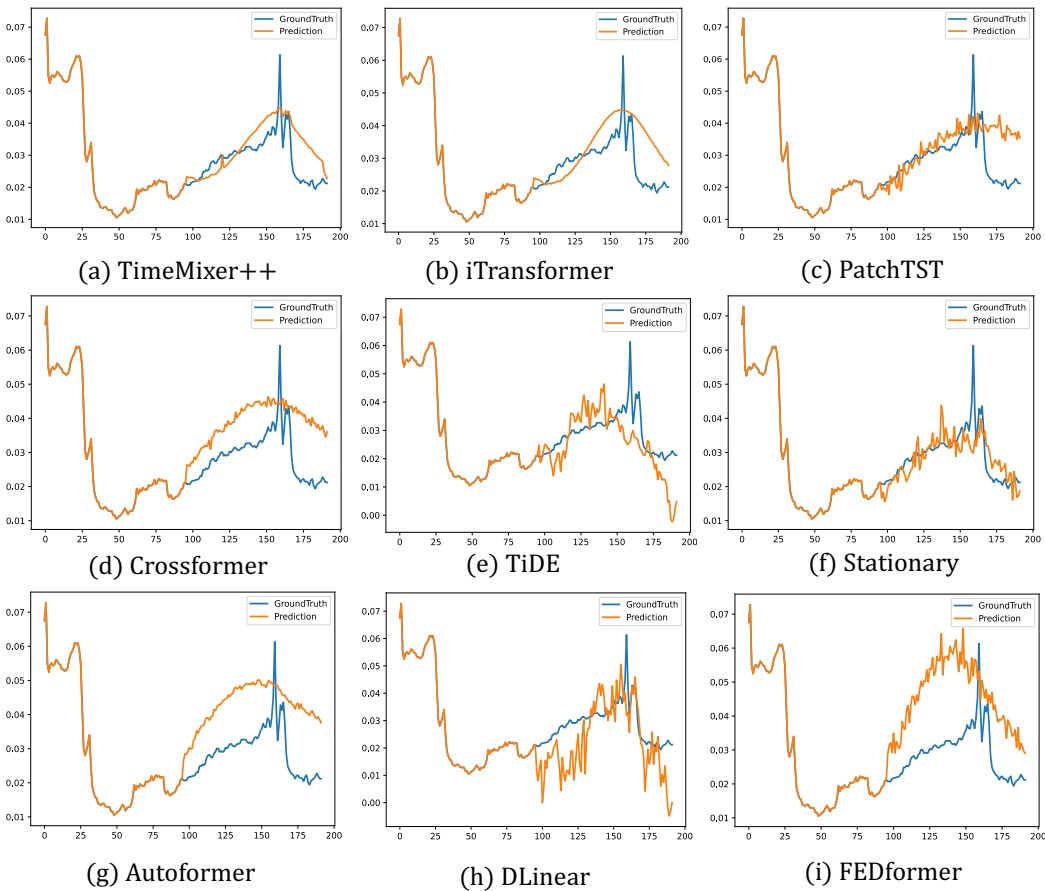

Figure 19: Prediction cases from Weather by different models under the input-96-predict-96 settings.

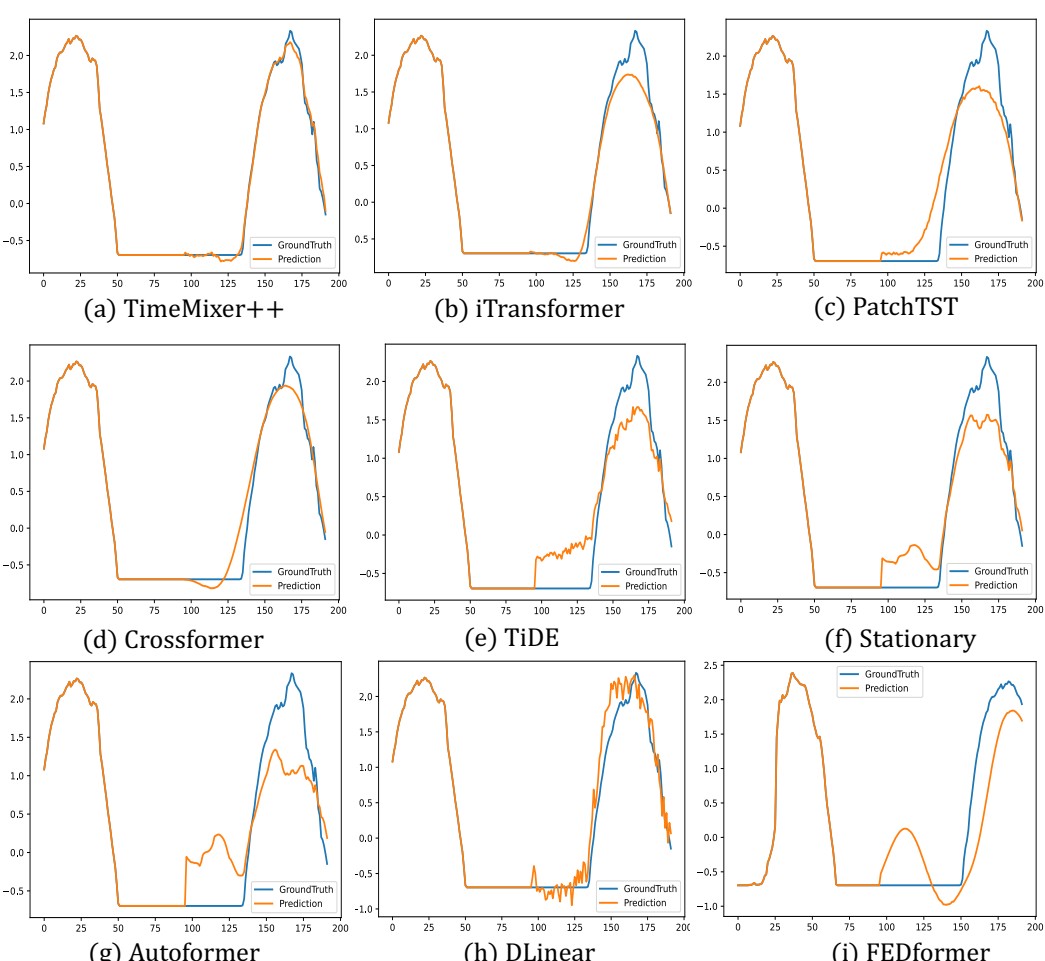

Figure 20: Showcases from Solar-Energy by different models under the input-96-predict-96 settings.

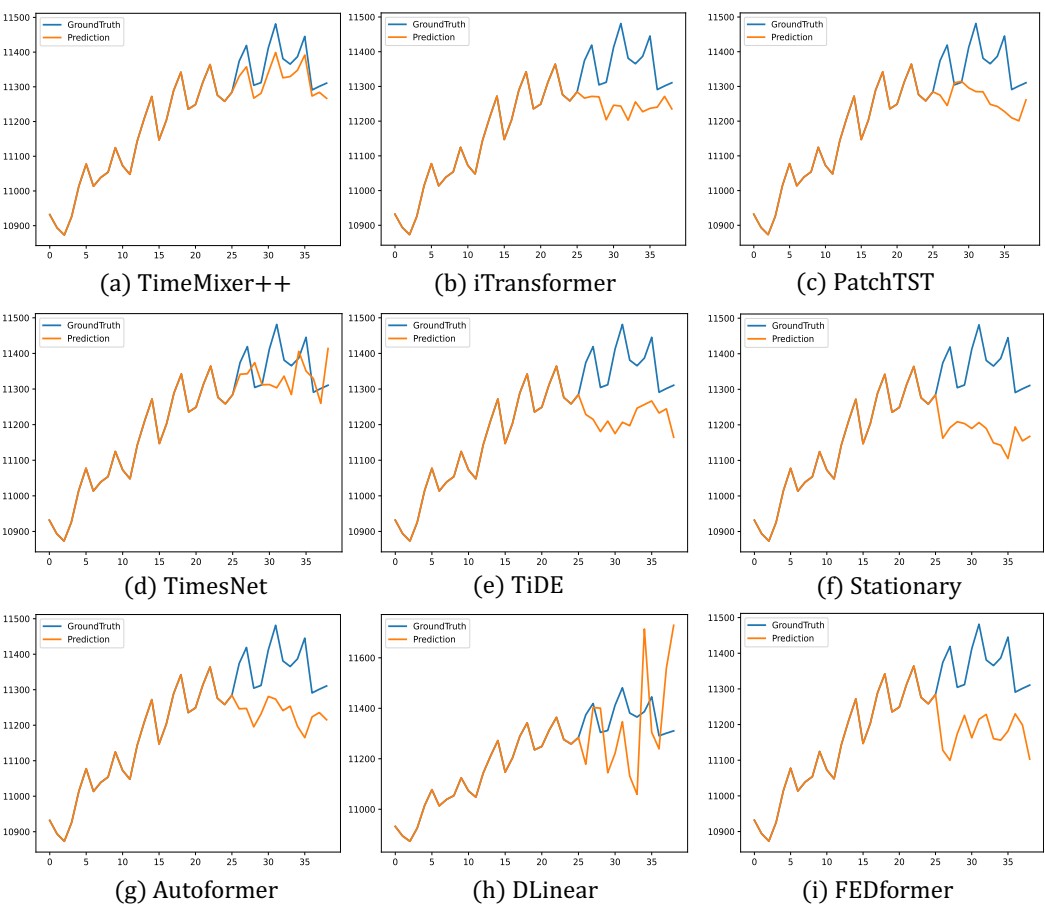

Figure 21: Showcases from the M4 dataset by different models under the input-36-predict-18 settings.

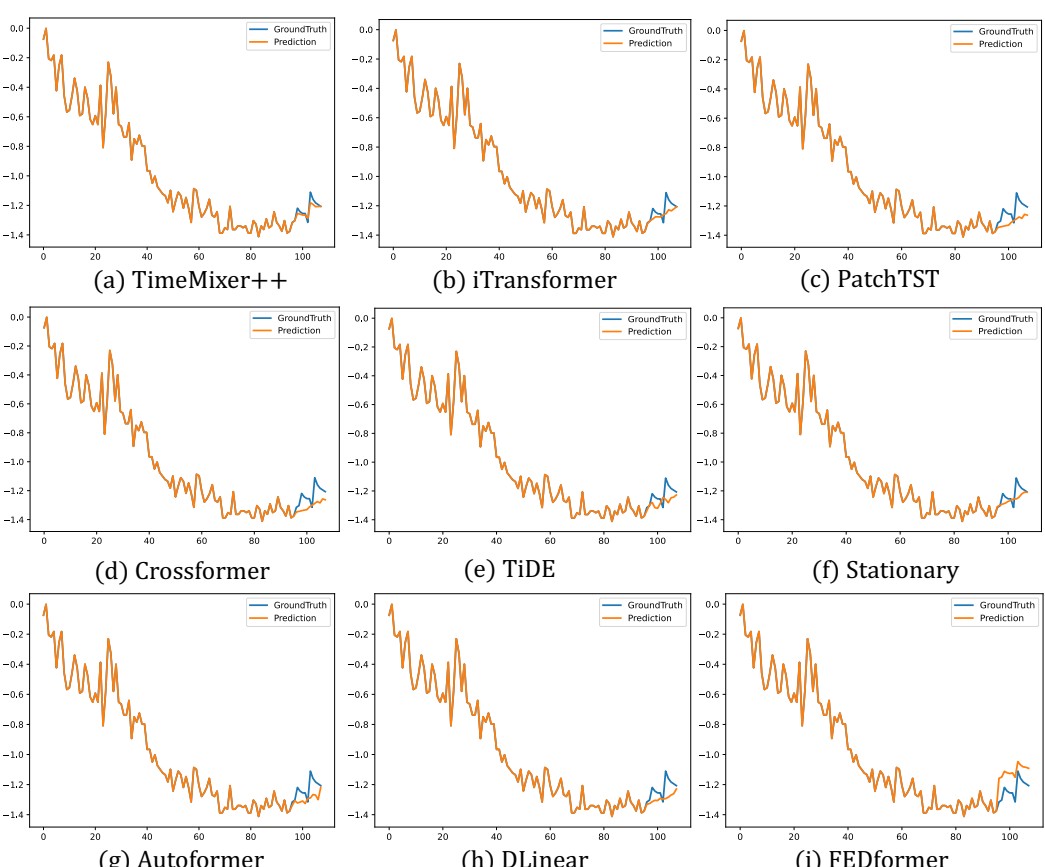

Figure 22: Showcases from PEMS03 by different models under the input-96-predict-12 settings.

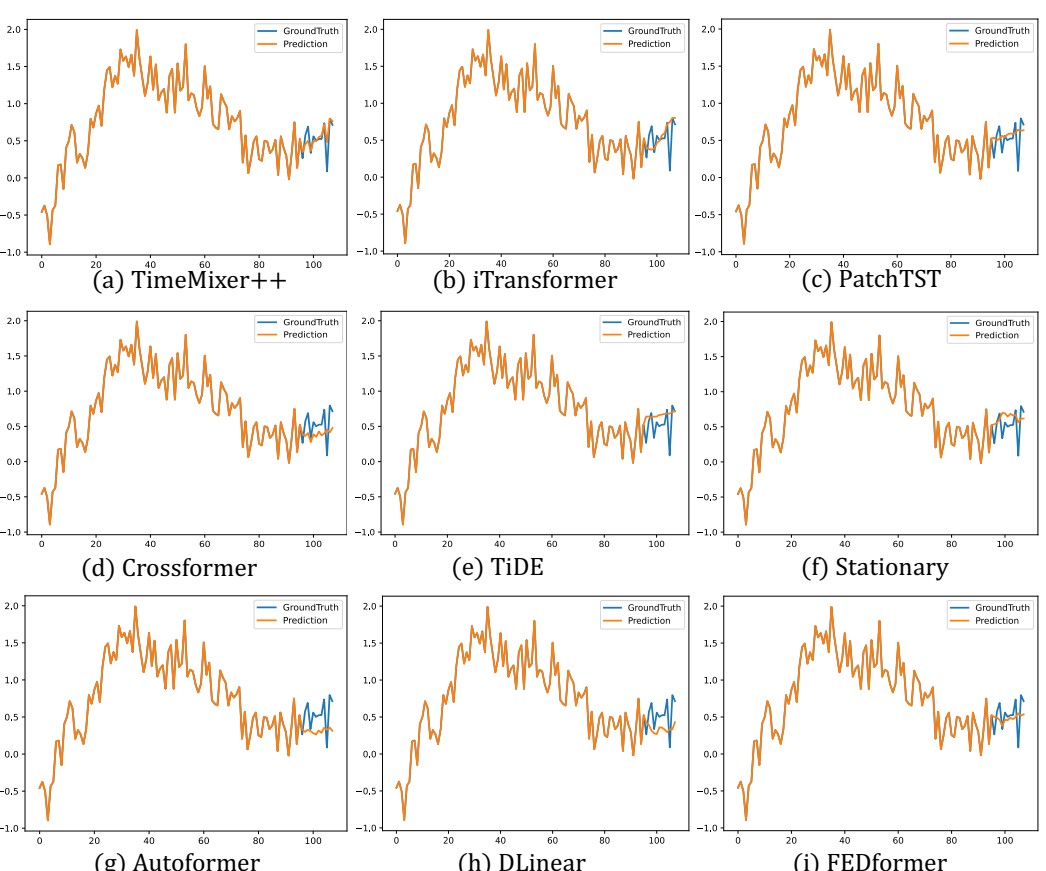

Figure 23: Showcases from PEMS04 by different models under the input-96-predict-12 settings.

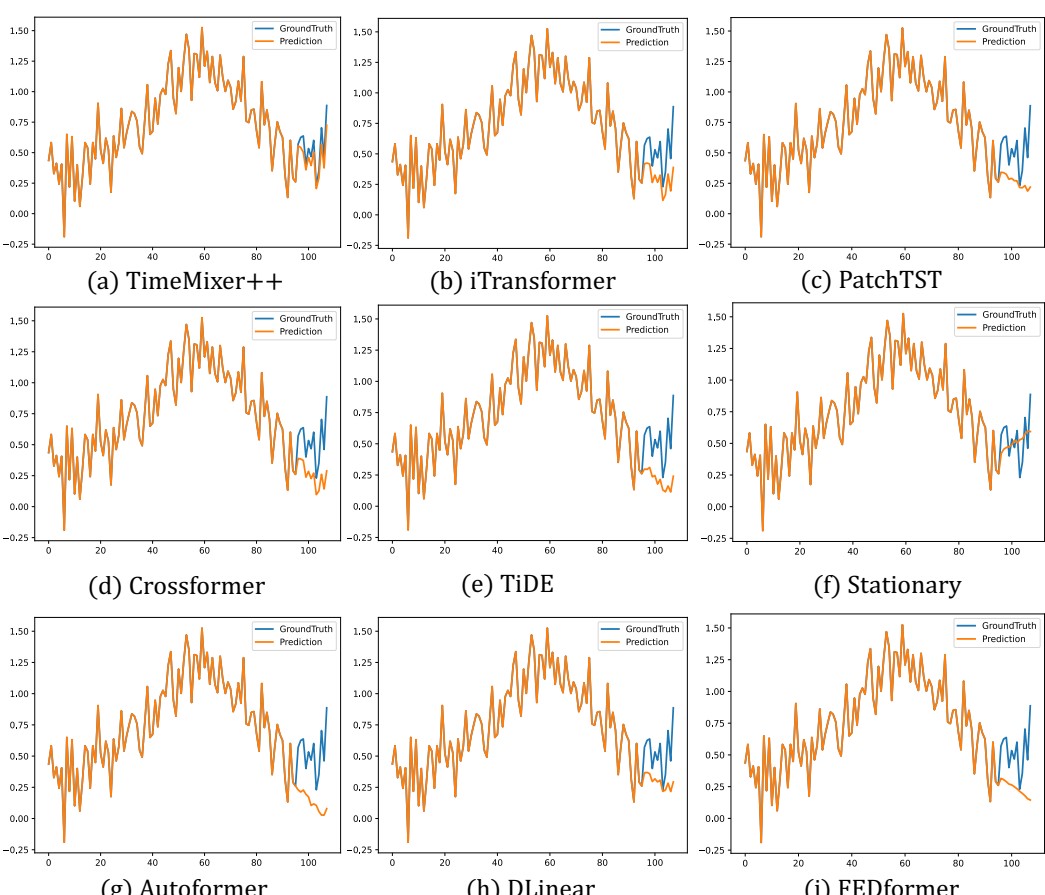

Figure 24: Showcases from PEMS07 by different models under the input-96-predict-12 settings.

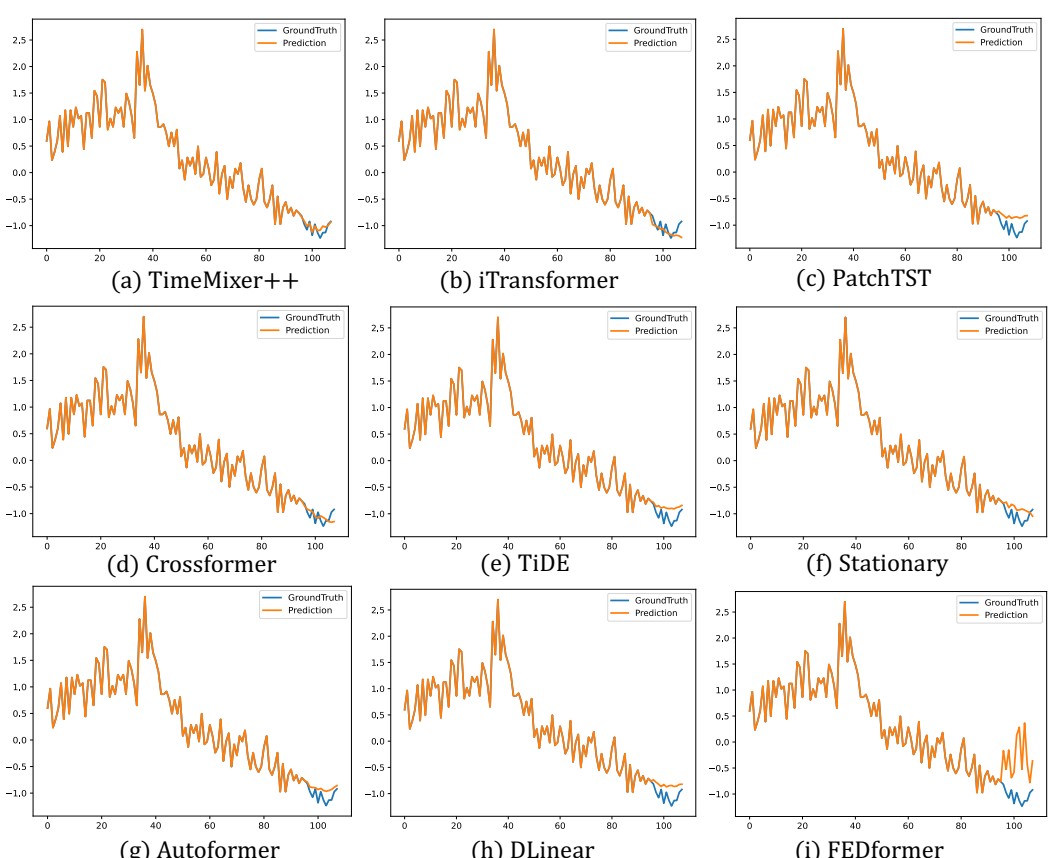

Figure 25: Showcases from PEMS08 by different models under the input-96-predict-12 settings.

