# OpenReview forum: "TimeMixer++: A General Time Series Pattern Machine for Universal Predictive Analysis"
_ICLR.cc/2025/Conference — ICLR 2025 Oral_

### Official Review · Reviewer_9h69 · 2024-10-21

**Soundness:** 2
**Presentation:** 3
**Contribution:** 2
**Rating:** 6
**Confidence:** 4

**Summary:**

The paper introduces a time series pattern machine method called TimeMixer++ for processing multiscale time series. The method transforms time series into multi-resolution time images to enable pattern extraction with respect to temporal and frequency domains followed by three dominant modules - (1) input projection; (2) a stack of Mixerblocks; (3) output projection.  Extensive experiments show the proposed method obtains improvement over the well-established competing methods.

**Strengths:**

The methods used in the paper (e.g., time imaging and image decomposition) are very interesting. The evaluation is comprehensive: the authors discuss long and short-term forecasting, zero-short forecasting, classification, and anomaly detection.

**Weaknesses:**

In terms of the forecasting results shown in Tables 3 and 4, the performance gain is negligible, and such minor improved performance certainly can be attributed to the parameter tuning, e.g., a well-tuned parameter settings for TimeMixer++ while a weak parameter settings for other competing methods.

The paper barely offers insights both theoretically and experimentally. The theoretical understanding of the improvement as well as its time imaging and multi-resolution mixing is lacking, mostly based on intuition and simply blending the models.

There are some papers that already discussed the use of frequency analysis and the frequency components extraction for model deployment (e.g., [1][2][3]) to capture the periodic patterns, and they all claim it can capture the global interaction and patterns among time series, so what is the benefits of introducing multi-resolution time imaging, and it is worthwhile to compare them in ablation study? In addition, it is encouraged to cite the papers [1][2][3] if not yet in the references.



References:
[1] Koopa: Learning Non-stationary Time Series Dynamics with Koopman Predictors https://arxiv.org/pdf/2305.18803
[2] TFDNet: Time-Frequency Enhanced Decomposed Network for Long-term Time Series Forecasting https://arxiv.org/abs/2308.13386
[3] FEDNET: FREQUENCY ENHANCED DECOMPOSED NETWORK FOR OUT-OF-DISTRIBUTION TIME SERIES CLASSIFICATION https://openreview.net/forum?id=OVu9DsOjgH

**Questions:**

[1] Questions follow from the points listed in the weakness section.
[2] What is the naming of the method, i.e., TimeMixer++ in terms of? or just because both the methods target processing multi-scale time series?
[3] What is the connection with the method TimeMixer?

---

> ### Author Response · Authors · 2024-11-21
> **Response to Reviewer 9h69 [Part 1]**
>
> Many thanks to Reviewer 9h69 for providing the insightful review and comments.
>
> > **W1:** "In terms of the forecasting results shown in Tables 3 and 4, the performance gain is negligible, and such minor improved performance certainly can be attributed to the parameter tuning."
>
> We would like to respectfully clarify that **the performance improvements achieved by TimeMixer++ are both substantial and consistent across all datasets and metrics, far exceeding what could be attributed to parameter tuning.** Below, we provide a focused discussion to address this concern.
>
> - As shown in **Table 3 (Forecasting Results)**, TimeMixer++ **consistently achieves the best performance** across all three metrics (MAE, MAPE, and RMSE). For example, TimeMixer++ achieves an **8.6%** relative improvement in MAE compared to the second-best method, TimeMixer. Additionally, it **significantly outperforms strong baselines** such as SCINet, Crossformer, and PatchTST, with relative reductions in MAE of **16.8%, 16.4%, and 30.9%**, respectively. These consistent improvements across datasets and metrics highlight TimeMixer++'s effectiveness, which **cannot be explained by minor parameter adjustments**.
>
> - In **Table 4 (Imputation Results)**, TimeMixer++ delivers the **best performance on 11 out of 12 metrics** across six datasets. Compared to TimesNet, the second-best model, TimeMixer++ achieves a **25.6% improvement** in average MSE (0.0632) and a **17.4% improvement** in average MAE (0.1487). These results further emphasize the model's robustness and ability to generalize across diverse imputation tasks.
>
> Regarding the concern about parameter tuning, we want to emphasize that **we ensured a rigorous and fair comparison by following a consistent experimental pipeline**. Specifically, for baselines with the same experimental settings as our main study, we **directly report** the results from TimesNet [2]. For scenarios where the settings differ or tasks were not implemented, we reproduced the baselines **using the benchmark framework from the time series library** [1], which is widely adopted in existing studies [2,3] and ensures high consistency. This pipeline is essential given the scope of our evaluation, which **includes 27 baselines and 30 benchmarks**. Deviating from this approach would introduce inconsistencies and undermine the reliability of the results.  The experimental setup is detailed in Appendix A.
>
> In summary, **the substantial and consistent improvements achieved by TimeMixer++, as demonstrated in Tables 3 and 4, clearly reflect the robustness and effectiveness of our approach.** These results were obtained by rigorously adhering to the established experimental pipeline.
>
> - [1] https://github.com/thuml/Time-Series-Library
> - [2] Wu, Haixu, et al. "Timesnet: Temporal 2d-variation modeling for general time series analysis." arXiv preprint arXiv:2210.02186 (2022).
> - [3] Liu, Yong, et al. "itransformer: Inverted transformers are effective for time series forecasting." arXiv preprint arXiv:2310.06625 (2023).

---

> ### Author Response · Authors · 2024-11-21
> **Response to Reviewer 9h69 [Part 2]**
>
> > **W2:** The paper barely offers insights both theoretically and experimentally. The theoretical understanding of the improvement as well as its time imaging and multi-resolution mixing is lacking.
>
> We would like to **clarify a potential misunderstanding** and **re-emphasize the position of our work within the literature**.
>
> - Our primary objective in this paper is to propose a novel pattern extraction model for general time series analysis, which we refer to as the **Time Series Pattern Machine (TSPM)**, as introduced on line 39 in the original paper. **We have not made any claims regarding theoretical contributions in this submission. Instead, our focus is on introducing a practical and innovative framework that empirically advances the state of the art across eight diverse time series analysis tasks.**
>
> - The core of our approach lies in disentangling seasonality and trend patterns from multi-resolution time images using Time Image Decomposition (TID), followed by Multi-Scale Mixing (MCM) and Multi-Resolution Mixing (MRM). **This design achieves empirically significant improvements and establishes new benchmarks across various tasks, contributing to the broader time series analysis literature.**
>
> - Moreover, our model is **inspired by established theories** in time series analysis and signal processing, particularly **multi-resolution analysis**, which is widely used to decompose signals into components that capture variations across scales. This theoretical framework, rooted in techniques like wavelet transforms and multi-scale signal processing, serves as the foundation for our approach. By introducing **the concept of multi-resolution time imaging**, we transform 1D multi-scale time series into 2D images, enabling a **structured disentanglement of seasonal and trend components in latent spaces**. This enables TimeMixer++ to effectively capture global and localized patterns in a way that is well-grounded in signal processing principles.
>
> To place our work in context, **iTransformer** and **TimesNet**, two well-established and state-of-the-art benchmark models in the literature, provide strong baselines for comparison. **To highlight the experimental contributions of TimeMixer++, we present a concise summary of the average performance of these models across multiple tasks**, demonstrating the robustness and adaptability of TimeMixer++.
>
> | Method        | Long term Forecasting |                  | Uni. Short Term Forecasting | Mul. Short Term Forecasting | Imputation |                  | Few-Shot |                  | Zero-Shot |                  | Classification | Anomaly Detection |
> |---------------|-----------------------|------------------|-----------------------------|------------------|-----------------------------|------------------|------------|------------------|----------|------------------|-----------|------------------|
> |               | MSE                   | MAE              | SMAPE                      | MAPE             | MSE                         | MAE              | MSE        | MAE              | MSE      | MAE      | ACC (%)        | F1-Score (%)      |
> | TimeMixer++   | **0.286**             | **0.301**        | **11.448**                 | **10.08**        | **0.063**                   | **0.149**        | **0.332**  | **0.371**        | **0.386**| **0.408**        | **75.9**  | **87.47**        |
> | TimesNet      | 0.363                 | 0.347            | 11.829                     | 12.69            | 0.085                       | 0.180            | 0.491      | 0.446            | 0.527    | 0.465            | 73.6      | 86.34            |
> | iTransformer  | 0.310                 | 0.305            | 12684                      | 12.55            | 0.103                       | 0.191            | 0.394      | 0.442            | 0.444    | 0.434            | 70.5      | 76.98            |
>
>
> We can have the following observations:
> - These results provide compelling evidence of the effectiveness of TimeMixer++ in **capturing diverse time series patterns**.
> - **TimeMixer++ achieves consistent improvements over TimesNet and iTransformer, underscoring its contribution to advancing the state of the art in general time series analysis**.
>
> Furthermore, we hope that the empirical insights presented in Figure 1 (right) will **inspire the research community** to further explore the TSPM paradigm through **the lens of representation learning**. By harnessing its **strong pattern extraction capabilities**, TimeMixer++ **exhibits robust adaptability across tasks**, as demonstrated by the comprehensive representation learning experiments in Appendix E of the _$\underline{\text{revised paper}}$_. These experiments provide valuable insights into why TimeMixer++ performs effectively across diverse tasks, highlighting its ability to adapt learned representations to task-specific requirements.
>
> We hope this additional information addresses the reviewer’s concerns.

---

> > ### Comment · Reviewer_9h69 · 2024-11-27
> >
> > Do you have any references for the parameter settings were used in these results?

---

> ### Author Response · Authors · 2024-11-21
> **Response to Reviewer 9h69 [Part 3]**
>
> > **W3:** There are some papers that already discussed the use of frequency analysis and the frequency components extraction for model deployment (e.g., [1][2][3]) to capture the periodic patterns, and they all claim it can capture the global interaction and patterns among time series, so what is the benefits of introducing multi-resolution time imaging, and it is worthwhile to compare them in ablation study? In addition, it is encouraged to cite the papers [1][2][3] if not yet in the references.
>
> We appreciate the reviewer’s insightful comments.
> While we recognize the relevance of the mentioned works, **a direct ablation comparison with these three studies may not be the most appropriate approach due to their distinct differences in model design, pattern learning capabilities, and respective objectives within the literature.**
>
>
> **(1) For Koopa [1]**
>
> - **The architecture of TimeMixer++ is fundamentally different from Koopa**. While Koopa leverages the Fast Fourier Transform (FFT) to extract time-variant and time-invariant components, it does not incorporate advanced mixing strategies, as it primarily relies on modern Koopman theory. Specifically, Koopa processes the two components separately using a purely MLP-based encoder-decoder architecture and combines their outputs **through summation** to make predictions. In contrast, TimeMixer++ employs two mixing strategies to hierarchically and adaptively learn the input-output mapping function.
> - Moreover, **Koopa is designed solely for forecasting tasks**, as presented in their paper. In contrast, **TimeMixer++ adopts learnable and flexible decomposition and mixing strategies, enabling it to achieve superior performance across general time series analysis tasks**, effectively serving as a comprehensive Time Series Pattern Machine (TSPM).
>
>
> **(2) For TFDNet [2]**
>
> - **The architecture of TimeMixer++ is also fundamentally different from TFDNet**. While TFDNet leverages the seasonal-trend decomposition and mixing strategy, the decomposition is conducted by moving average directly. Besides, **it does not incorporate advanced mixing strategies**. Similar to Koopa, TFDNet processes the two components separately using kernel strategies and feed-forward network (FFD),  and finally combines their outputs **through concatenation** to make predictions.
>
> - Moreover, **TFDNet is also designed solely for forecasting tasks**, as presented in their paper. In contrast, **by adopting flexible decomposition and mixing strategies, TimeMixer++ achieves superior performance across a wide range of time series analysis tasks**.
>
> **(3) For FEDNet [3]**
>
> - **The training paradigm and architecture of TimeMixer++ are fundamentally different from those of TFDNet**. While FEDNet employs frequency-domain decomposition to separate time-variant and time-invariant components, which are processed using **encoder-decoder architectures**, its training paradigm relies on **contrastive learning**.
>
> - Moreover, FEDNet was proposed to address **out-of-distribution (OOD) time series classification problems**. Their code is **no longer accessible** at this stage, making it infeasible for us to conduct the experiments.
>
>
> We highly appreciate reviewer's efforts and valuable feedback. In response to your comments, **we have updated the introduction and related work sections in the _$\underline{\text{revised paper}}$_, adding the corresponding citations to better clarify the position of our work.**
>
>
>
> - [1] Liu, Yong, et al. "Koopa: Learning non-stationary time series dynamics with Koopman predictors." *Advances in Neural Information Processing Systems 36* (2024).
> - [2] Luo, Yuxiao, Ziyu Lyu, and Xingyu Huang. "TFDNet: Time-Frequency Enhanced Decomposed Network for Long-term Time Series Forecasting." *arXiv preprint arXiv:2308.13386* (2023).
> - [3] FEDNet: Frequency Enhanced Decomposed Network for Out-of-distribution Time Series Classification. https://openreview.net/forum?id=OVu9DsOjgH

---

> ### Author Response · Authors · 2024-11-21
> **Response to Reviewer 9h69 [Part 4]**
>
> > **Q1&2:** "What is the naming of the method, i.e., TimeMixer++ in terms of? What is the connection with the method TimeMixer?"
>
> Thank you for your question.
>
> **Similarities:**
> Both **TimeMixer** and **TimeMixer++** model multi-scale time series by decomposing them into seasonal and trend components, which are subsequently mixed to capture underlying temporal patterns.
>
> **Differences**:
> **1. Decomposition**:
> - **TimeMixer**: Uses moving averages for seasonal and trend decomposition, which is limited in flexibility.
> - **TimeMixer++**: Replaces moving averages with **axial attention** applied to **multi-resolution time images**, enabling more precise and adaptive pattern extraction **in the latent space**.
>
> **2. Mixing Strategies**:
> - **TimeMixer**: Relies **solely on hierarchical MLPs for temporal mixing** and **ignores channel mixing**.
> - **TimeMixer++**: Introduces **hierarchical convolutions** with **inception blocks** for parameter-efficient mixing and enhances **channel mixing** through the use of channel-wise attention.
>
> **3. Roles as general TSPM**:
> - It is important to emphasize the fundamental difference in objectives. **TimeMixer++**, as a **General TSPM**, is designed to handle **general time series analysis tasks**. Its primary goal is to develop powerful representation capabilities that enable robust performance across diverse tasks.
> - In contrast, TimeMixer is specifically designed to optimize **time series forecasting**.
> - We present a **concise summary** of the average performance of these models across multiple tasks, demonstrating the effectiveness and adaptability of TimeMixer++.
>
>
> | Method        | Long term Forecasting |                  | Uni. Short Term Forecasting |Mul. Short Term Forecasting  | Imputation |                  | Few-Shot |                  | Zero-Shot |                  | Classification | Anomaly Detection |
> |---------------|-----------------------|------------------|-----------------------------|------------------|-----------------------------|------------------|------------|------------------|----------|------------------|-----------|------------------|
> |               | MSE                   | MAE              | SMAPE                      | MAPE             | MSE                         | MAE              | MSE        | MAE              | MSE      | MAE              | ACC (%)        | F1-Score (%)      |
> | TimeMixer++   | **0.286**             | **0.301**        | **11.448**                 | **10.08**        | **0.063**                   | **0.149**        | **0.332**  | **0.371**        | **0.386**| **0.408**        | **75.9**  | **87.47**        |
> | TimeMixer     | 0.314                 | 0.329            | 11.723                     | 10.59            | 0.103                       | 0.212            | 0.374      | 0.390            | 0.467    | 0.446            | /         | /                |

---

> > ### Comment · Reviewer_9h69 · 2024-11-27
> >
> > It looks good of these improvements, thanks for your clarifications.

---

> ### Author Response · Authors · 2024-11-25
> **Request of Reviewer's attention and feedback**
>
> Dear Reviewer,
>
> Thanks for your valuable review, which has inspired us to improve our paper further.
> This is a kind reminder that it has been four days since we submitted our rebuttal. We kindly ask if our responses have addressed your concerns.
>
> Following your suggestions, we have implemented the following updates:
>
> - **Clarify the position of our work**, emphasizing that we introduce a practical and innovative model that empirically
> advances the state of the art across eight time series analysis tasks.
> - **Revise the introduction and related work sections** in the _$\underline{\text{revised paper}}$_,  **adding relevant citations** and providing a
> detailed discussion of the mentioned works.
> - **Provide a detailed comparison between TimeMixer++ and TimeMixer**, including a result summary table to enhance clarity and insight.
>
>
>
> In this paper, we propose TimeMixer++ as a general Time Series Pattern Machine (TSPM), supported by extensive experiments, visualizations,
> and ablations to substantiate our claims.
> All the revisions have been incorporated into the  _$\underline{\text{revised paper}}$_ for your review.
>
> Thank you again for your dedication and feedback. We look forward to hearing your thoughts on our revisions.

---

> ### Author Response · Authors · 2024-11-28
> **Thank you for your Feedback and Request for Further Support**
>
> Thank you for raising the score!
> The results listed in Part 2 were calculated as the mean of the results reported for each experiment in the original paper.
> The parameter settings and references are detailed in Appendix A of the _$\underline{\text{revised paper}}$_.
> Specifically,
> we set the initial learning rate as $10^{-2}$ or $10^{-3}$ and use the ADAM optimizer with L2 loss for model optimization.
> The batch size is 512.
> By default, we configure the number of MixerBlocks $L$ to 2 and set the number of resolutions $K$ to 3.
> We choose the number of scales $M$ according to the time series length to balance performance and efficiency.
> To handle longer series in long-term forecasting, we set $M$ to 3. As for short-term forecasting with
> limited series length, we set $M$ to 1.
> For baselines under the same experimental settings as our main study, we directly report the results from TimesNet [2],
> following standard practice as in prior works [1, 4, 5].
> In scenarios where experimental settings differed or tasks were not previously implemented, we reproduced the baseline results using
> the benchmark framework from the Time-Series Library [2, 3]. This framework is well-known and widely adopted in existing studies [1, 2, 5] and ensures
> reproducibility and consistency. We have supplemented and refined the relevant parameter settings and references in the  _$\underline{\text{revised paper}}$_.
>
> If there are any aspects of the paper that you feel require further clarification or improvement, we would be happy to address them.
> We sincerely appreciate your recognition of our work, and we hope to receive your further support and constructive feedback moving forward.
> Thank you once again for your time and insightful suggestions.
>
> - [1] Liu, Yong, et al. "itransformer: Inverted transformers are effective for time series forecasting." arXiv preprint arXiv:2310.06625 (2023).
> - [2] Wu, Haixu, et al. "Timesnet: Temporal 2d-variation modeling for general time series analysis." arXiv preprint arXiv:2210.02186 (2022).
> - [3] https://github.com/thuml/Time-Series-Library
> - [4] Bian, Yuxuan, et al. "Multi-patch prediction: Adapting llms for time series representation learning." arXiv preprint arXiv:2402.04852 (2024).
> - [5] Liu, Xu, et al. "Unitime: A language-empowered unified model for cross-domain time series forecasting." Proceedings of the ACM on Web Conference 2024. 2024.

---

### Official Review · Reviewer_LKyA · 2024-10-29

**Soundness:** 3
**Presentation:** 3
**Contribution:** 4
**Rating:** 8
**Confidence:** 5

**Summary:**

The paper presents TimeMixer++, an advanced framework designed to enhance general time series analysis. TimeMixer++ integrates multi-resolution time imaging, multi-scale mixing, and dual-axis attention mechanisms to effectively capture and adapt to diverse patterns within time series data. This innovative approach allows for robust and flexible analysis across various temporal scales and resolutions.

**Strengths:**

1. the authors introduce a robust framework TimeMixer++ that leverages multi-resolution time imaging, multi-scale mixing, and dual-axis attention to enhance general time series analysis. They present SOTA results on four different tasks.
2. the integration of both multi-scale and multi-resolution mixing strategies for adaptive pattern extraction demonstrates innovation.
3. the manuscript and appendix are well-prepared, but the authors have not yet released the promised code.

**Weaknesses:**

1. the fonts in the figures should be enlarged for better readability. For example, in Figure 1 (right), the label "Benchmarking model performance across representation analysis in four tasks" appears blurred. Additionally, consider using a single set of legends for all four tasks to enhance clarity.
2. the source code repository has not released for reproducing, i will consider raising the score if the released repository and the consistency of the results.
3. more detail on how it compares to recent models like TimesNet and iTransformer on specific time series tasks would strengthen the paper’s claims.
4. including a discussion on computational efficiency (e.g., FLOPs, memory usage) for different tasks could enhance the paper’s utility.

**Questions:**

See weaknesses (W2, W3, W4).

---

> ### Author Response · Authors · 2024-11-21
> **Response to Reviewer LKyA [Part 1]**
>
> We would like to sincerely thank Reviewer LKyA for providing a detailed review and insightful suggestions.
>
> > **Q1:**  "The fonts in the figures should be enlarged for better readability. For example, in Figure 1 (right), the label "Benchmarking model performance across representation analysis in four tasks" appears blurred. Additionally, consider using a single set of legends for all four tasks to enhance clarity."
>
> Thank you for your valuable feedback. We appreciate your observation regarding the figure's readability. In the _$\underline{\text{revised paper}}$_, we have **updated Figure 1 and provided an enlarged and clearer version in Appendix E**. We believe these improvements improve the visual quality and interpretability of the figure.

---

> ### Author Response · Authors · 2024-11-21
> **Response to Reviewer LKyA [Part 3]**
>
> > **Q3:** "More detail on how it compares to recent models like TimesNet and iTransformer on specific time series tasks would strengthen the paper’s claims."
>
> Thank you for your feedback! We would like to compare TimeMixer++, TimesNet, and iTransformer in terms of **model design, benefits, empirical evidence, and implementation details.**
>
> **(1) Model Design**
>
> The strength of TimeMixer++ lies in its **flexible and effective pattern decomposition and mixing strategies**. Specifically, TimeMixer++ processes multi-scale time series using four key components: (1) Multi-Resolution Time Imaging (MRTI), (2) Time Image Decomposition (TID), (3) Multi-Scale Mixing (MCM), and (4) Multi-Resolution Mixing (MRM).
>
>
> While **TimesNet** also analyzes time series in the frequency domain by transforming 1D time series into 2D tensors, there are key differences:
> - **Pattern Disentanglement**: TimesNet does not disentangle seasonal and trend patterns, limiting its flexibility in handling complex time series data across diverse tasks.
> - **Mixing Strategies**: TimeMixer++ defines multiple scales in the time domain and various resolutions in the frequency domain through down-sampling. It employs task-adaptive strategies like MCM and MRM to extract representative patterns. TimesNet, however, **overlooks time-domain mixing**, which reduces its adaptability.
>
>
> While **iTransformer** applies attention mechanisms for channel mixing, there are notable differences:
> - **Building Blocks**: iTransformer primarily uses **feed-forward networks (FFN)** for encoding 1D time series. In contrast, TimeMixer++ transforms time series into **multi-resolution 2D time images**, enabling dual-axis attention for pattern decomposition and hierarchical convolutions for mixing.
>
> **(2) Benefits of TimeMixer++**
>
> The flexibility of TimeMixer++ is evident in Figure 1 (right), which highlights the relationship between **CKA similarity** and task effectiveness across different scenarios.
>
> - **In forecasting**, TimeMixer++ demonstrates a clear advantage with the **highest CKA similarity (0.94)** and the lowest MSE (0.23), showcasing its ability to effectively align consistent representations with task-specific needs. A similar trend is observed in anomaly detection tasks.
>
> - **For classification**, TimeMixer++ demonstrates superior adaptability, achieving the best accuracy (90%) with **the lowest CKA similarity (0.75)**, effectively learning diverse representations. A similar trend is observed in imputation tasks.
>
> By **dynamically adapting** to the diverse CKA-effectiveness relationships across tasks, **TimeMixer++ consistently outperforms TimesNet and iTransformer**, demonstrating superior flexibility and effectiveness in extracting diverse time series patterns.
>
>
> **(3) Empirical Evidence**
>
> | Method| Long term Forecasting || Uni. Short Term Forecasting | Mul. Short Term Forecasting | Imputation |  | Few-Shot |                  | Zero-Shot | | Classification | Anomaly Detection |
> |--|----|----|----|------|---|---|----|-------|-----|-----|--------|------|
> | | MSE| MAE| SMAPE| MAPE| MSE| MAE | MSE| MAE| MSE| MAE| ACC (%)  | F1-Score (%) |
> | TimeMixer++   | **0.286**| **0.301**| **11.448**| **10.08**| **0.063**| **0.149**| **0.332**  | **0.371**| **0.386**| **0.408** | **75.9**|**87.47**|
> | TimesNet| 0.363| 0.347| 11.829| 12.69| 0.085| 0.180| 0.491| 0.446| 0.527| 0.465| 73.6| 86.34 |
> | iTransformer  | 0.310| 0.305| 12.684| 12.55| 0.103| 0.191| 0.394| 0.442| 0.444 | 0.434| 70.5 | 76.98|
>
> - These experimental results validate the design advantages of TimeMixer++, demonstrating how its flexible and effective pattern decomposition and mixing strategies—enabled by components like MRTI, TID, MCM, and MRM—**consistently outperform competing models like TimesNet and iTransformer across diverse tasks**.
>
> **(4). Implementation Details**
>
> The experimental setup is detailed in Appendix A. **For baselines with the same experimental settings** as our main study, we **directly report** the results from TimesNet [2]. For scenarios **where the settings differ or tasks are not implemented**, we **reproduced** the baselines using the benchmark framework from the **time series library** [1], which is **widely adopted in existing studies** [2,3] and ensures high consistency. The details of the hyperparameter configurations are provided in Appendix A. This pipeline is essential given the scope of our evaluation, which **includes 27 baselines and 30 benchmarks**. Deviating from this approach would introduce inconsistencies and undermine the reliability of the results.
>
> We hope this additional information addresses the reviewer’s concerns.
>
>
> - [1] https://github.com/thuml/Time-Series-Library
> - [2] Wu, Haixu, et al. "Timesnet: Temporal 2d-variation modeling for general time series analysis." arXiv preprint arXiv:2210.02186 (2022).
> - [3] Liu, Yong, et al. "itransformer: Inverted transformers are effective for time series forecasting." arXiv preprint arXiv:2310.06625 (2023).

---

> > ### Comment · Reviewer_LKyA · 2024-11-21
> >
> > Thank you for your rebuttal. As promised, I have updated my rating since you updated the anonymous GitHub repository.

---

> > > ### Author Response · Authors · 2024-11-22
> > >
> > > We are thrilled that our responses have effectively addressed your questions and comments. We would like to express our sincerest gratitude for taking the time to review our paper and provide us with such detailed feedback.

---

### Official Review · Reviewer_e5Jj · 2024-11-04

**Soundness:** 4
**Presentation:** 4
**Contribution:** 4
**Rating:** 10
**Confidence:** 4

**Summary:**

The paper presents TIMEMIXER++, a general-purpose model for various time series tasks, including forecasting, classification, anomaly detection, and imputation. Utilizing a multi-scale, multi-resolution framework, the proposed method transforms time series data into multi-resolution images to capture complex temporal and frequency-domain patterns, enabling flexibility across analytical applications. The model’s approach includes dual-axis attention for decomposing seasonal and trend components and hierarchical multi-scale and multi-resolution mixing to integrate patterns across scales. The proposal achieves strong performance across eight benchmark tasks, outperforming both general-purpose and task-specific models. This work contributes to advancing time series analysis with new state-of-the-art benchmarks across settings.

**Strengths:**

S1. The proposed model captures both short- and long-term dependencies by transforming time series data into multi-resolution images, enabling the analysis of complex temporal and frequency-domain patterns that challenge traditional models. The authors validate this with experimental results showing the new architecture outperforms SOTA models on most standard benchmarks. The ablation study helps validate the importance of the individual parts of the architecture – the channel mixing, image decomposition, and multi-scale and multi-resolution mixing. This approach continues to validate the benefits of integrating image analysis techniques with time series tasks.

S2. The architecture is flexible for supporting different kinds of time-series tasks. The hierarchical multi-scale and multi-resolution mixing modules enable the model to flexibly adapt across various time series tasks, from forecasting to anomaly detection, promoting robust and accurate performance across applications.

S3. Empirical Validation: the testing in this paper on eight benchmark time series tasks, including the hyperparameter ablation results, shows TIMEMIXER++ consistently surpasses both general-purpose and task-specific models, affirming its potential as a high-performance, general-purpose solution for time series analysis. The experiments were very thorough.

**Weaknesses:**

W1. There is little exploration of scaling of model size, which would be an interesting avenue for validating the model architecture in a zero shot setting. The current zero-shot experiments are primarily in-domain and not cross-task.

**Questions:**

Q1. The proposed architecture adds significant computational cost to the internal representation of the model compared to vanilla transformers and some of the previously proposed models. It seems this does not have a significant effect on the training time compute and memory complexity of the model. Have the authors conducted any studies to compare the inference-time cost of TM++ compared to other methods?

Q2. As mentioned by authors, some time series tasks (imputation, anomaly detection) benefit more from diverse representations while others like forecasting and classification benefit from consistent representation. Given this, is there any way to leverage a routing model dependent on the proposed task type, which could lower the inference-time cost of this model?

Q3. MTS-Mixer (Li et. al., 2023) presents another approach to channel decomposition which similarly outperformed competing models, but they found the approach worked best with MLPs rather than attention-based models. Have the authors explored this technique for separating from attention mechanisms which could lead to further efficiency and model performance?

---

> ### Author Response · Authors · 2024-11-21
> **Response to Reviewer e5Jj [Part 1]**
>
> We sincerely appreciate reviewer e5Jj  for considering our work is novel and solid, and we greatly appreciate the acknowledgement of our contributions. We have addressed the specific concerns raised by the reviewer as detailed below:
>
> > **W1:** "There is little exploration of scaling of model size, which would be an interesting avenue for validating the model architecture in a zero shot setting. The current zero-shot experiments are primarily in-domain and not cross-task."
>
> (1) We appreciate the reviewer’s insightful suggestions regarding the scaling of model size. As highlighted in Appendix L, exploring the scalability of TimeMixer++ is a direction for future work. In this study, we introduced a powerful backbone model as an initial step towards building a universal time-series pattern machine (TSPM).
>
> (2) We also greatly appreciate your valuable suggestion regarding the zero-shot setting. In response to your concern, we **conducted additional experiments on two well-established cross-domain datasets, M3 and M4, under zero-shot conditions.** The results are summarized below:
>
> | Method        | **TimeMixer++** | FPT   | DLinear | PatchTST | AutoTimes | TimesNet | Nsformer | FEDformer | Informer | Reformer |
> |---------------|-----------------|-------|---------|----------|-----------|---------|----------|-----------|----------|---------|
> | **M4 → M3**  | **12.49**       | 13.06 | 14.03   | 13.06    | 12.75     | 14.17   | 15.29    | 13.53     | 15.82    | 13.37   |
> | **M3 → M4**  | **12.76**       | 13.13 | 15.34   | 13.23    | 13.04     | 14.55   | 14.33    | 15.05     | 19.05    | 14.09   |
>
>
> M4 → M3 means training the model on the datasets of M4 and then evaluating on M3, and vice versa. **TimeMixer++ demonstrates the best performance across both tasks**, outperforming other methods and showcasing its superior ability to generalize temporal patterns without task-specific training.
> TimeMixer++ consistently delivers the lowest errors, with **improvements ranging from 2% to 33%** compared to competing methods. All of these results have been included in Table.13 (Appendix.D) of the _$\underline{\text{revised paper}}$_.
>
> > **Q1:** "The proposed architecture adds significant computational cost to the internal representation of the model compared to vanilla transformers ... Have the authors conducted any studies to compare the inference-time cost of TM++ compared to other methods?"
>
> Thank you for your insightful comments and suggestions regarding the inference-time cost of TimeMixer++. We greatly appreciate your feedback.
>
> **(1). Theoretical Analysis of Time Complexity**
>
> Assuming the input time series has a length of $T$ and a channel count of $C$ (with $C \ll T$ in practice):
>
> - **Vanilla Transformer**: The time complexity is $O(T^2)$ due to the application of full attention along the temporal dimension.
> - **TimeMixer++ Input Projection**: TimeMixer++ applies channel-wise full attention only in the input projection step for channel mixing, which has a time complexity of $O(C^2)$.
> - **Stacked MixerBlock in TimeMixer++**: In the stacked MixerBlock, TimeMixer++ transforms the input time series into time images for more efficient modeling. Specifically, rather than employing full attention, we utilize **efficient dual-axis attention with a time complexity of $O(T \sqrt{T})$** for seasonal-trend decomposition, which generates seasonal and trend images. These images are then processed with efficient convolutional methods. As shown in Figure 13, by avoiding the use of full attention along the temporal dimension, we achieve improvements in training efficiency.
>
>
> **(2). Experimental Results: Inference Time Comparison**
>
> We conducted experiments to evaluate inference time on long-term forecasting tasks. Below is a summary of the findings:
>
> | Model Name   | TimeMixer++ | iTransformer | PatchTST | FEDformer | TimeMixer | TIDE | TimesNet | SCINet |
> |--------------|-------------|--------------|----------|-----------|-----------|------|----------|--------|
> | **Inference Time (ms/iter)** | 90          | 130          | 105      | 240       | 90        | 85   | 160      | 150    |
>
> We can have the following observations:
> - TimeMixer++ achieves **90 ms/iter inference time**, matching the performance of TiDE (85 ms/iter).
> - TimeMixer++ is significantly faster than:
>   - TimesNet (160 ms/iter): **43.75% speedup**
>   - SCINet (150 ms/iter): **40% speedup**
> - Furthermore, as shown in the appendix F of the _$\underline{\text{revised paper}}$_, TimeMixer++ achieves significantly lower prediction error than baselines at comparable inference speed. For example, on ETTm1, it reduces MSE by 10.4% compared to TiDE, with both achieving 85-90 ms/iter inference time.
>
> All results are provided in the Appendix D of the _$\underline{\text{revised paper}}$_.

---

> ### Author Response · Authors · 2024-11-21
> **Response to Reviewer e5Jj [Part 2]**
>
> > **Q2:** "As mentioned by authors, some time series tasks (imputation, anomaly detection) benefit more from diverse representations while others like forecasting and classification benefit from consistent representation. Given this, is there any way to leverage a routing model dependent on the proposed task type, which could lower the inference-time cost of this model?"
>
> Thank you for your insightful suggestion regarding routing mechanisms. Your idea offers valuable guidance for potential future research directions. To enhance efficiency in handling diverse time-series tasks, designing a multitask model with an Mixture-of-Experts (MoE) mechanism could offer a promising solution. In this design, specialized experts would dynamically adapt to different tasks, enabling diverse representations for imputation and anomaly detection, while maintaining consistent representations for forecasting and classification. Incorporating an MoE into the multi-resolution time imaging module and the multi-scale mixing module, as illustrated in Figure 2, could be a promising approach. We appreciate your thoughtful feedback and will consider exploring these possibilities in future work.
>
>
> > **Q3:** "MTS-Mixer presents another approach to channel decomposition which similarly outperformed competing models, but they found the approach worked best with MLPs rather than attention-based models. Have the authors explored this technique?"
>
> Thank you very much for your thoughtful comment.
> - **Channel dependence is indeed considered valuable and informative**, as demonstrated by our ablation experiments in Figure 7 and recent research[1-3].
> - **Both MLP-based and attention-based models are currently the two dominant paradigms for handling channel dependencies.** Following MTS-Mixer, subsequent works such as Crossformer[1], iTransformer[2], and Moirai[3] have shown that attention mechanisms can also effectively capture channel dependencies.
>
> Your insight is valuable. We had conducted experiments to explore different combinations of strategies for channel mixing. The detailed results of these strategies are provided here for your reference:
>
> | Method | MLP-mixing (MSE) | MLP-mixing (MAE) | Attention-mixing (MSE) | Attention-mixing (MAE) |
> |---------|------------------|------------------|------------------------|------------------------|
> | 96      | 0.587            | 0.271            | 0.412                  | 0.297                  |
> | 192     | 0.599            | 0.292            | 0.434                  | 0.289                  |
> | 336     | 0.627            | 0.346            | 0.452                  | 0.297                  |
> | 720     | 0.641            | 0.377            | 0.483                  | 0.311                  |
>
> We evaluated two channel mixing strategies, MLP-mixing and Attention-mixing, on the Traffic dataset, which **comprises 862 channels**.
> The input context length is fixed at 96, and the prediction horizons are set to {96, 192, 336, 720}.
>
> From the table, we observe that in TimeMixer++, **Attention-mixing consistently outperforms MLP-mixing in terms of mean squared error (MSE) for all prediction horizons**, particularly at shorter horizons. For example, at the 96-step prediction horizon, Attention-mixing achieves an MSE of 0.412, which is a **29.8% improvement** over MLP-mixing. Similarly, at the 720-step horizon, the MSE reduction with Attention-mixing is **24.6%**.
>
> These results have also been included in the Appendix D of the _$\underline{\text{revised paper}}$_.
>
> - [1] Zhang, Yunhao, and Junchi Yan. "Crossformer: Transformer utilizing cross-dimension dependency for multivariate time series forecasting." The eleventh international conference on learning representations. 2023.
> - [2] Liu, Yong, et al. "itransformer: Inverted transformers are effective for time series forecasting." arXiv preprint arXiv:2310.06625 (2023).
> - [3] Woo, Gerald, et al. "Unified training of universal time series forecasting transformers." arXiv preprint arXiv:2402.02592 (2024).

---

> > ### Comment · Reviewer_e5Jj · 2024-11-22
> >
> > Thank you for the comprehensive response. It is clear from the additional experiments analyzing cross domain performance and inference complexity that TimeMixer++ retains its advantages over other methods.
> >
> > The considerations for future work and alternative model architectures add additional colour to your work. I will retain my current rating.

---

> > > ### Author Response · Authors · 2024-11-22
> > >
> > > We appreciate your thoughtful review of our work and your recognition of its contributions. Your insightful comments have been useful in helping us improve our paper further.

---

### Author Response · Authors · 2024-11-24
**Summary of Revisions**

We sincerely thank all the reviewers for their detailed reviews,
which are instructive for us to improve our paper further.
We have revised the paper according to the comments, and the
edits have been highlighted in **RED**.

This paper introduces TimeMixer++, a novel and general time series pattern machine for universal predictive analysis.
TimeMixer++ disentangles seasonality and trend patterns
in latent space through Time Image Decomposition (TID) and adaptively integrates these patterns using
Multi-Scale Mixing (MCM) and Multi-Resolution Mixing (MRM). **It achieves state-of-the-art performance
across eight diverse time series tasks, outperforming 27 baselines on 30 benchmarks**.

The reviewers generally held positive opinions of our paper, noting that the proposed method **"demonstrates innovation"**
and
is **"very interesting"**; **"the manuscript and appendix are well-prepared"**;
**"the experiments were very thorough"** and **"comprehensive"**;  and that
we **"present SOTA results"**.


The reviewers also raised insightful and constructive concerns. Here is the summary of the major revisions:

- **Provide cross-domain zero-shot forecasting results (Reviewer e5Jj)**:
Following the reviewer's suggestion, we conducted a zero-shot forecasting experiment
on two additional cross-domain datasets, M3 and M4. TimeMixer++ continues to perform best
in this setting. For complete results, please refer to Appendix D.

- **Add inference time comparison (Reviewer e5Jj)**:
To address the reviewer's request, we have reported the inference time of TimeMixer++
along with seven baselines on long-term forecasting tasks to provide a more comprehensive
understanding of the efficiency analysis. TimeMixer++ achieves an inference speed of 90 ms/iter,
which matches that of TiDE and surpasses other models. For the complete results, please refer to Appendix D.

- **Add ablations on the channel mixing module (Reviewer e5Jj)**:
We have included ablation results evaluating the choice of channel mixing based on a multi-layer
perceptron (MLP). The results demonstrate that our proposed attention-mixing approach consistently
outperforms MLP-mixing. The updated results can be found in Appendix D.

- **Update the figures (Reviewer LKyA):**
Following the reviewer's suggestion, we have updated Figure 1 in the main text and Figure 12 in Appendix E.


- **Clarify differences among Koopa, TFDNet, and FEDNet (Reviewer 9h69)**:
We clarify that TimeMixer++ is distinct from Koopa, TFDNet, and FEDNet in its technical design, pattern
learning capabilities, and respective objectives within the literature.
Additionally, we have updated the Introduction and Related Work sections, adding relevant citations to
highlight our contributions.

The valuable suggestions from reviewers are very helpful for us to revise the paper to a better shape. We'd be
happy to answer any further questions.

---

### Comment · Area_Chair_eU5n · 2024-11-25
**Acknowledge the author responses**

Dear Reviewers,

Thank you very much for your effort. As the discussion period is coming to an end, please acknowledge the author responses and adjust the rating if necessary.

Sincerely,
AC

---

### Public Comment · ~Kashif_Rasul1 · 2025-02-12
**MixerBlock questions**

Why are you referring to the inputs of the mixer block as time series? presumably, at that point, the inputs are representations of the original multivariate data of feature-length $d_\mathrm{model}$  coming from the channel attention?

why are seq of representations $\mathbf{x}^l_m$ a 1-D time series in line 243? If it's the output from the channel attention it has shape $d_\mathrm{model} \times T/2^m$ no?

What is the motivation for decomposing the representations from the channel attention layer? My intuition was that at that point the output from the channel attention would not resemble time series, like trend or seasonality, even more so up the layers?

Did you compare your model to prototypical mixer architectures like MLP-mixer, Conv-Mixer, and ViT style models that can easily be up-cylced for the multivariate forecasting task?

thank you!

---

> ### Public Comment · ~Shiyu_Wang3 · 2025-02-14
> **Thank you for your constructive comments.**
>
> Dear Kashif,
>
> Thank you very much for your interest in our work and for your constructive comments. We greatly appreciate your thoughtful questions and suggestions, which have provided us with valuable insights. Below, we address your questions in detail:
>
> **1. Regarding the inputs to the mixer block and the notation of time series representations**
>
> Thank you for pointing this out. You are correct that the inputs to the mixer block are time-series representations with the shape $\mathbb{R}^{\lfloor T/2^M \rfloor \times d_{\text{model}}}$, and $x_m^l$ in line 243 also represents a time series with the same shape. We refer to these as "time series" because they retain the temporal dimension, consistent with conventions in prior works such as Autoformer[2]. Specifically, the "1D time series" terminology in line 243 follows the usage in TimesNet[4], where a time series with the shape $\mathbb{R}^{T \times C}$ is described as a 1D time series.
>
> We sincerely are grateful for your suggestion, as we recognize that this terminology might lead to unnecessary ambiguity or misunderstanding. To improve clarity, we will revise the phrasing to "time-series representation" in the final version of our paper. Thank you for bringing this to our attention.
>
> **2. On the motivation for decomposing representations from the channel attention layer**
>
> The channel attention layer in our model serves as part of the input projection, similar to the input projection embedding approaches introduced since Informer[1]. In these methods, the channel dimension of multivariate time series is embedded, while the temporal and channel dimensions remain orthogonal. Even after embedding, the temporal variation characteristics are preserved, and the decomposition primarily focuses on the temporal dimension[6].
>
> Our motivation for adopting decomposition stems from the successes of prior works such as Autoformer[2], Fedformer[3], and MICN[5], which have demonstrated the effectiveness of decomposition in capturing temporal patterns in deep spaces[6]. While our work builds upon these ideas, we acknowledge that decomposition methods in deep learning for time series remain an open area of research with significant potential for further exploration.
>
> **3. Comparison with prototypical mixer architectures**
>
> Indeed, pioneering works like MLP-Mixer have inspired our exploration of mixer architectures for time series. For handling 2D time series, we are currently experimenting with more straightforward approaches. Your observation is insightful, and we are aware of recent works leveraging ViT for 2D time series[7]. This promising direction highlights the potential for deeper intersections between time-series and CV fields, paving the way for multimodal methods that seamlessly integrate time-series and image modalities and opening new possibilities for multivariate forecasting.
>
> Once again, thank you for your thoughtful observations and for taking the time to share your suggestions. Your insights have been truly inspiring, and we sincerely appreciate your engagement.
>
> Best regards,
>
> Authors.
>
> [1] Zhou, Haoyi, et al. (2021). Informer: Beyond efficient transformer for long sequence time-series forecasting. In *Proceedings of AAAI Conference on Artificial Intelligence*.
>
> [2] Wu, Haixu, et al. (2021). Autoformer: Decomposition transformers with auto-correlation for long-term series forecasting. *Advances in Neural Information Processing Systems*.
>
> [3] Zhou, Tian, et al. (2022). FEDformer: Frequency enhanced decomposed transformer for long-term series forecasting. In *Proceedings of the 39th International Conference on Machine Learning*.
>
> [4] Wu, Haixu, et al. (2023). TimesNet: Temporal 2D-variation modeling for general time series analysis. In *International Conference on Learning Representations*.
>
> [5] Wang Huiqiang, et al. (2023). MICN: Multi-scale local and global context modeling for long-term series forecasting. In *International Conference on Learning Representations*.
>
> [6] Wang Yuxuan, et al. (2024). Deep Time Series Models: A Comprehensive Survey and Benchmark. In *Transactions on Pattern Analysis and Machine Intelligence*.
>
> [7] Zhong Siru, et al. (2025). Time-VLM: Exploring Multimodal Vision-Language Models for Augmented Time Series Forecasting. In *arXiv preprint*.

---

### Public Comment · ~Danny_Dongyeop_Han1 · 2025-03-01
**Questions regarding the paper**

First, thank you for your amazing work! I am interested in applying your model in our domain (medical timeseries), and had a few questions if you don’t mind.

**Questions regarding pretraining** : In Appendix L “Limitations and Future Work”, you say TimeMixer++ can be an effective backbone model, and will explore scaling the model. We also want to do this for our domain, but realized that some aspects of TimeMixer++ are not well suited for pretraining and wanted to ask your opinion if you don’t mind :
1. **Do you have any suggestions on what type of SSL pretext task could be used?**
    - As far as I know, current timeseries foundation models (e.g., MOIRAI, TimeMOE, Timer-XL) use GPT/BERT-like pretext tasks for pretraining. However, because your model don’t use patches (tokens), using GPT/BERT-like pretext tasks seems impossible. Do you have suggestions on how to resolve this (either by modifying the model or choosing a different pretext task?)
    - Also, it seems that current time-series foundation models only use single patch sizes as inputs. As an outsider this perplexes me. If possible, could you please tell me why? Your paper and others seem to argue that using single sized tokens to represent timeseries is not ideal, yet many foundation models use tokens.
2. **How would you modify your model to adapt to varying number of input variables?**
    - Due to how input projection is designed, it seems that the model cannot be trained on datasets with different number of channels. Do you have any suggestions on how to resolve this?
    - Also, was there a reason why you mixed the channels early (during input projection from $C$ to $d_{model}$), and keep the channel dimension and model them in the MixerBlock?
3. **Do you think TimeMixer++ could adapt to different input lengths?**
    - As far as I understand, it seems that at least computationally the model can be applied to different input lengths. Is my understanding correct? Do you think that fine-tuning the model to a task with much shorter input length compared to the pretraining dataset will be ok?



**Questions regarding the model itself :**

1. **Would your model still work well if the data always has FFT amplitudes with a 1/f trend?**  : Due to the inherent 1/f trend in our data, selecting the top-k frequencies during MRTI would always return the k lowest frequency values, making the model ignore higher frequency features (which are crucial for certain classification tasks in our field). Do you think it will be OK for us to use a different top-k frequency selection method (e.g., selecting the top-k after detrending the 1/f component)? Or should we stick to the original method?
2. **Is $p_k$ actually $p_{k,m}$? :** It seems that $p_k$, obtained from running FFT and picking the Top-K periods is the value with respect to the scale $M$. Wouldn't it be that for different scales the same period length would be different number of timepoints (for example, $p_k$ obtained in scale $M$ should equal $2p_k$ in scale $M-1$?)
3. **Details on the 2D convolutions you used in Time Image Decomposition and Multiscale Mixing? :** Could you please tell the kernel sizes and such that were used for the 2D convolutions you used in your model? It would greatly help in understanding your paper!

Thank you!

---

> ### Public Comment · ~Shiyu_Wang3 · 2025-03-17
> **Part 1: Thank you for your thoughtful comments.**
>
> Dear Danny,
>
> Thank you for your thoughtful and detailed questions! We're very glad to see your interest in applying TimeMixer++ to medical time series data. Below, we’ve provided answers to your questions:
>
> **1. Regarding the SSL pretext task on point-wise token or patch-wise token**
>
> Your question is profound and highly relevant, as it touches on a fundamental issue in time series modeling: how to define and process tokens for time series data. Currently, there are two mainstream approaches: **point-wise tokens** [1] [5] [6] and **patch-wise tokens** [4]. Additionally, Autoformer[2] and iTransformer[3] propose a third perspective, namely **series-wise tokens**. To address this question, it is necessary to revisit the concept of a token.
>
> In its simplest terms, a token is a term borrowed from NLP[9,10], where **a token is the smallest unit of text processed by a model, which can be a word, subword, character, or special symbol, depending on the tokenization method used**. This concept has been adapted for time series modeling. Fundamentally, a token represents the smallest unit of data processed by a model. A straightforward approach is to treat each individual point in a time series as a token, which has been the dominant paradigm in time series modeling for a long time.
>
> However, starting with PatchTST, inspired by the success of Vision Transformers (ViT)[8] in the image domain, a new idea emerged: introducing the concept of **patches** into time series modeling. This involves grouping subsequences into patches and treating each patch as a token. This raises a critical question: **how should patches be defined in time series data?** The current approach, largely following PatchTST, involves extracting fixed-length subsequences from the original series using a sliding window. However, this approach faces a significant challenge in time series data: the trends and periodicities of different sequences often vary greatly. Time series data contain both rich global features and diverse local features, and the choice of patch length and sliding window strategy can significantly impact the model. This essentially becomes a hyperparameter selection problem. Poor choices can not only degrade prediction accuracy but may also distort the inherent characteristics of the time series.
>
> Given this, **point-wise tokens**, which preserve the most complete temporal information, remain a reasonable choice. On the other hand, iTransformer[3] takes an entirely different approach by treating the entire input sequence as a single token, offering a novel perspective.
>
> Returning to the original question, none of these approaches—whether point-wise tokens, patch-wise tokens, or series-wise tokens—are likely to be perfect solutions. Time series data differ significantly from text and images, and directly transplanting solutions from these domains may not always be appropriate. This is an area that requires further exploration.
>
> That said, **patch-wise tokens** have a notable advantage in the context of pretraining time series models: they reduce the number of tokens compared to point-wise tokens (since multiple points are grouped into a single patch), thereby improving training and inference efficiency. This is likely one of the reasons why many time series foundation models have adopted this approach. Time-MoE[7], like our approach, continues to use point-wise tokens. However, it leverages the design of **Multi-resolution Forecasting**, which helps improve efficiency to some extent.
>
>
> **2. Regarding the any-variate input variables**
>
> To be frank, our work has not focused on the problem of **any-variate input variables**, but it is indeed a key and trending topic in the time series domain. To achieve any-variate input variables, significant modifications to the current model architecture would be required. In fact, handling any-variate inputs is an inherent advantage of transformers, as the attention mechanism is naturally capable of addressing this issue. However, as you pointed out, using projections to transform variable dimensions imposes parameterized constraints, which limit the model's ability to handle truly any-variate input variables.
>
> As mentioned earlier, iTransformer is a **series-wise token** model, and its architecture is specifically designed to address this challenge. If we aim to achieve similar functionality, we could take inspiration from its architectural design.
>
> Regarding the question of **why channel mixing is performed during the input projection stage**, the reason lies in the need to enable early interactions between different channels. As the input to subsequent modules, channel mixing at the input projection stage allows for the earliest possible exchange of information across channels. This is analogous to why it is necessary to embed time series points at the very beginning of the process.

---

> ### Public Comment · ~Shiyu_Wang3 · 2025-03-17
> **Part 2: Thank you for your thoughtful comments.**
>
> **3. Regarding the any-variate input lengths**
>
> In fact, handling **any-variate input lengths** is typically a strength of transformer models[10], particularly those with **decoder-only** or **encoder-decoder** architectures, which are inherently designed to process variable-length sequences. On the other hand, **encoder-only** architectures are generally limited to handling fixed-length sequences. Modifications to the underlying architecture would be necessary to enable our model to handle variable-length input sequences. This could involve approaches such as padding or adopting a decoder-only architecture. I would suggest referring to the design of Time-MoE[7], which is specifically tailored for variable-length sequences and serves as a large-scale time series model.
>
>
> **4. Regarding the data always has FFT amplitudes with a 1/f trend**
> You mentioned that the inherent 1/f trend in your data causes the Top-K frequency selection in MRTI to always return the lowest K frequencies, potentially overlooking high-frequency features that are crucial for certain classification tasks. Our recommendations are as follows:
>
> **Using a Detrending Method**: If high-frequency features are critical to your task, applying a detrending method (e.g., removing the 1/f component) before selecting the Top-K frequencies is a reasonable choice. This approach will help you fairly select key features across the frequency spectrum without being biased by the 1/f trend.
>
> **Retaining the Original Method**: If your task is more sensitive to low-frequency features, or if the 1/f trend itself contains important information, retaining the original method might be preferable. The final choice should depend on your understanding of the data characteristics and task requirements.
>
> Whether to modify the Top-K selection method ultimately depends on the needs of your task. If high-frequency features are indeed essential, we recommend trying detrending before frequency selection.
>
>
> **5. Regarding the $p_k$ or $p_{k,m}$**
>
> The reason we do not run FFT and pick the Top-K periods at different scales is that the Top-K periods calculated at the original scale cannot be directly applied to the downsampled scales. After downsampling, the lower scales already incorporate global, macro-level information, making it more appropriate to use a unified set of Top-K periods. Additionally, calculating separate Top-K periods for each scale would introduce a nested loop structure, significantly increasing computational complexity.
>
> **6. Regarding the Details on the 2D convolutions**
>
> We adopted TimesNet's implementation of the Inception_Block [5] and followed the same configuration. You may refer to it for further details.
>
> Thank you once again for your interest in our work. Medical time series is a vast and significant field, and exploring this area is crucial for real-world applications. Your insightful questions have been incredibly valuable and have provided us with meaningful inspiration.
>
> Best regards,
>
> Authors.
>
> [1] Zhou, Haoyi, et al. (2021). Informer: Beyond efficient transformer for long sequence time-series forecasting. In *Proceedings of AAAI Conference on Artificial Intelligence*.
>
> [2] Wu, Haixu, et al. (2021). Autoformer: Decomposition transformers with auto-correlation for long-term series forecasting. *Advances in Neural Information Processing Systems*.
>
> [3] Yong, Liu, et al. (2024). iTransformer: Inverted Transformers Are Effective for Time Series Forecasting. In *International Conference on Learning Representations*.
>
> [4] Nie, Yuqi, et al. (2023). A Time Series is Worth 64 Words: Long-term Forecasting with Transformers. In *International Conference on Learning Representations*.
>
> [5] Wu, Haixu et al. (2023). TimesNet: Temporal 2D-variation modeling for general time series analysis. In *International Conference on Learning Representations*.
>
> [6] Wang, Yuxuan, et al. (2024). Deep Time Series Models: A Comprehensive Survey and Benchmark. In *Transactions on Pattern Analysis and Machine Intelligence*.
>
> [7] Shi, Xiaoming, et al. (2025). Time-MoE: Billion-Scale Time Series Foundation Models with Mixture of Experts. In *International Conference on Learning Representations*.
>
> [8] Alexey Dosovitskiy, et al. (2021). An Image is Worth 16x16 Words: Transformers for Image Recognition at Scale. In *International Conference on Learning Representations*.
>
> [9] Rico Sennrich, et al. (2016). Neural Machine Translation of Rare Words with Subword Units. In *Annual Meeting of the Association for Computational Linguistics*.
>
> [10] Ashish Vaswani, et al. (2017). Attention Is All You Need. In *Conference on Neural Information Processing System*.

---

### Meta-Review · Area_Chair_eU5n · 2024-12-19

**Metareview:**

This paper presents TimeMixer++, a general-purpose model for various time series tasks, including forecasting, classification, anomaly detection, and imputation.  The proposed model achieves state-of-the-art performance across 8 time series analytical tasks, consistently surpassing both general-purpose and task-specific models.  These results are obviously impressive, and this paper is worthwhile to receive further attention.  Thus, I would like to recommend an accept as a spotlight paper.

**Additional Comments On Reviewer Discussion:**

All the reviewers were satisfied with the authors' responses during the discussion period.  One reviewer increased his/her rating.

---

### Decision · Program_Chairs · 2025-01-22

Accept (Oral)